# Somatic mosaicism in ALS and FTD identifies focal mutations associated with widespread degeneration

Zinan Zhou [1,2,3,12], Junho Kim[1,2,3,4,12], August Yue Huang [1,2,3,12], Matthew Nolan[5], Junseok Park[1,2,3], Ryan Doan[1,3], Taehwan Shin[1,2,3], Michael B. Miller [1,6], Mingyun Bae[1,2,3], Boxun Zhao [1,2,3], Jinhyeong Kim [4], Brian Chhouk[1,2,3], Katherine Morillo[1,2,3], Rebecca C. Yeh[1,2,3], Connor Kenny[1,2,3], Jennifer E. Neil[1,2,3,7], Chao-Zong Lee[5], Takuya Ohkubo [8,9], John Ravits[9], Olaf Ansorge [10], Lyle W. Ostrow [11], Clotilde Lagier-Tourenne [5,13] ✉, Eunjung Alice Lee [1,2,3,13] ✉ & Christopher A. Walsh [1,2,3,7,13] ✉

Although mutations in many genes cause familial amyotrophic lateral sclerosis (ALS) and frontotemporal dementia (FTD), most cases are sporadic (sALS and sFTD) with unclear etiology. Here we tested whether somatic mutations contribute to sALS and sFTD by deep targeted sequencing of 88 neurodegeneration-related genes in postmortem brain and spinal cord samples from 399 sporadic cases and 144 controls. Predicted deleterious somatic variants in ALS/FTD genes were observed in 2.1% of sporadic cases lacking deleterious germline variants. These variants occurred at very low allele fractions (typically <2%) and were often focal and enriched in disease-affected regions. Analysis of bulk RNA-sequencing data from an additional cohort identified deleterious somatic variants in *DYNC1H1* and *LMNA*, genes associated with pediatric motor neuron degeneration. Targeted long-read sequencing further identified one sFTD case with de novo somatic *C9orf72* repeat expansions. Together, these findings suggest that rare, focal somatic variants can contribute to sALS and sFTD and drive widespread neurodegeneration.

Amyotrophic lateral sclerosis (ALS), a disease in which premature loss of upper motor neurons (UMNs) and lower motor neurons (LMNs) leads to fatal paralysis, shows clinical, genetic and pathological overlap with frontotemporal dementia (FTD), a neurodegenerative disorder characterized by behavioral, language and memory dysfunction[1]. A total of 5–22% of individuals with ALS develop FTD, and ~15% of those with FTD eventually develop ALS[2]. ALS and FTD also share common pathology, with cytoplasmic inclusions of TAR DNA binding protein (TDP-43) found in almost all ALS brains and in half of FTD brains[3,4]. ALS typically begins focally and spreads regionally as the disease progresses[5,6], although whether degeneration begins in UMNs, LMNs or both simultaneously

has remained controversial[7,8], with some studies suggesting that focality can manifest independently in UMNs and LMNs[5,9]. TDP-43 pathology also follows stereotypical patterns in ALS and FTD brains[9–11], thought to reflect focal onset and intercellular transmission of TDP-43 inclusions in a prion-like manner, as shown in cell and animal models[12–18].

Whereas over 30 genes are implicated in ALS and FTD[19], most causative genes are linked to familial ALS (fALS) and familial FTD (fFTD), while 90–95% cases are sporadic ALS (sALS) and sporadic FTD (sFTD) without a family history[20]. The focal onset of ALS and FTD, their stereotypical spread and the increased risk in smokers[21] have raised interest in potential roles of somatic mosaic mutations in the pathogenesis of ALS and

A full list of affiliations appears at the end of the paper. ✉e-mail: clagier-tourenne@mgh.harvard.edu; ealee@childrens.harvard.edu; christopher.walsh@childrens.harvard.edu

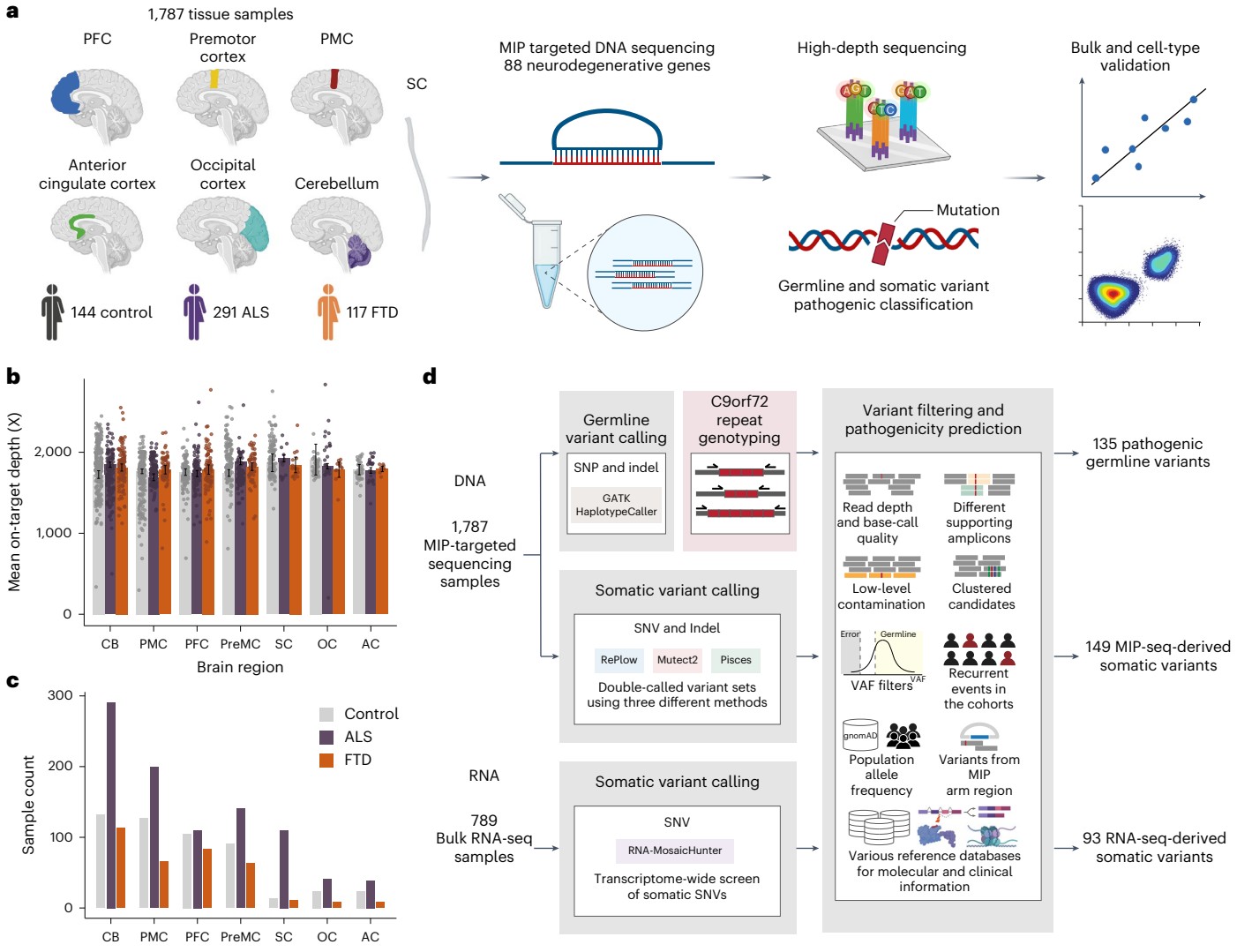

**Fig. 1 | Experimental and analysis strategies. a**, Overall scheme of the experiments. Genomic DNA isolated from 1,787 postmortem tissue samples of multiple brain regions and SCs of 144 control, 291 ALS and 117 FTD cases were used for MIP capture sequencing with ultrahigh depth. **b**,**c**, Mean sequencing depth (**b**) and number of tissue samples (**c**) in different brain regions and SCs of control, ALS and FTD cases. Control, $n = 516$; ALS, $n = 938$; FTD, $n = 375$. Please note that 42 samples from nine ALS–FTD cases were included in both conditions. Error bars = 95% CI. **d**, Methodological pipelines to identify germline and somatic variants. Germline variants were called by GATK HaplotypeCaller. *C9orf72* genotypes of ALS and FTD cases were determined by RP-PCR. Somatic variants were called by RePlow, Mutect2 and Pisces. Additional somatic variants were called from 789 bulk RNA-seq profiles of multiple brain regions and SCs of ALS cases generated by the NYGC ALS Consortium using RNA-MosaicHunter. Schematics in **a** and **d** created in BioRender; Zhou, Z. https://biorender.com/s3p5mmn (2026). CB, cerebellum; PreMC, premotor cortex; OC, occipital cortex; AC, anterior cingulate cortex.

FTD[22]. Somatic mutations are increasingly recognized as prevalent in normal-appearing tissues, but those responsible for neurological conditions are often limited to the central nervous system (CNS)[23] and hence undetectable by DNA sequencing of non-CNS tissues. Recent studies have evaluated the contributions of somatic mutation to Alzheimer's and Parkinson's diseases directly using postmortem brain tissues[24].

In this study, we assessed potential contributions of somatic variants—distinguished by their variant allele frequencies (VAFs)—to sALS and sFTD using deep sequencing of a panel of neurodegeneration/dementia-associated genes on postmortem tissues of various brain regions and spinal cords (SCs) from 399 unique sALS and sFTD cases. Our study identified new predicted deleterious somatic variants in 2.1% of sALS and sFTD cases without pathogenic or predicted deleterious germline variants. Protein-altering (missense/nonsense/frameshift) somatic variants showed enrichment in sALS and sFTD cases and in disease-affected brain regions, supporting roles in disease

pathogenesis. Regional analysis revealed the focal nature of predicted deleterious somatic variants in the primary motor cortex (PMC) and SC, not only supporting independent disease initiation in UMNs and LMNs, but also strongly supporting models of ALS and FTD in which the disease spreads beyond a relatively confined region containing a somatic variant. Complementary analyses of bulk RNA sequencing (RNA-seq) and targeted long-read sequencing of brain and SC tissues further revealed somatic variants in genes not previously associated with ALS or FTD, as well as a de novo somatic *C9orf72* repeat expansion. Together, our study opens new avenues for understanding the etiology of sALS and sFTD.

## Results

### Deep targeted sequencing of neurodegenerative genes in sALS and sFTD brains

To directly detect somatic variants in sALS and sFTD brains, we obtained postmortem frozen tissues of several brain regions and SCs from

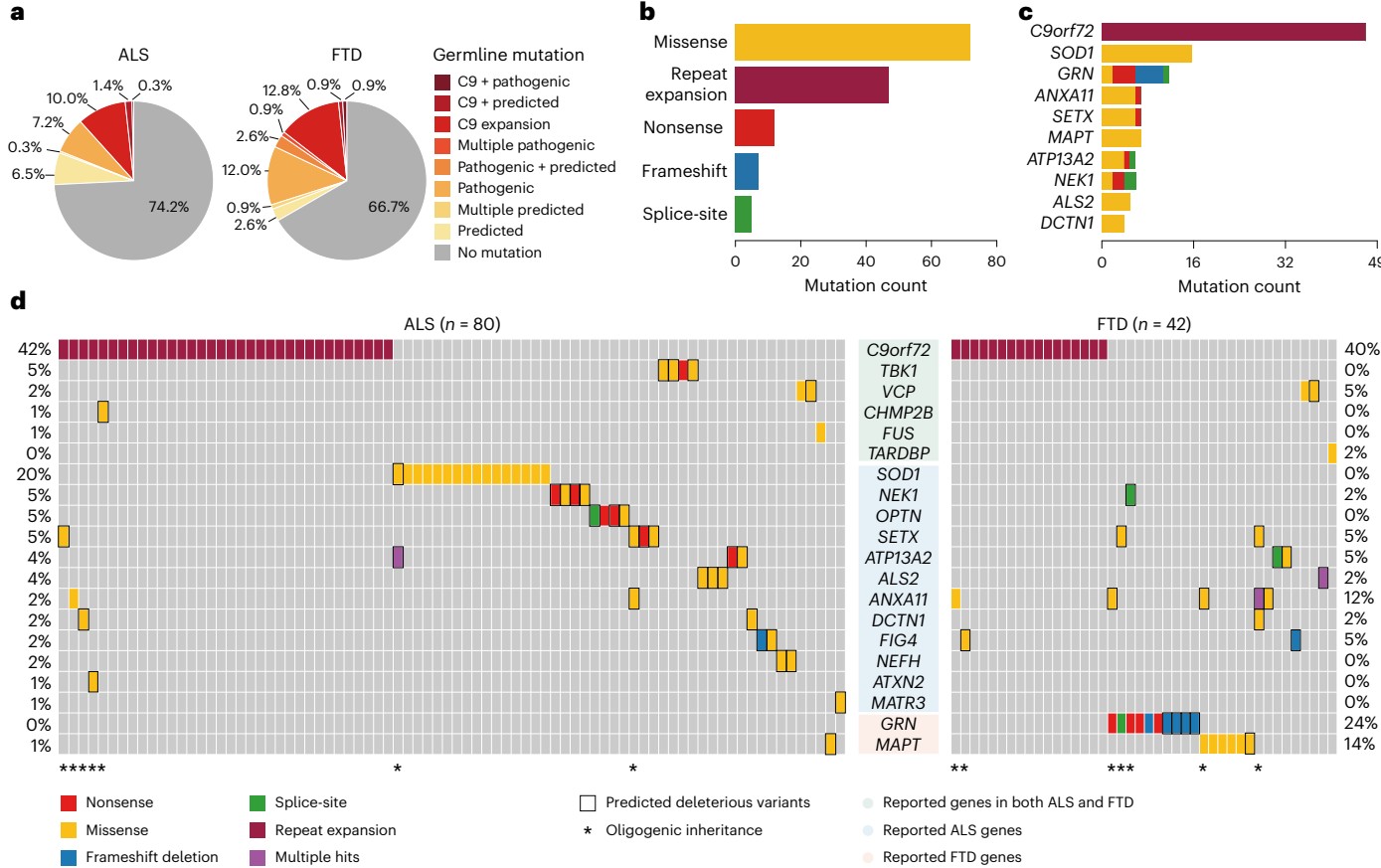

**Fig. 2 | *C9orf72* repeat expansion and pathogenic germline variants in ALS/FTD genes are prevalent in ALS and FTD. a**, Proportions of ALS and FTD cases with *C9orf72* repeat expansion, pathogenic and predicted deleterious germline variants in ALS/FTD genes. Cases with multiple predicted deleterious variants are indicated with '+' sign. Please note that cases with a heterozygous variant in recessive genes (for example, *ALS2*, *ATP13A2*) are not included in the proportions. **b**, Distribution of *C9orf72* repeat expansion, pathogenic and predicted deleterious germline variants in ALS/FTD genes classified by variant types. **c**, Ranking of the top ten mutated ALS/FTD genes. **d**, Visualization of ALS and FTD cases (vertical columns) with pathogenic and predicted deleterious germline variants (horizontal rows) in ALS/FTD genes. Color codes indicate the types of variants. Rectangular outline represents predicted deleterious variants. Genes are grouped by their known involvement in diseases. An asterisk indicates cases with multiple pathogenic or predicted deleterious variants.

individuals diagnosed with sALS or sFTD, as well as from age-matched controls through the Massachusetts Alzheimer's Disease Research Center, Oxford Brain Bank and Target ALS Foundation (Fig. 1a and Supplementary Table 1). Additional brain tissues from ALS, FTD and control cases, without a record of family history but with an age of death above 45 years old, were also obtained from the NIH NeuroBioBank. We performed molecular inversion probe (MIP)-panel sequencing[25] of 88 neurodegeneration-associated genes at ~1,800× deduplicated depth across 1,787 samples from 291 ALS, 117 FTD and 144 neurotypical control individuals (Fig. 1a,b, Supplementary Fig. 1, Supplementary Table 2 and Supplementary Note).

### Pathogenic germline variants in sALS and sFTD cases

We first identified pathogenic germline single-nucleotide variants (SNVs) and short insertions and deletions (indels) using Genome Analysis Toolkit (GATK), followed by stringent filtering (Fig. 1d), with functional annotation and deleteriousness prediction performed using ANNOVAR[26] and multiple clinical databases. In addition, *C9orf72* repeat expansions, the most common inherited cause of ALS and FTD[27,28], were genotyped by a repeat-primed PCR (RP-PCR) assay (Supplementary Fig. 2). Overall, 20.6% (60 of 291) ALS, 25.6% (30 of 117) FTD and 0.7% (1 of 144) control cases carried *C9orf72* repeat expansions or pathogenic germline mutations in ALS and FTD genes (Fig. 2a and Supplementary Tables 3 and 4). Missense variants

represented the most prevalent variant type (Fig. 2b). *C9orf72* was the most frequently mutated gene, followed by *SOD1* in ALS, and *GRN* and *MAPT* in FTD (Fig. 2c,d). The overall fractions of *C9orf72* repeat expansion carriers (10.6% in ALS; 12.0% in FTD) were slightly higher than previously reported in sporadic cases, but still lower than in familial cases[29–31].

In addition, predicted deleterious germline variants in dominant ALS/FTD genes were found in an additional 14.1% of ALS, 18.8% of FTD and 4.9% of control cases with significant enrichment observed in both ALS (odds ratio (OR) = 3.20, 95% confidence interval (CI) = 1.37–8.69, $P = 3.2 \times 10^{-3}$) and FTD (OR = 4.51, 95% CI = 1.77–13.01, $P = 5.5 \times 10^{-4}$) cases (Fig. 2a and Supplementary Tables 3 and 4). While most predicted deleterious variants were nonsynonymous SNVs requiring functional validation, two new *GRN* frameshift variants (p.L46Rfs*18 and p.D250Tfs*6) identified in FTD cases were probably pathogenic (Supplementary Table 3), as loss-of-function *GRN* mutations are known to cause FTD in a dominant manner[32,33]. Consistent with previous reports, we found evidence of possible oligogenic inheritance. This includes 1 case carrying both *C9orf72* expansion and a pathogenic variant in *ANXA11*, as well as 12 additional cases carrying combinations of pathogenic and predicted deleterious variants (Fig. 2d and Supplementary Table 3)[34–37]. Interestingly, we also identified cross-disease variants (variants in ALS genes in FTD and vice versa), highlighting potential shared genetic mechanisms (Fig. 2d).

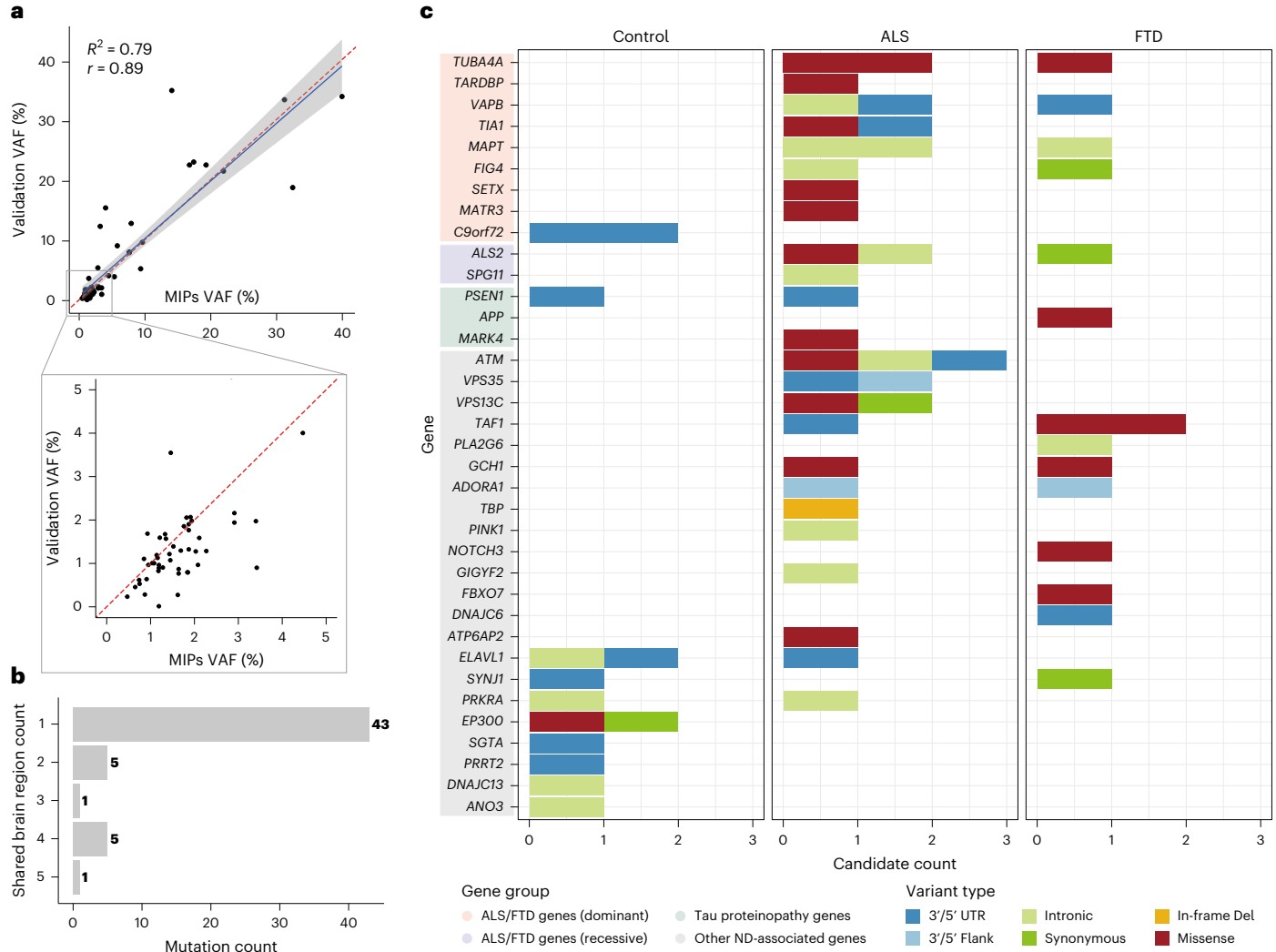

**Fig. 3 | Somatic variants in MIP sequencing data tend to be focal, protein-altering and are almost exclusively restricted to disease cases. a**, The observed VAFs of somatic variants in amplicon sequencing validation were consistent with the VAFs in original MIP sequencing. Sixty-four somatic variants were validated and included in the plot. Shaded band indicates 95% CI of the fitted linear regression. The red dashed line indicates the identity line ($y = x$), representing perfect agreement between MIP-derived and validation VAFs. **b**, Total somatic variant counts classified by the number of brain regions in which a given variant was identified. **c**, Distribution of somatic variants in all neurodegenerative genes. Genes are categorized into four groups—ALS/FTD-related dominant genes, ALS/FTD-related recessive genes, Tau proteinopathy-related genes and other neurodegeneration/dementia-associated genes. Color codes indicate variant types. Please note that somatic variants identified in controls are unlikely to alter function, with just one missense variant (red) and the remaining being synonymous or noncoding substitutions. ND, neurodegeneration.

## Identification of somatic SNVs and indels from MIP sequencing data

We developed a custom pipeline integrating RePlow[38], Mutect2 (ref. 39) and Pisces[40] for calling somatic SNVs and indels in our MIP sequencing data (Fig. 1d). We selected somatic variants identified by at least two of the three callers (double-called variants) followed by multistep variant filters to remove false positive candidates. Unlike heterozygous germline variants with VAFs around 50%, heterozygous somatic variants have VAFs less than 50%, and we only called somatic variants with VAFs below 35%. To benchmark our pipeline, we performed spike-in experiments by mixing two human samples from the Genome in a Bottle Consortium at low VAFs and observed high sensitivity and precision with a low false positive rate (FPR) across VAFs (Extended Data Fig. 1 and Supplementary Note).

We applied our custom pipeline to 1,787 samples to obtain initial somatic variant candidates. Among these, we identified and removed candidates arising from potential sample contamination in 29 samples (Supplementary Note). The remaining call set contained 98 unique somatic SNV and indel candidates (a total of 149 candidates, including shared ones observed in multiple regions of the same individuals; Supplementary Table 5). Variants with low VAFs (<5%) were more common in disease cases than in normal controls (Extended Data Fig. 2). All somatic candidates were validated by deep amplicon sequencing, except one for which primers could not be designed; 34 exonic candidates were additionally assessed by droplet digital PCR (ddPCR). Thirteen candidates were actually germline variants, consistent with known VAF deviation in MIPs sequencing due to probe hybridization and amplicon design biases[41,42]. After excluding these, 64 of 85 candidates (75.2%) were validated as somatic (Supplementary Table 6). The VAFs of validated amplicon sequencing candidates showed a strong correlation with their original VAFs in the MIP sequencing data (Fig. 3a). All subsequent somatic variant analyses were conducted using only the validated candidates.

## Somatic variants in disease-relevant genes are enriched in ALS and FTD cases lacking pathogenic germline variants

To examine the burden and potential roles of somatic variants in ALS and FTD, we focused on cases that lack pathogenic or predicted deleterious germline variants in ALS/FTD genes (germline-free cases), including cases with heterozygous germline variants in recessive ALS/FTD genes (for example, *ATP13A2*, *ALS2*) that were predicted to be deleterious. Fifty-five unique somatic variants across the targeted regions were identified among 696, 243 and 516 samples from 216 germline-free ALS cases, 78 germline-free FTD cases and 144 neurotypical controls, respectively. Most of them (78.2%, 43 of 55) were focal to a single tissue region and present at very low VAFs (Fig. 3b and Extended Data Fig. 2), consistent with late-arising[43], CNS-restricted events. Mutational signature analysis[44] revealed predominance of clock-like signatures (SBS5 and SBS1; Extended Data Fig. 3), consistent with previous studies of somatic mutagenesis in the normal human brain[45,46].

Across all targeted neurodegeneration/dementia-related genes, protein-altering somatic variants showed clear separation between disease and control groups (Fig. 3c), with only one such variant observed in controls compared with 15 in ALS and 7 in FTD cases. Both exonic and protein-altering somatic variants were significantly enriched in ALS and FTD (Fig. 4a; exonic, $P = 0.044$ and $P = 0.0014$; protein-altering, $P = 0.019$ and $P = 0.010$; linear mixed model; Supplementary Note), whereas intronic and noncoding somatic variants showed no enrichment. Consistent with this, ratios of nonsynonymous to noncoding variants were increased in ALS and FTD at both group and individual levels (Extended Data Fig. 4), indicating disease-specific accumulation of potentially damaging variants.

When restricted to ALS/FTD genes, exonic and protein-altering variants in established ALS genes were specifically enriched in germline-free ALS cases (Fig. 4b; exonic, $P = 0.046$; protein-altering, $P = 0.026$; linear mixed model). In contrast, germline-free FTD cases showed no enrichment in known FTD genes but showed significant enrichment in other neurodegeneration/dementia-related genes (Fig. 4a,b and Extended Data Fig. 5), consistent with broader pathological heterogeneity of FTD, and perhaps implicating a shared genetic architecture with other neurodegenerative diseases. No protein-altering variants in ALS/FTD genes were observed in controls or in ALS/FTD cases carrying pathogenic germline variants (Supplementary Tables 6 and 7).

Somatic variants also exhibited region-specific enrichment in disease-affected regions of germline-free ALS and FTD cases (Fig. 4c and Extended Data Fig. 6). Across all targeted genes, enrichment was observed in the PMC in ALS cases (Extended Data Fig. 6; all, $P = 0.020$; exonic, $P = 0.026$; protein-altering, $P = 0.012$; linear mixed model) and in the prefrontal cortex (PFC) of FTD brains (Extended Data Fig. 6; exonic, $P = 2.3 \times 10^{-3}$; protein-altering, $P = 7.7 \times 10^{-4}$; linear mixed model). In contrast, the premotor cortex—located immediately between the PMC and PFC—showed no enrichment for either condition. When restricted to ALS/FTD genes, ALS cases still demonstrated the enrichment only in the PMC (Fig. 4c; all, $P = 8.3 \times 10^{-3}$; exonic, $P = 0.060$; protein-altering, $P = 0.029$; linear mixed model), while FTD cases showed no significant

enrichment in the PFC, further supporting a broader genetic heterogeneity in FTD. The SC in ALS showed a modest increase in exonic and protein-altering variants, although this analysis is limited by the small number of control samples and wide CIs (Fig. 4c). Together, these diagnosis-specific and region-specific patterns suggest that functional somatic variants may contribute to the pathogenesis of sALS and sFTD.

## Predicted deleterious somatic variants have restricted regional distributions and are enriched in hypodiploid cells

Pathogenicity prediction identified six predicted deleterious somatic SNVs in known ALS and FTD genes (Supplementary Table 7), which account for 2.3% and 1.3% germline-free ALS and FTD cases, respectively (2.1% for overall). All variants in ALS cases were observed in PMC or SC, the most severely affected regions in ALS, emphasizing remarkable topographic specificity. All somatic variants occurred in disease genes with dominant inheritance when found in the germline setting, except for one sALS case with a predicted deleterious somatic *ALS2* (p.T787R) variant identified in the SC. *ALS2* is an autosomal recessive disease gene[47,48], and this individual also carried a predicted deleterious germline *ALS2* (p.Q24R) variant in addition to the identified somatic variant. Both *ALS2* variants were predicted to be deleterious, consistent with a 'second hit' mechanism at the cellular level in a small proportion of SC cells.

We selected four predicted deleterious somatic SNVs—*TIA1* (p.H54R), *MATR3* (p.K594I), *ALS2* (p.T787R) and *TARDBP* (p.L248F)—for detailed analysis of regional and cell-type distributions. Amplicon sequencing across multiple CNS regions showed that *MATR3* (p.K594I) and *TARDBP* (p.L248F) were restricted to the PMC (Fig. 4d and Supplementary Table 8), whereas *TIA1* (p.H54R) and *ALS2* (p.T787R) showed their highest VAFs in the SC (2.16% and 0.97%, respectively), where they were originally identified, and were detected at much lower levels in other brain regions (Fig. 4d and Supplementary Table 8). All four somatic SNVs were absent from the cerebellum. The ultralow VAFs and focal distribution of these variants suggest that they probably arose late in development and were thus likely CNS-restricted. Together with the enrichment of exonic and protein-altering somatic variants in disease-affected tissue regions, these findings also support the focal onset of ALS at the genetic level in these sporadic cases. Cells carrying damaging somatic variants could form initial lesions, likely TDP-43 inclusions, in UMNs and LMNs, with pathology ultimately spreading to other regions of the motor system that lack or carry exceedingly low levels of the variant, but nonetheless show robust postmortem pathology otherwise indistinguishable from germline cases. Consistent with this hypothesis, the case harboring the *TARDBP* (p.L248F) variant showed the highest phospho-TDP-43 (pTDP-43) level in the PMC, where the variant was detected (Extended Data Fig. 7). Quantification across seven brain regions revealed significant regional differences in pTDP-43 levels ($F = 3.20$, $P = 0.0099$, one-way ANOVA (analysis of variance)), with the motor cortex showing significantly higher pathology compared to the hippocampus ($P = 0.0042$), middle temporal gyrus ($P = 0.040$) and occipital cortex ($P = 0.032$) in post hoc Tukey's HSD (honest significant difference) tests. These results support the idea

**Fig. 4 | Somatic variants are enriched in ALS and FTD cases and disease-related tissue regions. a**, Enrichment of somatic variants in different genomic regions of germline-free ALS and FTD cases compared to normal controls. **b**, Enrichment of exonic and protein-altering somatic variants in two different groups of disease-related genes (ALS genes and FTD genes) compared to normal controls. **c**, Enrichment of somatic variants in ALS/FTD genes across different brain regions of germline-free ALS and FTD cases compared to normal controls. Missing points indicate no variants were observed. Disease effect sizes were estimated using an individual-level linear mixed-effects model by comparing variant burden per germline-free disease individual with that of normal individuals (Methods). Significance and 95% CI were estimated while controlling for potential confounding factors, including average read-depth, sex, postmortem

interval, sequencing batch and number of samples per donor. Sample sizes were as follows: control, $n = 144$; ALS, $n = 216$; FTD, $n = 78$ (biological replicates). Unadjusted $P$ values are shown, as the tests examine biologically distinct but nonindependent categories with differing background mutation structures. **d**, Regional distribution of VAFs of several predicted deleterious somatic variants in individual brains and SCs. Brain cortex is annotated by Brodmann areas. The color spectrum indicates the VAFs of somatic variants in amplicon sequencing. White indicates regions without the somatic variants. Red highlight indicates the region of initial detection by MIP sequencing. Schematics in **c** and **d** created in BioRender; **c**, Zhou, Z. https://biorender.com/s3p5mmn (2026); **d**, Zhou, Z. https://biorender.com/v6v72yx (2026).

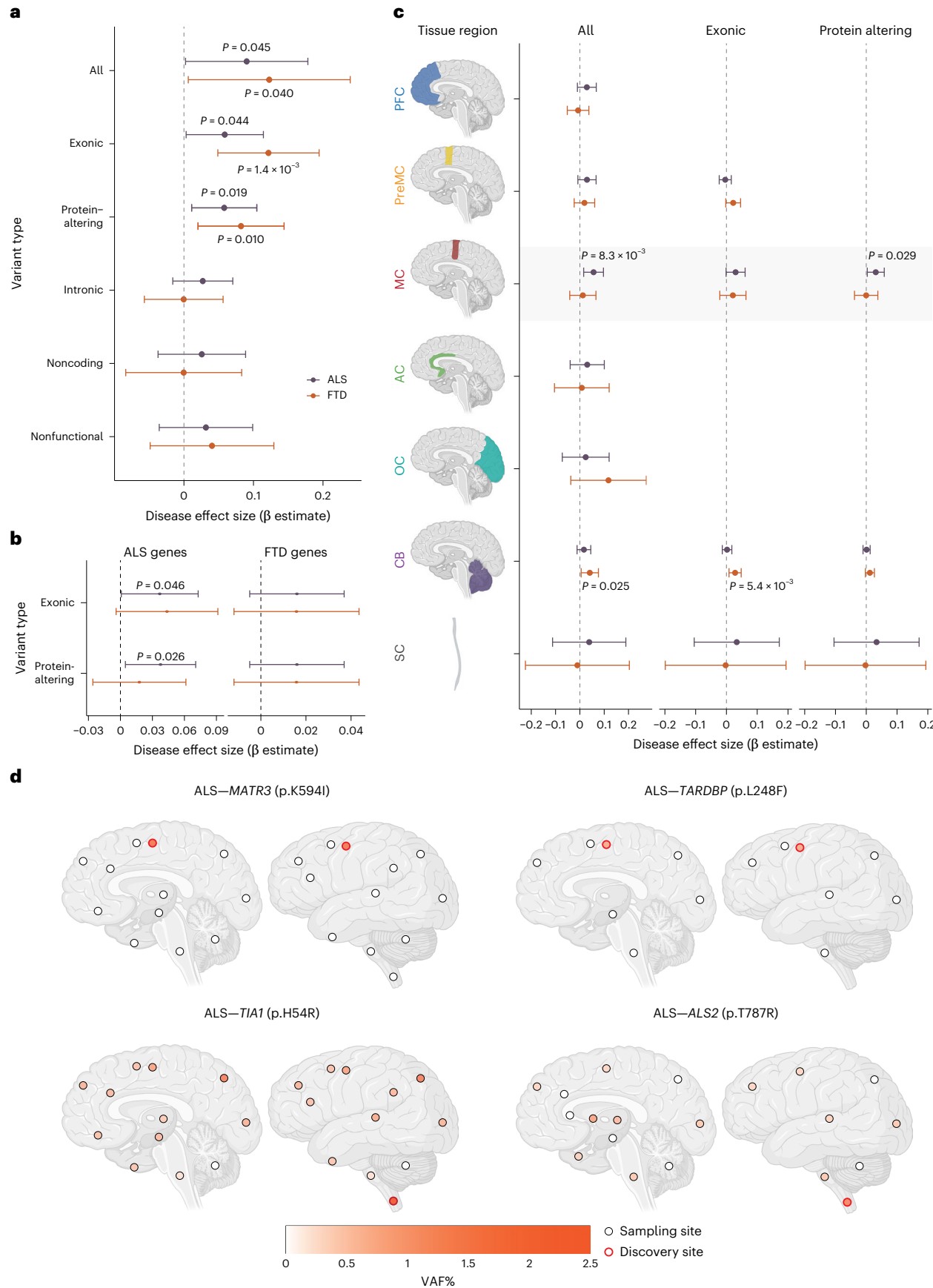

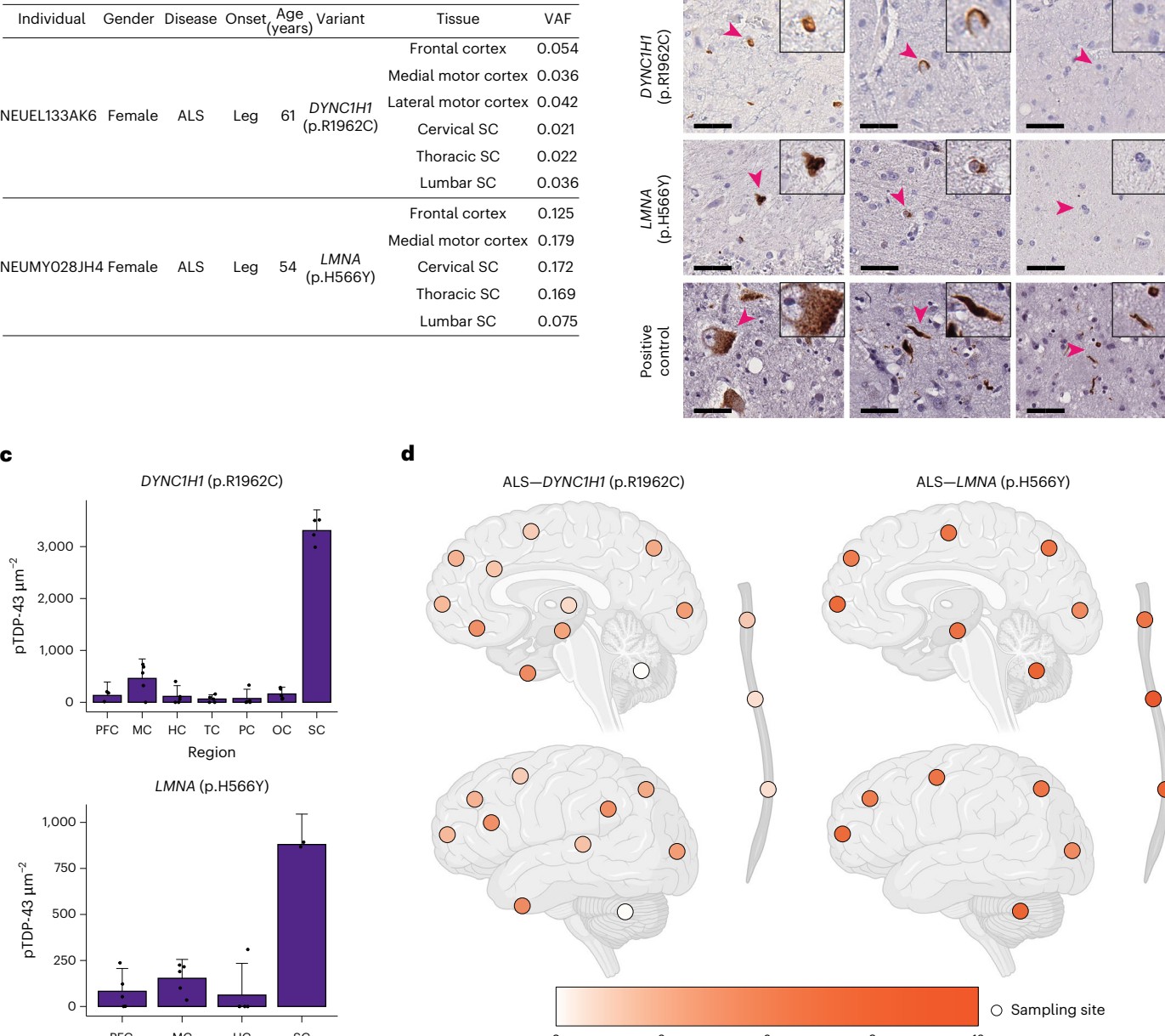

**Fig. 5 | Somatic variants in *DYNC1H1* and *LMNA* in sALS. a**, Two deleterious somatic SNVs that were shared by multiple tissue regions of the ALS cases. **b**, Sections of the lumbar SC, motor cortex and hippocampus of the two sALS cases stained with a pTDP-43 antibody. Scale bars = 40 μm. Arrowheads indicate the cells shown in the insets, which are magnified to twice their original size. **c**, Quantification of pTDP-43 staining of CNS tissue sections of the two sALS cases with *DYNC1H1* and *LMNA* somatic variants. For *DYNC1H1*, tissue-section-level biological replicates included three from PFC, four from SC and five from MC,

HC, TC, PC and OC. For *LMNA*, tissue-section-level biological replicates included two from SC and five from PFC, MC and HC. Bar graph = mean ± 95% CI. **d**, Regional distribution of VAFs of somatic variants in individual brains and SCs. Brain cortex is annotated by Brodmann areas. The color spectrum indicates the VAFs of somatic variants in amplicon sequencing. White indicates regions without the somatic variants. Schematic in **d** created in BioRender; Zhou, Z. https://biorender.com/v6v72yx (2026). HC, hippocampus; TC, temporal cortex; PC, parietal cortex.

that focal somatic variants may initiate pathology that later spreads to broader brain regions.

We then assessed the distribution of these four somatic SNVs across cell types by performing amplicon sequencing of DNA from neuronal (NeuN⁺), glial (NeuN⁻), diploid, polyploid and hypodiploid nuclei isolated by fluorescence-activated nuclei sorting from the regions in which the variants were originally identified (Extended Data Fig. 8a,b).

Interestingly, *TIA1* (p.H54R), *MATR3* (p.K594I) and *ALS2* (p.T787R) were enriched in hypodiploid nuclei (Extended Data Fig. 8c), which likely represent apoptotic cells (Supplementary Note)[49–52], suggesting a potential association with cell death. Surprisingly, these three variants were identified in all cell fractions, but were more enriched in non-neuronal than neuronal populations (Extended Data Fig. 8c,d). This finding could imply that neurons may exhibit a cell-type-specific

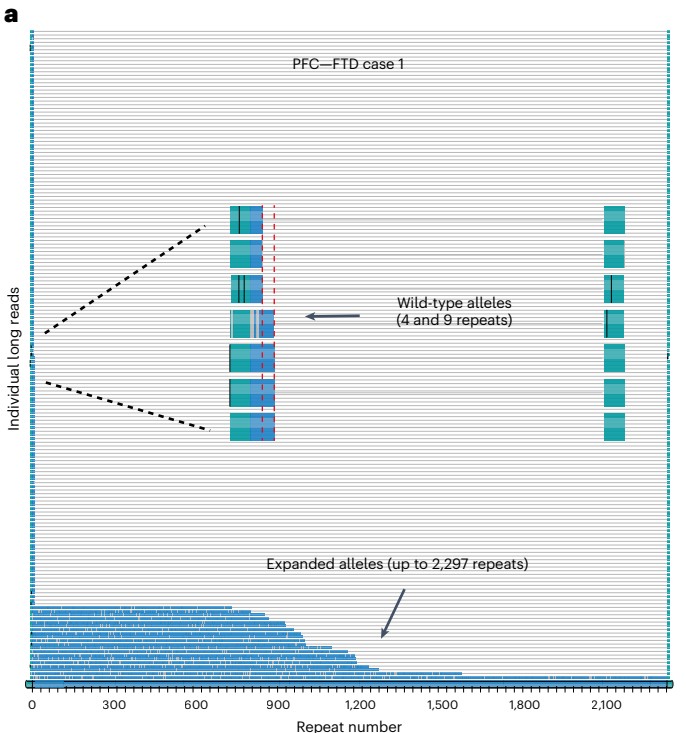

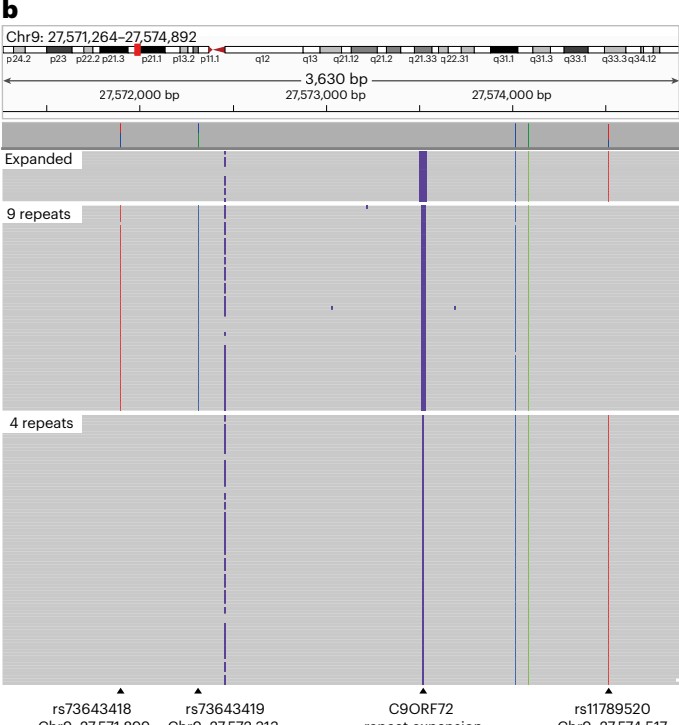

**Fig. 6 | Somatic *C9orf72* repeat expansion in an FTD case. a**, Waterfall plot showing targeted long-read sequencing results from the PFC tissue of an sFTD case. Two wild-type alleles with 4 and 9 repeats, respectively, are observed, along with somatic *C9orf72* repeat expansions ranging from 748 to 2,297 repeats. Each row represents an individual long read; the *C9orf72* repeat region is shown in blue, and the flanking regions are shown in green. Red dashed lines mark the sizes of the wild-type alleles. The *x* axis denotes the number of GGGGCC hexanucleotide repeats. **b**, IGV screenshot showing haplotype phasing based on SNPs flanking the *C9orf72* repeat expansion. Long reads are grouped by repeat size category (expanded, 9 repeats, 4 repeats). SNPs rs73643418, rs73643419 and rs11789520 are heterozygous in this individual. All expanded alleles share the same SNP haplotype as the 4-repeat allele.

vulnerability to somatic variants that damage ALS/FTD genes. However, further research is needed to confirm and better understand these potential associations and mechanisms. In contrast, the *TARDBP* (p.L248F) variant was validated only in the original PMC sample at very low VAF (~0.5%) by both amplicon sequencing and ddPCR, but was not detected in sorted cell fractions from an additional adjacent PMC sample, indicating a highly focal event.

## RNA-MosaicHunter identifies additional predicted deleterious somatic variants in bulk RNA-seq data of sALS cases

To complement our targeted sequencing of neurodegenerative genes, which identified predicted deleterious somatic variants in a small proportion of sALS and sFTD cases, we performed a transcriptome-wide screen for somatic variants using bulk RNA-seq data to explore whether previously unknown genes cause disease in a mosaic state. We profiled predicted deleterious somatic variants across all expressed genes in bulk RNA-seq data from 789 postmortem brain and SC tissue samples of 143 sALS cases and 23 age-matched controls generated by the New York Genome Center (NYGC) ALS Consortium (Supplementary Table 9; 81 sALS and 11 control cases overlapped with our MIP cohort) using RNA-MosaicHunter[53], which detects clonal somatic variants from bulk RNA-seq data using a Bayesian probabilistic model. Due to limited bulk RNA-seq depth, this approach is sensitive only to somatic variants with VAFs >~5% and does not detect somatic variants at ultralow levels.

Although we did not observe a significant increase in the burden of total or predicted deleterious somatic variants in germline-free sALS cases (Supplementary Fig. 3), we identified predicted deleterious somatic SNVs in *DYNC1H1* and *LMNA* in multiple CNS regions of two sALS cases lacking other pathogenic or predicted deleterious

germline/somatic variants (Fig. 5a and Supplementary Table 10); both cases overlapped with our MIP cohort. Heterozygously acting, generally de novo, variants in *DYNC1H1* and *LMNA* have been found in patients with phenotypes resembling spinal muscular atrophy[54–57], a motor neuron disease genetically distinct but sharing some pathological overlap with ALS[58]. Both individuals presented with leg-onset ALS and predominant SC TDP-43 pathology (Fig. 5a–c). Amplicon sequencing revealed broad CNS distribution of the *LMNA* (p.H566Y) variant (VAF = 5.3–12.3%) and the *DYNC1H1* (p.R1962C) variant (VAF = 0.1–5.2%), with the *DYNC1H1* variant extremely low in the cerebellum (0.1%), thoracic SC (0.8%) and lumbar SC (0.8%; Fig. 5d and Supplementary Table 8). Notably, the *DYNC1H1* (p.R1962C) variant was undetectable in cultured fibroblasts from the patient (Supplementary Table 8), indicating a late developmental, CNS-restricted origin. The broad CNS distribution of these variants aligns with our previous observations that somatic variants above ~5% VAFs are typically detected across the CNS[59], with lower regional VAFs potentially reflecting selective neuronal loss. The *DYNC1H1* (p.R1962C) variant is highly pathogenic, as it completely abolishes dynein motor function in vitro[60], and causes severe malformations of cortical development and delayed psychomotor development in patients carrying this germline variant[61,62]. Although the *LMNA* (p.H566Y) variant has not previously been reported, germline *LMNA* variants cause autosomal-dominant laminopathies with early lethality, including Hutchinson–Gilford progeria and congenital muscular dystrophy[63,64]. Thus, germline variants in both genes would ordinarily preclude ALS, but the mosaic state may allow for a normal early life followed by late-onset neurodegeneration. These findings suggest that further genome-wide exploration of brain tissue for somatic variants could reveal additional ALS genes that cause early lethality in the germline state.

## Somatic *C9orf72* repeat expansions detected by targeted long-read sequencing

The germline *C9orf72* repeat expansion is the most common genetic cause of ALS and FTD[27,28]. Repeat lengths >30 are generally considered pathogenic, with most affected individuals carrying hundreds to thousands of repeats. Intermediate expansions with 20–30 repeats confer an increased disease risk, while expansions with <20 repeats are regarded as normal[27,65]. While instability of pathogenic repeat expansions (>30 repeats) across tissues within the same individual has been reported[66–68], somatic expansion from nonexpanded alleles has not been described. In our ALS and FTD cohort, the RP-PCR analysis identified four ALS and FTD cases with both wild-type and expanded alleles showing reduced repeat peak heights[69] (Extended Data Fig. 9), suggestive of somatic expansion. Targeted long-read sequencing in brain tissues from these cases confirmed the presence of both wild-type and expanded alleles with dozens to thousands of repeats. Remarkably, one FTD case carried two short wild-type alleles (4 and 5 repeats), together with highly expanded pathogenic alleles (Fig. 6a). Haplotype phasing based on heterozygous single-nucleotide polymorphisms (SNPs) flanking the repeat expansions showed that all expanded alleles shared the same SNP haplotype as the 4-repeat allele (Fig. 6b), strongly suggesting spontaneous somatic *C9orf72* repeat expansion of a nonexpanded allele, although future analysis of more cases is needed to further characterize this possibility. In the remaining three cases, one wild-type allele and a spectrum of expanded alleles were detected, with minimum sizes of 20, 56 and 64 repeats (Extended Data Fig. 10), consistent with somatic expansion from intermediate or short pathogenic alleles, although the possibility of contraction cannot be excluded. Together, these findings suggest that somatic *C9orf72* repeat expansions may contribute to sALS and sFTD. Nevertheless, even long-read sequencing lacked sufficient sensitivity to comprehensively assess all germline-free cases ('Discussion'), and the prevalence of somatic *C9orf72* expansions remains to be determined.

## Discussion

Our data provide several important insights into sALS and sFTD. First, approximately 30% disease cases carried pathogenic or predicted deleterious germline variants in ALS/FTD genes, although our pathogenicity prediction might be overestimated. This advocates for a shift from family history-based to genetic testing-based disease classification, consistent with recently published guidelines for ALS genetic testing and counseling[70]. Second, a small but significant fraction (~2.1%) of germline-free sporadic cases harbored predicted deleterious somatic variants in known ALS or FTD genes, with disease-specific and region-specific distributions in ALS, providing proof of concept for a contribution of somatic variants to disease pathogenesis. In addition, we identified genes associated with severe pediatric degenerative diseases that may contribute to ALS in the somatic state, broadening the spectrum of potential disease genes. Finally, targeted long-read sequencing revealed somatic *C9orf72* repeat expansions arising from wild-type, intermediate and short pathogenic alleles in ALS and FTD cases.

While the case-control enrichment of somatic variants supports a pathogenic role, these variants occurred at surprisingly low VAFs and showed topographic restriction consistent with focal disease onset. Such variants likely arose late in development and were not shared by other tissue regions, exemplified by the *TARDBP* (p.L248F) variant detectable only at its site of discovery. The focal nature of these events would preclude detection by routine genetic testing with blood or other peripheral samples and support a mechanism by which degeneration may spread from a site containing mutant cells to anatomically connected regions, perhaps through intercellular transmission of TDP-43 proteinopathy[12–18,71,72]. The identification of predicted deleterious somatic variants in the PMC or SC from individuals with ALS suggests that ALS may initiate in UMNs or LMNs before involving both.

While performed on a limited number of variants, our cell-type analysis revealed that several predicted deleterious somatic variants were shared across cell types and enriched in glia compared to neurons. However, this apparent glial enrichment may reflect preferential loss of neurons carrying these variants, consistent with their enrichment in hypodiploid cells, a population likely associated with apoptotic cell death. Together, these findings support a model in which deleterious somatic variants contribute to focal disease initiation, with subsequent neuronal loss leading to a reduction in VAFs over time.

Although only ~2.1% germline-free disease cases carried predicted deleterious somatic variants in ALS/FTD genes in our MIP sequencing data, this is likely greatly underestimated due to limited sensitivity for detecting ultralow-VAF variants (Extended Data Fig. 1). Detecting such variants remains technically challenging[46], and broader sampling across CNS regions is also limited. However, future duplex sequencing approaches promise the ability to define the minimal VAFs that can initiate disease. On the other hand, several of our statistical comparisons approached nominal significance, highlighting the need for larger cohorts and more sensitive approaches.

While MIP sequencing did not allow for the detection of somatic *C9orf72* repeat expansions in our samples, targeted long-read sequencing identified ALS and FTD cases with somatic expansions of varying repeat sizes, including an FTD case carrying two wild-type and highly expanded alleles, consistent with de novo expansion. These findings suggest that somatic *C9orf72* repeat expansions may contribute to sALS and sFTD and could exhibit cell-type-specific patterns of expansion, analogous to somatic *HTT* CAG repeat expansion in striatal projection neurons[73]. However, the current targeted long-read sequencing approach preferentially enriches shorter alleles and provides limited flanking sequence, precluding accurate VAF estimation and robust haplotype phasing. Future targeted long-read sequencing with extended flanking coverage will be required to determine the origin and prevalence of somatic *C9orf72* repeat expansions.

Finally, the identification of somatic SNVs in *DYNC1H1* and *LMNA* suggests that genes predisposing to ALS and FTD in the somatic state may extend beyond those identified in germline studies. Although certain alleles in these genes cause motor neuron degeneration in the form of spinal muscular atrophy, other alleles—including *DYNC1H1* p.R1962C[61,62]—cause severe pediatric disease that would preclude late-life ALS in the germline but may be compatible with disease in the somatic state. Similarly, enrichment in germline-free FTD cases emerged only when all targeted neurodegenerative genes were considered, suggesting a broader array of FTD-related genes. Together, these findings highlight the potential of future genome-wide approaches to uncover additional somatic genetic mechanisms and illuminate the topographic spread of pathology from focal origins.

## Online content

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

[1]Division of Genetics and Genomics, Boston Children's Hospital, Boston, MA, USA. [2]Manton Center for Orphan Disease, Boston Children's Hospital, Boston, MA, USA. [3]Department of Pediatrics, Harvard Medical School, Boston, MA, USA. [4]Department of Biological Sciences, Sungkyunkwan University, Suwon, South Korea. [5]Department of Neurology, The Sean M. Healey and AMG Center for ALS at Mass General, Massachusetts General Hospital, Harvard Medical School, Boston, MA, USA. [6]Department of Pathology, Brigham and Women's Hospital, Harvard Medical School, Boston, MA, USA. [7]Howard Hughes Medical Institute, Boston Children's Hospital, Boston, MA, USA. [8]Department of Neurology, Yokohama City Minato Red Cross Hospital, Yokohama, Japan. [9]Department of Neurosciences, School of Medicine, University of California, San Diego, La Jolla, CA, USA. [10]Nuffield Department of Clinical Neurosciences, University of Oxford, Oxford, UK. [11]Department of Neurology, Lewis Katz School of Medicine at Temple University, Philadelphia, PA, USA. [12]These authors contributed equally: Zinan Zhou, Junho Kim, August Yue Huang. [13]These authors jointly supervised this work: Clotilde Lagier-Tourenne, Eunjung Alice Lee, Christopher A. Walsh. ✉e-mail: clagier-tourenne@mgh.harvard.edu; ealee@childrens.harvard.edu; christopher.walsh@childrens.harvard.edu

## Methods

### Tissue sources and sample preparation

Fresh-frozen postmortem human brain and SC tissues were collected by the Massachusetts Alzheimer's Disease Research Center, Oxford Brain Bank, Target ALS Foundation and NIH NeuroBioBank (Supplementary Table 1) according to their respective institutional protocols, written authorization and informed consent; the tissues were subsequently obtained for this study with the approval of the Boston Children's Hospital Institutional Review Board. Research on these deidentified specimens and data was performed at Boston Children's Hospital with approval from the Committee on Clinical Investigation. Furthermore, sALS and sFTD cases were selected based on the available clinical records. ALS and FTD cases without a clear recording of family histories were also selected if the age of death was above 45 years old.

gDNA of these tissue samples was extracted using the EZ1 Advanced XL (Qiagen) system, followed by an additional purification using AMPure XP Beads (Beckman Coulter).

### MIP panel design

A double-stranded DNA MIP panel targeting 1.4 Mb across exons and exon–intron junctions of 88 neurodegenerative genes was designed using custom scripts incorporating MIPgen[74] using the human reference genome, hg19, with Mly1 restriction sites masked with 'N' using BEDTools. The final panel of 26,439 MIPs captures an average fragment length of 209 bp, including the extension and ligation arms to ensure overlapping of the forward and reverse sequencing reads. The panel successfully targets 92.7% bases, including flanking intronic regions, with >98% exonic bases covered with an average of at least two unique MIPs. All MIPs were designed to include a custom backbone consisting of primer binding sites and dual 5-nt unique molecular identifier (UMI). MIPs were rebalanced in the pool based on the percent of GC content within the regions to improve coverage at GC-rich regions. We increased the number of copies of MIPs that bind to GC-rich regions with the following criteria: 60–70% GC = 2 copies, 70–80% GC = 5 copies, 80–90% GC = 8 copies and >90% GC = 10 copies. Common primer binding and Mly1 restriction enzyme sites were added to both ends of the oligo sequences to enable blunt-end removal of the primer binding sites. The forward and reverse complement sequences were printed into a single ssDNA pool by CustomArray (Bothell). The resulting panel was amplified at a low cycle number (12×), digested with Mly1 enzyme for 12 h at 37 °C, and purified using Qiagen Nucleotide Removal Kit.

### MIP capture and library construction

Two hundred and fifty nanograms of gDNA were first hybridized in a 15 µl reaction with 1.5 µl of Ampligase 10× reaction buffer (VWR), 1.5 µl of the reverse blocking oligo (5′-NNNNGAAGTCGAAGGGCTATAGGCTGCCATCACANNNN-3′) and the MIP pool at 63 nM for 10 min at 95 °C and 24 h at 60 °C. Gap-fill/ligation was then performed by adding 1 unit of Phusion high-fidelity DNA polymerase (Thermo Fisher Scientific), 4 units of Ampligase DNA ligase (Epicentre), 0.2 µl of Ampligase 10× reaction buffer, 0.6 µl of dNTPs (10 mM) and 1 µl of nuclease-free water to the MIP capture product and incubated at 60 °C for 1 h. For exonuclease digestion, 50 units of Exonuclease III (Thermo Fisher Scientific), 10 units of Exonuclease I (Thermo Fisher Scientific), 0.2 µl of Ampligase 10× reaction buffer (VWR) and 2.05 µl of nuclease-free water were added to the gap-fill/ligation product, which was incubated for 40 min at 37 °C and 5 min at 95 °C. Ten microliter of the captured library is amplified in a 50 µl final reaction by adding 1 unit of Phusion Hot Start II DNA polymerase (Thermo Fisher Scientific), 10 µl of 5× HF buffer, 1 µl of dNTPs (10 mM), 1 µl of the universal MIP barcode forward primer (10 µM), 1 µl of the individual barcode reserve primer (10 µM) and 26.5 µl of nuclease-free water. MIP library amplification was then performed under the following conditions: 98 °C for 30 s; 16 cycles of 98 °C for 10 s, 60 °C for 30 s and 72 °C for 30 s; 72 °C for 2 min. MIP library was then purified using 2× AMPure XP Beads (Beckman Coulter)

and quantified by the Quant-iT dsDNA Assay HS Kit (Thermo Fisher Scientific). Ninety-six MIP libraries were pooled together and sequenced on one lane of Illumina HiSeq X.

### Preprocessing and read mapping of MIP sequencing data

MIP sequencing primers were removed first from the raw FASTQ files using Cutadapt[75] (v2.4; 5′ adaptor of the first read−CATACGAGATCCGTAATCGGGAAGCTGAAG; 3′ adaptor of the first read−ACACTACCGTCGGATCGTGCGTGT; 5′ adaptor of the second read−GCTAAGGGCCTAACTGGCCGCTTCACTG; 3′ adaptor of the second read−CTTCAGCTTCCCGATTACGGATCTCGTATG). Trimmed reads were aligned to the human reference genome (GRCh37) using BWA-MEM[76] (v0.7.15) and sorting and indexing were performed using SAMtools[77] (v1.3.1). From the aligned BAM file, off-target reads were removed by checking the overlaps with the target regions using BEDTools[78] (v2.26.0). MIP arm regions were masked by soft-clipping for each read using BAMClipper[79] (v1.0.1). UMI information was extracted, and then mapped reads were deduplicated based on the mapping coordinates and the shared UMI using UMI-tools[80] (v1.0.0). Base quality score recalibration and local realignment were performed using the GATK (v3.7)[81], generating final analysis-ready BAMs.

### Variant calling for germline variants

Initial candidates of germline SNVs and indels were identified using GATK HaplotypeCaller with default parameter settings. Low-quality candidates were filtered out if any of the following conditions is not satisfied: (1) ≥10 variant-supporting reads, (2) ≥20 total read-depth at the variant site, (3) VAF ≥ 0.3, (4) GATK QUAL ≥ 50 and (5) identified in all brain regions from the same individual except for the samples failed to cover the variant site (<10 reads). Possible pathogenic germline variants were further selected by satisfying all the following conditions: (1) present in less than 0.1% of the population in any ancestry group of public databases including dbSNP[82], the 1000 Genomes Project[83], the Exome Aggregation Consortium[84], the Genome Aggregation Database[85], the NHLBI Exome Sequencing Project (ESP6500)[86], the Greater Middle East variome project[87] and Kaviar database[88]; (2) candidates observed only in disease or control groups but not in both; (3) possible protein-altering candidates (missense, nonsense, frameshift or splicing variants); and (4) affecting 30 ALS-related and FTD-related genes. Deleteriousness prediction module ('Computational prediction of variant deleteriousness') was then applied to the remaining candidates, and predicted deleterious variants were reported as final deleterious germline variants. ANNOVAR[26] was used to annotate the genomic region, affected genes, population allele frequency and exonic variant functions. SpliceAI[89] was additionally used to identify more splice-altering variants. Candidates with δ score >0.5 were considered to be possible splicing variants.

### RP-PCR assay for *C9orf72* repeat expansion genotyping

RP-PCR of the *C9orf72* repeat expansion was performed in a 30-µl PCR reaction with 150 ng of gDNA, 15 µl of 2× FastStart PCR Master (Roche), 2 µl of DMSO, 5 µl of 5× Q-solution (Qiagen), 1 µl of 5 mM 7-deaza-dGTP (NEB), 1 µl of 25 mM MgCl$_2$ (Qiagen) and 1 µl of the primer mix (40 µM of the forward primer−5′-/56-FAM/AGTCGCTAGAGGCGAAAGC-3′; 20 µM of the reverse primer−5′-TACGCATCCCAGTTTGAGACGGGGGCCGGGGCCGGGGCCGGGG-3′; 40 µM of the anchor/tail primer−5′-TACGCATCCCAGTTTGAGACG-3′). The reaction was performed with touchdown PCR cycling conditions consisting of 15 min at 95 °C, followed by cycles of 94 °C for 1 min, annealing starting at 70 °C for 1 min and extension at 72 °C for 3 min, ending with a final extension step of 10 min at 72 °C. The annealing temperature was decreased in 2 °C steps as follows: 70 °C for two cycles, 68 °C for three cycles, 66 °C for four cycles, 64 °C for five cycles, 62 °C for six cycles, 60 °C for seven cycles, 58 °C for eight cycles and 56 °C for five cycles. The RP-PCR products were separated by the SeqStudio Genetic

Analyzer (Thermo Fisher Scientific) with the GeneScan 600 LIZ Dye Size Standard (Thermo Fisher Scientific). Results of fragment sizes were analyzed by Peak Scanner Software (v1.0; Thermo Fisher Scientific).

## Somatic variant calling from MIP sequencing data

Three different callers, RePlow (v1.1.0)[38], Mutect2 (v4.1.5)[39] and Pisces (v5.2.11)[40], were used to generate initial candidate sets. RePlow is optimized to detect low-VAF somatic variants in deep targeted sequencing data. It generates a profile of background errors per substitution type and uses these distributions as priors in a Bayesian model to estimate the probability of a variant candidate being an error. Mutect2 is one of the most widely used somatic variant callers, particularly sensitive to detecting low-VAF variants in impure and heterogeneous samples. It uses a Bayesian classifier to evaluate the likelihood of a variant being genuine versus a sequencing error and applies multiple filters to reduce false positives. Pisces is specifically designed for detecting somatic mutations from amplicon sequencing data, particularly in cases where no matched control sample is available. It stitches paired-end reads into consensus reads and recalibrates variant quality scores specifically tuned to address amplification-related errors such as thermal damage or deamination, often observed in formalin-fixed, paraffin-embedded samples.

Each sample was analyzed by all three callers using single-sample mode. Default parameter settings were used, except for the adjustments to disable the coverage limit. Variants that passed all the filters from each caller were used to make three different initial sets. Candidates identified by only one caller were discarded, and those called by at least two callers were retained as a double-call set. For indels, double-calls between Mutect2 and Pisces were used as somatic indel candidates, as RePlow does not support indel detection. For SNVs, among double-calls, Mutect2–Pisces pairs were additionally filtered out due to high FPRs and low validation rates in the benchmarking dataset (Extended Data Fig. 1d). Remaining RePlow-based SNV double-calls and indel candidates were subject to multistep variant filters to further remove false positive candidates.

Unlike germline variant calling, somatic variant calling aims to reliably detect low-VAF variants up to ~0.5%, which requires enough supporting evidence to control the FPR. Calling thresholds such as variant-supporting read count, read-depth at the variant site and average base-call quality were determined by the benchmarking data. Somatic variants were selected satisfying all the following conditions: (1) ≥50 total read-depth at the variant site, (2) ≥15 variant-supporting reads excluding the reads with the variant allele on their probe-arm regions, (3) >30 average base-call quality of variant allele, (4) ≥2 different types of variant-supporting amplicons, (5) $0.001 \leq VAF \leq 0.4$, (6) ≤3 variant candidates within 20-bp window from the same sample, (7) present in less than 0.1% population in any ancestry group of public databases and (8) observed in no more than two different individuals.

Somatic variants were further annotated using similar criteria to select predicted deleterious germline variants. Among the final candidates, variants that are (1) observed only in disease or control groups but not in both, (2) possible protein-altering variants, and (3) affecting ALS-related and FTD-related genes were selected and applied for the deleteriousness prediction module. ANNOVAR and SpliceAI were used to annotate variants with various genomic information and detect additional splice-altering variants, respectively.

## Computational prediction of variant deleteriousness

Deleteriousness prediction module was applied to filtered germline and somatic variants to refine the predicted deleterious (potentially pathogenic) candidate sets. Variants that were previously reported as benign/likely benign in the clinical databases (ClinVar[90] and Human Gene Mutation Database[91]) were excluded from the predicted candidate set. Nonsense, frameshift and canonical splicing variants (±1–2 splice sites) were assumed to disrupt gene function and were included in the predicted deleterious set. For missense variants, the dbNSFP database[92] was used to adopt multiple computational algorithms (SIFT[93], PolyPhen2 (ref. 94), LRT[95], MutationTaster[96], MutationAssessor[97], FATHMM[98], FATHMM-MKL[99], PROVEAN[100], MetaSVM[101], MetaLR[101]), considering damaging effects at different levels such as biochemical property, protein structure and evolutionary conservation. Categorical prediction results of each algorithm were delivered by ANNOVAR. A missense variant was selected to be predicted deleterious if at least three different algorithms predicted damaging effects (deleterious for SIFT, LRT, FATHMM, PROVEAN, MetaSVM and MetaLR; probably damaging for PolyPhen2; disease-causing for MutationTaster). Possibly/likely damaging predictions were excluded for more conservative selection. For ALS/FTD-related genes, previously reported inheritance patterns (dominant/recessive) were carefully checked. For recessive genes, two independent variants in the same gene were required to determine whether a given individual was affected by predicted deleterious variants.

## Benchmarking with spike-in datasets

Extracted gDNA from two Coriell cell lines (GM12878 and GM24695) was used to generate spike-in data. Extracted DNA was mixed at five different levels to mimic low-level somatic variants, targeting the VAFs of 0.5%, 1%, 2.5%, 5% and 10%. Genomic DNA from GM12878 cells was spiked into DNA from GM24695; unique germline SNPs in GM12878 served as somatic variants. Genomic position and genotype information for germline SNPs of Coriell samples were obtained from NIST high-confidence call sets[102]. A total of 165 SNPs (57 homozygous and 108 heterozygous SNPs) were initially covered by our designed MIP panel. Among them, we only used heterozygous SNPs for benchmarking to accurately reflect target VAFs. In addition, we excluded the spike-in variants from benchmarking if (1) a given site could be covered by only one amplicon, or (2) have read-depth of less than 100, as these are low-quality variants unable to be detected due to inherent issues with panel design rather than the variant calling process. RePlow, Mutect2, Pisces and their combinations were tested. Detected variants not in the benchmark set were considered to be false positives, except for GM24695 germline SNPs and excluded spike-in variants.

## Somatic variant calling from bulk RNA-seq data

Raw BAM files of bulk RNA-seq and matched whole-genome sequencing data for sALS and control cases of the NYGC ALS Consortium were obtained from the New York Genome Center. RNA-seq reads extracted from raw BAM files were aligned to the GRCh38 human reference genome by STAR (v2.5.0a)[103] in the two-pass mode with the reference gene annotation (Gencode, v39). The aligned BAM files were processed by Picard (v1.138) to remove duplicates, and then by GATK (v3.6)[104] for SplitNCigarReads, indel realignment and base quality recalibration. We further excluded reads that were improperly paired or had ambiguous alignment.

Somatic SNVs were called by RNA-MosaicHunter (v1.0) with default parameters. Derived from MosaicHunter[105], which was designed for somatic variant calling in DNA sequencing, RNA-MosaicHunter incorporates a Bayesian genotyper and a series of empirical filters to systematically distinguish somatic variants from technical artifacts and germline variants, with 59% sensitivity and 94% precision benchmarked using cancer datasets. Specifically, germline variants identified from the matched whole-genome sequencing data from the same individual were excluded. We excluded A-to-G candidates because they are most likely led by the widespread A-to-I(G) RNA editing events in the human genome. To remove recurrent artifacts, we considered only exonic candidates that were called in one or two individuals. We further excluded candidates present in human polymorphism databases, including dbSNP[82], the 1000 Genomes Project[106], the Exome Sequencing Project[107] and the Exome Aggregation Consortium[108].

## Long-read targeted sequencing

The PureTarget kit (PacBio) was used to prepare sequencing libraries with 2 μg of high-molecular-weight DNA following the manufacturer's protocol. Pooled libraries were sequenced on one Revio SMRT Cell. Raw HiFi reads were aligned to the human reference genome (GRCh38) using minimap2 (v2.28)[109], and sorting and indexing were performed using SAMtools (v1.3.1)[77]. The Tandem Repeat Genotyping Tool (v1.1.1)[110] was used to determine the *C9orf72* repeat counts for each sample (chr9: 27573528-27573546, (GGGGCC)n). A waterfall plot was generated using TRVZ to visualize repeat counts with mosaicism.

## Nuclei isolation and whole-genome amplification

Isolation of total (DAPI⁺), neuronal (NeuN⁺), non-neuronal (NeuN⁻) and damaged (low DAPI) nuclei was achieved by fluorescence-activated nuclei sorting together with nuclear staining of NeuN (Millipore, MAB377; clone A60, 1:1,500) and DAPI following a previously published study[111]. Five hundred nuclei of each cell population were sorted into wells of 96-well plates.

Sorted nuclei were subjected to genome amplification using the Primary Template-directed Amplification kit (BioSkryb, 100136) following the manufacturer's protocol.

## Amplicon sequencing

Primer sets targeting each identified somatic SNV were designed using BatchPrimer3 (Supplementary Table 11). Amplicon was amplified for 25 cycles in a 50 μl PCR reaction with 50 ng of gDNA, 1 unit of Phusion Hot Start II DNA polymerase (Thermo Fisher Scientific), 10 μl of 5× HF buffer, 1 μl of dNTPs (10 mM) and 10 μl of each primer (10 μM). Amplicon PCR products were purified using a double-sided size selection (0.65× + 1.05×) with AMPure XP beads (Beckman Coulter, A63882). Purified amplicons were then pooled based on the concentrations measured by the Quant-iT dsDNA Assay HS Kit (Thermo Fisher Scientific) and sequenced using Amplicon-EZ (Genewiz).

## Immunohistochemistry

Immunohistochemistry was performed using DAB (3,3′-diaminobenzidine) detection, following published protocols[112]. Briefly, 7-μm formalin-fixed, paraffin-embedded sections were dewaxed with CitriSolve, then rehydrated through decreasing ethanol concentrations. Antigen retrieval was performed using sodium citrate buffer pH 6.0 at 121 °C for 15 min. Endogenous peroxidases were blocked using 3% hydrogen peroxide solution, and nonspecific binding was blocked using 10% normal goat serum. Sections were then incubated overnight at 4 °C with primary antibody (pTDP-43 mouse polyclonal; Cosmo Bio, CAC-TIP-PTD-P03; 1:10,000). After washing with TBS-Triton, sections were incubated with a Horseradish peroxidase-conjugated goat antimouse secondary (Dako) for 1 h at room temperature. Horseradish peroxidase signal was detected using DAB substrate (Dako) applied for 15 min. Counterstaining was performed using Coles hematoxylin for 1 min. Sections were then dehydrated, cleared using CitriSolve and mounted using glass coverslips. All sections were viewed using a Leica upright light microscope and assessed for section quality before whole-slide digital scanning.

## Quantification of pTDP-43 burden by immunohistochemistry

Stained sections were scanned using a NanoZoomer whole-slide digital imager at ×40 magnification. Images were then visualized and quantified using QuPath image analysis software and algorithms following the published protocols[112]. Briefly, for cortical/cerebellar sections, five regions of interest (ROIs) measuring 3 mm² (1,000 × 3,000 μm) were placed equidistantly around a single gyrus with the short end of the ROI placed at the pial surface. Pathology was then quantified using a positive pixel count within each ROI and measurements were averaged to provide an output of positive pixels per mm². For SC sections, a square ROI (2.25 mm²) was placed on each side of the central canal within the anterior horn and measurements were averaged.

## Reporting summary

Further information on research design is available in the Nature Portfolio Reporting Summary linked to this article.

## Data availability

The bulk RNA-seq data generated by the NYGC ALS Consortium are available through controlled access through the Target ALS Data Portal (https://dataengine.targetals.org/). Access requires acceptance of the Target ALS Data Use Agreement and submission of a data access request through the portal application process. Additional details are provided in the Target ALS Data Portal User Manual. The MIP-based targeted sequencing data and long-read sequencing data generated in this study have been deposited in dbGaP under accession phs003530, with access governed by human participant privacy regulations. Germline and somatic variants identified and validated in this study are listed in the Supplementary Tables.

## Code availability

The source code and default configuration file of RNA-MosaicHunter have been published and are available at https://github.com/AugustHuang/RNA-MosaicHunter. Other custom analysis scripts used in this study are publicly available on GitHub (https://github.com/kimjh607/ALS-FTD-somatic-mosaicism) and archived on Zenodo (https://doi.org/10.5281/zenodo.18682277)[113].

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

## Acknowledgements

The authors thank the Massachusetts Alzheimer's Disease Research Center, Oxford Brain Bank, Target ALS Foundation (Biobank Core Facility at St. Joseph's Hospital and Barrow Neurological Institute, Georgetown Brain Bank, Eleanor and Lou Gehrig ALS Center at Columbia University and UCSD ALS bank) and NIH NeuroBioBank (Harvard Brain Tissue Resource Center, Mount Sinai/JJ Peters VA Medical Center NIH Brain and Tissue Repository, Brain Endowment Bank of University of Miami, University of Pittsburgh Neuropathology Brain Bank, University of Maryland Brian and Tissue Bank and UCLA Human Brain and Spinal Fluid Resource Center) for providing fresh-frozen human tissues. The authors thank the Target ALS Human Postmortem Tissue Core, New York Genome Center for Genomics of Neurodegenerative Disease, Amyotrophic Lateral Sclerosis Association and TOW Foundation for providing the bulk RNA-seq data. The authors thank the donors and families for their contributions, and D. Gonzalez for assistance with tissue procurement. The authors thank the Research Computing group at Harvard Medical School and Boston Children's Hospital. The brains shown in Figs. 4 and 5 were illustrated by A. Lai (Walsh Lab) with input from the authors. This work was supported by the PRMRP Discovery Award (W81XWH2010028 to Z.Z.), the Edward R. and Anne G. Lefler Center postdoctoral fellowship (to Z.Z.), the American Heart Association Career Development Award (23CDA1046074 to Z.Z.), the National Research Foundation of Korea (2022R1C1C1010430, RS-2023-00217881 and RS-2025-02215360 to Junho Kim and R01 AG088082 to A.Y.H.), the Alzheimer's Association Research Fellowship (to A.Y.H. and grant R56 AG079857 to A.Y.H., C.A.W. and E.A.L.), a Cullen Education and Research Foundation Young Investigator award from the Healey Center (to M.N.), a Holloway Postdoctoral Fellowship from the Association for Frontotemporal Degeneration (to M.N. and K08 AG065502 to M.B.M.), donors of the Alzheimer's Disease Research program of the BrightFocus Foundation (A20201292F to M.B.M.), the Doris Duke Charitable Foundation Clinical Scientist Development award (2021183 to M.B.M. and K01 AG051791 to E.A.L.), the Suh Kyungbae Foundation (to E.A.L., DP2 AG072437 to E.A.L., R01 NS032457 to C.A.W. and R01 AG070921 to C.A.W. and E.A.L.), a Massachusetts Alzheimer's Disease Research Center pilot grant (to C.L.-T. and C.A.W.) and the Allen Discovery Center program, a Paul G. Allen Frontiers Group advised program of the Paul G. Allen Family Foundation (to C.A.W. and E.A.L.). C.L.-T. is supported by the Araminta Broch-Healey Endowed Chair in ALS. C.A.W. is an Investigator of the Howard Hughes Medical Institute. The funders had no role in the study design, data collection and analysis, decision to publish or preparation of the manuscript.

## Author contributions

Z.Z., Junho Kim, A.Y.H., E.A.L., C.L.-T and C.A.W. conceived and designed the study. Z.Z. performed tissue processing, MIP panel sequencing, cell sorting, amplicon sequencing and targeted long-read sequencing. Junho Kim performed bioinformatic analysis for MIP and validation data with assistance from R.D., T.S. and B.Z. A.Y.H. performed bioinformatic analysis for bulk RNA-seq data with assistance from J.P. and M.B. Jinhyeong Kim performed bioinformatic analysis for long-read sequencing data. M.N. optimized and performed immunofluorescence imaging and quantification, and generated

data shown for this study. Z.Z., M.B.M. and R.D. designed the MIP panel. B.C., K.M., R.C.Y. and C.-Z.L. helped with tissue processing and amplicon sequencing. C.K. provided technical support for MIP sequencing. J.E.N. contributed tissue procurement and ethics expertise. T.O. and J.R. provided immunofluorescence images and interpretation of disease pathology. L.W.O. and O.A. provided fresh-frozen human tissues and interpreted disease pathology. C.L.-T., E.A.L. and C.A.W. supervised the study. Z.Z., Junho Kim, A.Y.H., C.L.-T., E.A.L. and C.A.W. wrote the manuscript.

## Competing interests

C.L.-T. serves on the scientific advisory board and/or has received consulting fees from SOLA Biosciences, Libra Therapeutics, Arbor Biotechnologies, Dewpoint Therapeutics, Mitsubishi Tanabe Pharma Holdings America, Sanofi, AUTTX, Carthera, the Milken Institute and Applied Genetic Technologies. E.A.L. serves on the scientific advisory board of Genome Insight. C.A.W. is a paid consultant (cash, no equity) to CAMP4 Therapeutics (cash, no equity) and Maze Therapeutics (cash and equity), is a member of the scientific advisory board for Bioskryb Genomics (cash) and is a founding advisor for Mosaica Medicines (equity). These companies did not fund this research project and had no role in its conception or execution. The other authors declare no competing interests.

## Additional information

**Extended data** is available for this paper at https://doi.org/10.1038/s41588-026-02570-6.

**Correspondence and requests for materials** should be addressed to Clotilde Lagier-Tourenne, Eunjung Alice Lee or Christopher A. Walsh.

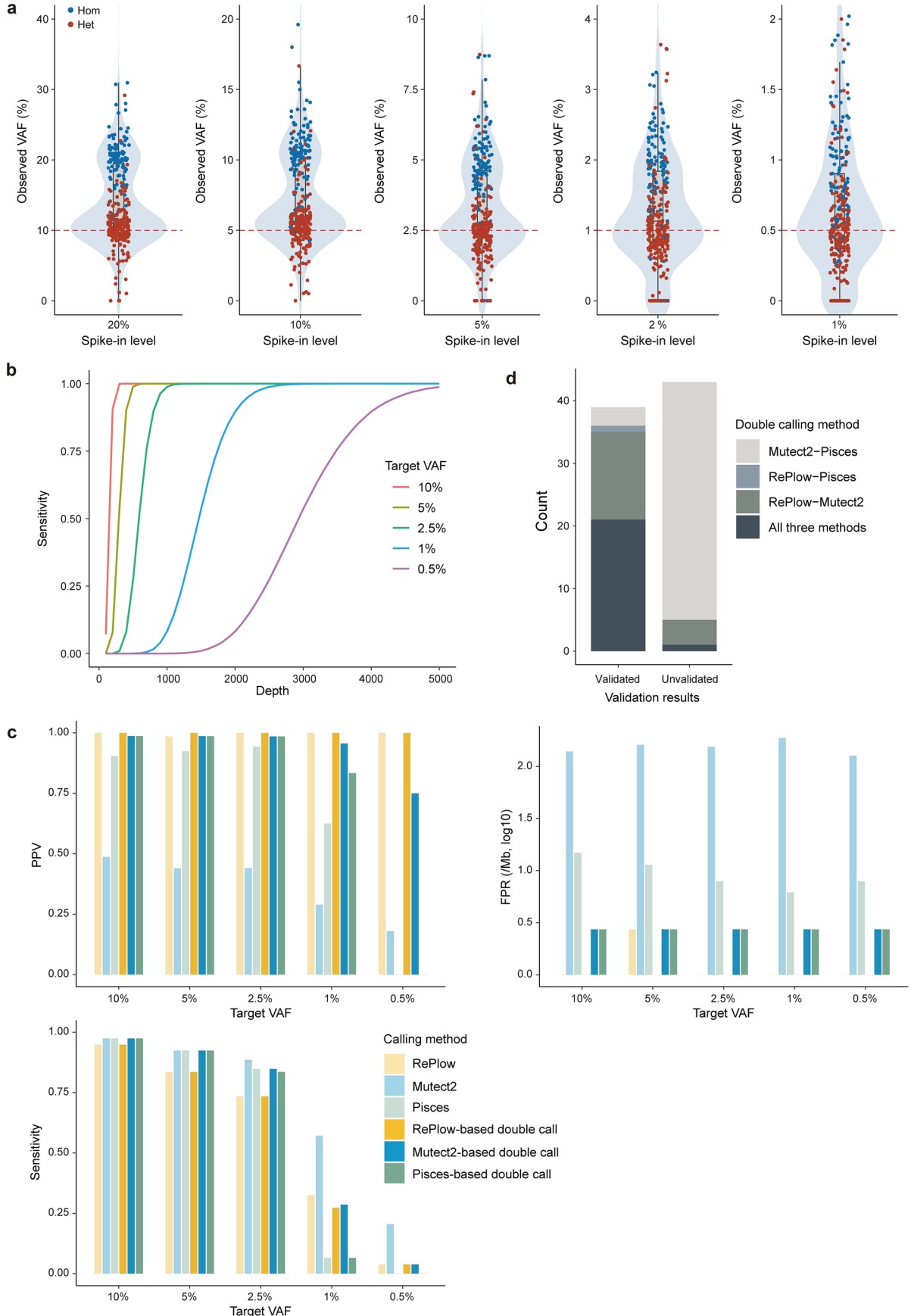

**Extended Data Fig. 1 | See next page for caption.**

**Extended Data Fig. 1 | Evaluation of the variant calling method for somatic variant using spike-in and validation experiments. a**, VAF distribution of 165 artificial variants from the MIP-sequenced spike-in data. Each violin plot denotes benchmark variants across spike-in mixtures at five different levels (20%, 10%, 5%, 2%, and 1%). Box plots indicate the median (centerline) and interquartile range (25th–75th percentiles), with whiskers extending to the most extreme values within 1.5× the interquartile range. Each dot represents the VAF of an individual variant. Red dotted lines represent the target VAF of a given mixture. **b**, Theoretical sensitivity of variant detection across varying VAF levels and sequencing depths under error-free conditions. **c**, Benchmarking of three different callers and their combinations using spike-in data. Please note that only heterozygous variants were used for benchmarking to specifically evaluate performance for the target VAFs. PPV, positive predictive value; FPR, false positive rate. **d**, Validation results of identified double-called variants using deep amplicon sequencing. Each color represents a different double-calling combination.

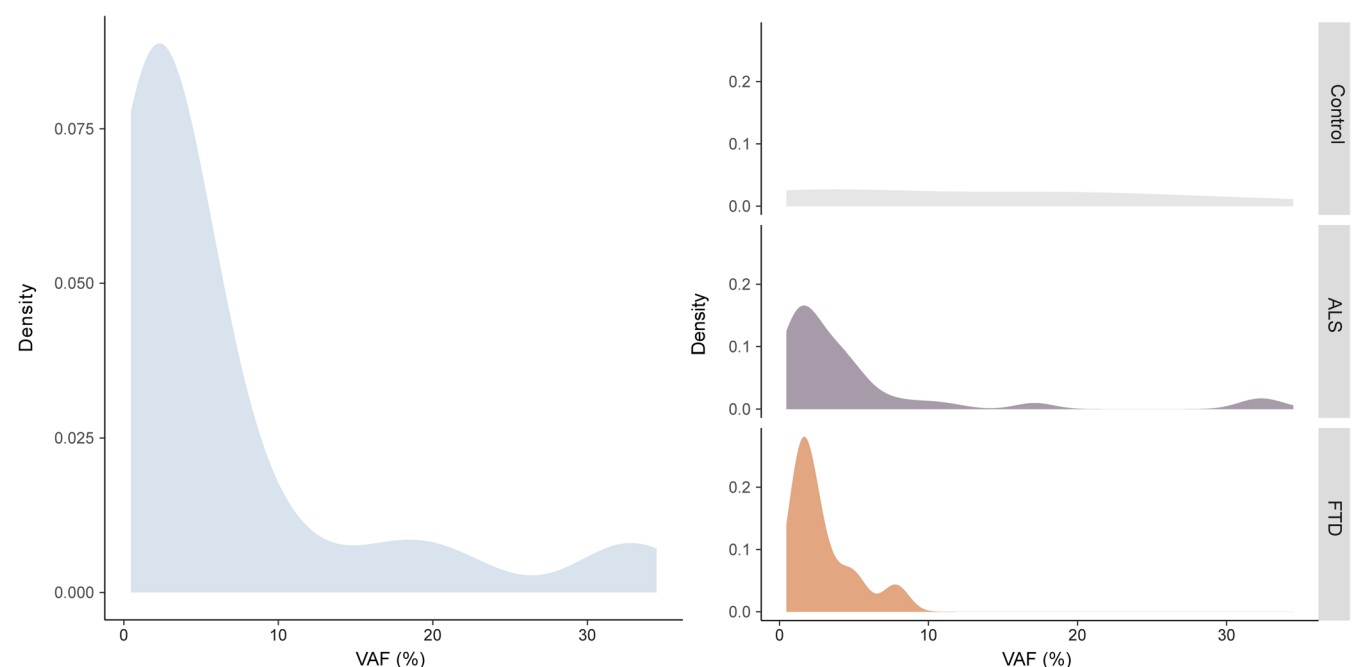

**Extended Data Fig. 2 | Variant allele fraction (VAF) distribution of identified somatic variants.** VAF distributions of total somatic variants (left) and across different clinical conditions (right) are shown in density plots. All VAFs were used to draw the figure if a given variant was observed multiple times in different brain regions.

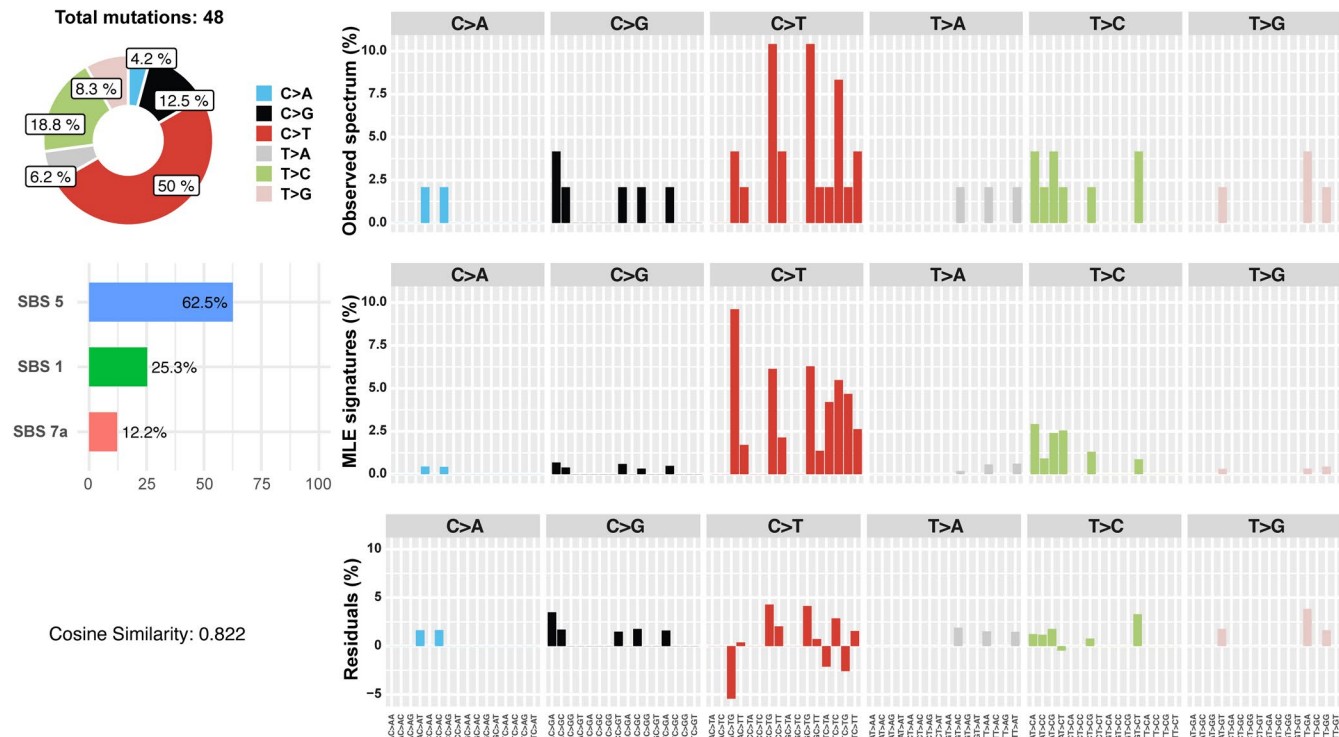

**Extended Data Fig. 3 | Somatic SNV profile and underlying mutational processes.** Mutational signature analysis using Mutalisk revealed two clock-like signatures (SBS5 and SBS1) as major underlying mutational mechanisms. Please note that Mutalisk accounts for SNVs only, resulting in a total of 48 unique variants obtained from germline-free individuals. SBS1 indicates mutations caused by deamination of methylated cytosine in cycling cells and is directly associated with stem cell division and mitosis.

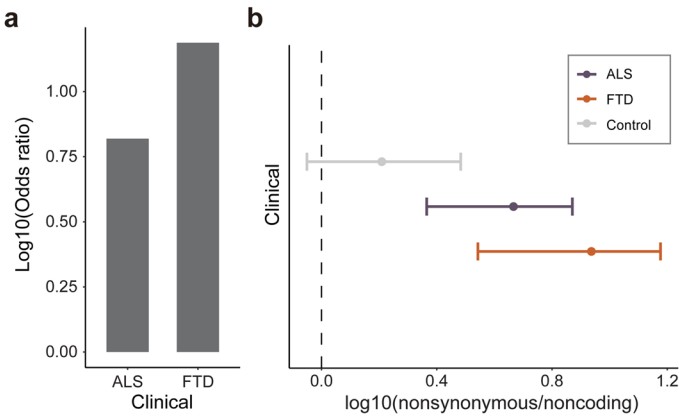

**Extended Data Fig. 4 | Ratios of nonsynonymous to noncoding variants in germline-free ALS and FTD cases. a**, Group-level $\log_{10}$(odd ratios) of nonsynonymous vs. noncoding variants for ALS and FTD cases compared to controls. **b**, Individual-level distributions of $\log_{10}$(nonsynonymous/noncoding) ratios for ALS, FTD, and controls. Control, $n = 144$; ALS, $n = 216$; FTD, $n = 78$; biological replicates. Data are shown as mean ± 95% CI.

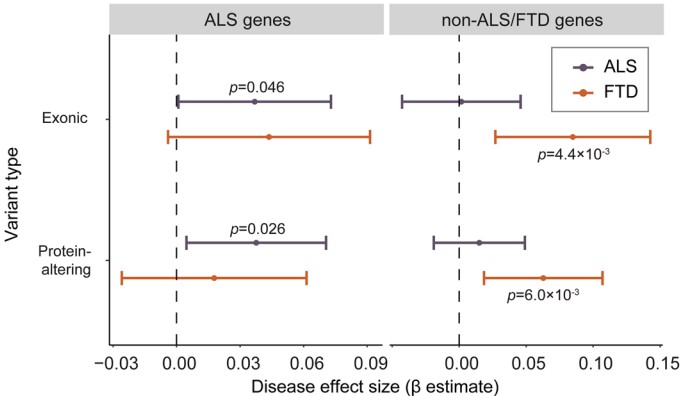

**Extended Data Fig. 5 | Enrichment of exonic and protein-altering variants in germline-free ALS and FTD cases.** Somatic variant burdens in ALS genes and non-ALS/FTD neurodegeneration/dementia-related genes were compared between germline-free ALS and FTD cases and normal controls. Enrichment significance and 95% confidence intervals were estimated using a linear mixed model, adjusting for average read-depth, sex, postmortem interval, sequencing batch, and number of samples per donor as potential confounders. Unadjusted *P* values are shown, as the tests examine biologically distinct but nonindependent categories with differing background mutation structures. Control, *n* = 144; ALS, *n* = 216; FTD, *n* = 78; biological replicates.

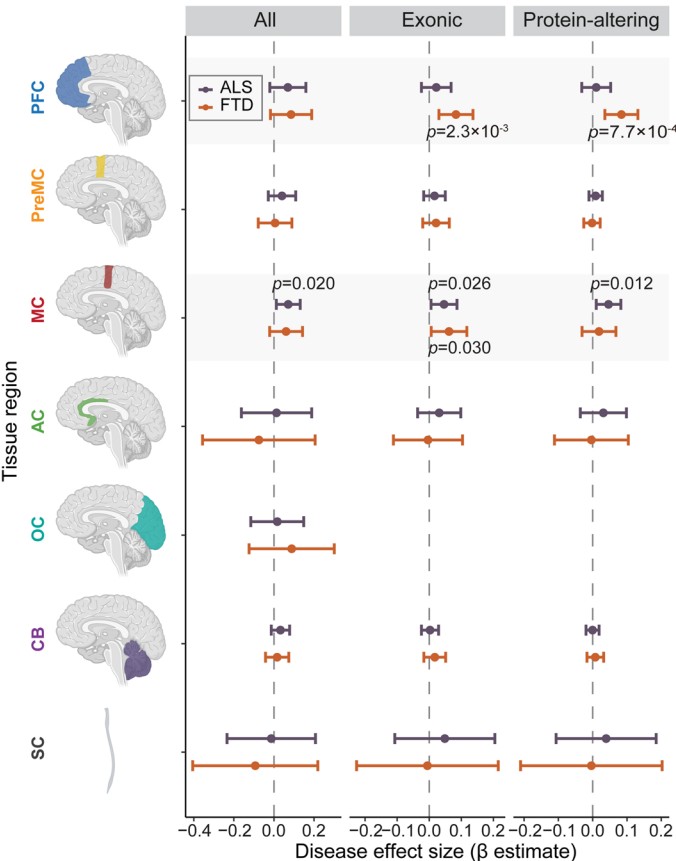

**Extended Data Fig. 6 | Enrichment of overall somatic variants across different brain regions of germline-free ALS and FTD cases compared to normal controls.** All somatic variants within the entire set of targeted neurodegenerative genes were analyzed to test for enrichment. The significance of enrichment and 95% CI were estimated while controlling for potential confounding factors, including average read-depth, sequencing batch, and sampled individuals, using a linear mixed model. Unadjusted *P* values are shown, as the tests examine biologically distinct but nonindependent categories with differing background mutation structures. Control, *n* = 144; ALS, *n* = 216; FTD, *n* = 78; biological replicates. Figure created in BioRender; Zhou, Z. https://BioRender.com/v6v72yx (2026).

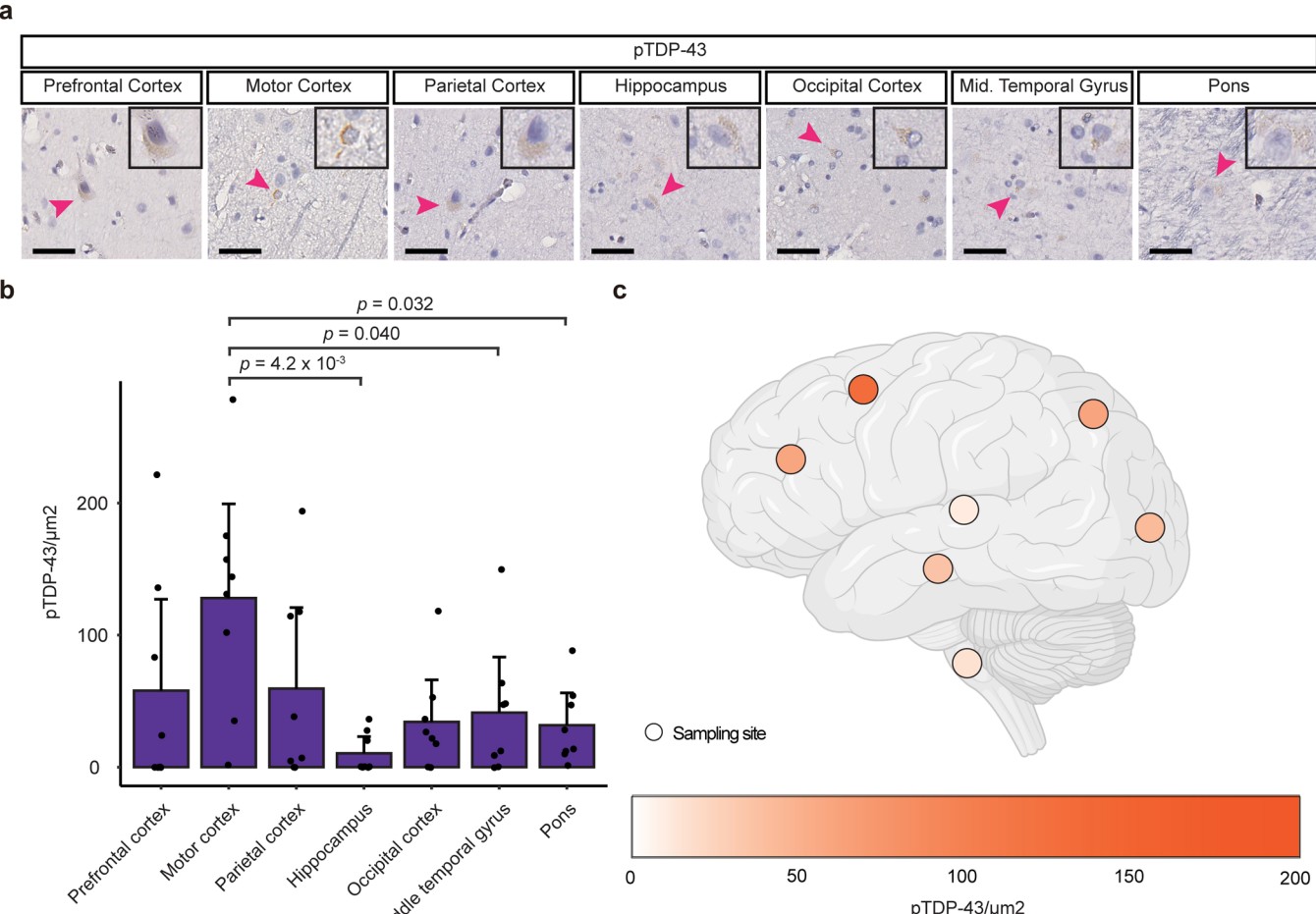

**Extended Data Fig. 7 | Quantification of phospho-TDP-43 staining in brain tissue sections from the sALS case carrying the *TARDBP* (p.L248F) somatic variant. a**, Sections of different brain regions of the sALS case stained with a phospho-TDP-43 antibody. Scale bar, 40 μm. Arrowheads indicate the cells shown in the insets, which are magnified to twice their original size. **b**, pTDP-43 levels across multiple brain regions. Each dot represents the quantification of pTDP-43 in a nonoverlapping tissue section. Eight tissue-section-level biological replicates were included per brain region. Differences in pTDP-43 levels across brain regions were evaluated by one-way ANOVA with post hoc Tukey's HSD correction for multiple comparisons. Bar graph, mean ± 95% CI. **c**, Regional distribution of mean pTDP-43 levels across brain regions. Schematic in **c** created in BioRender; Zhou, Z. https://BioRender.com/v6v72yx (2026).

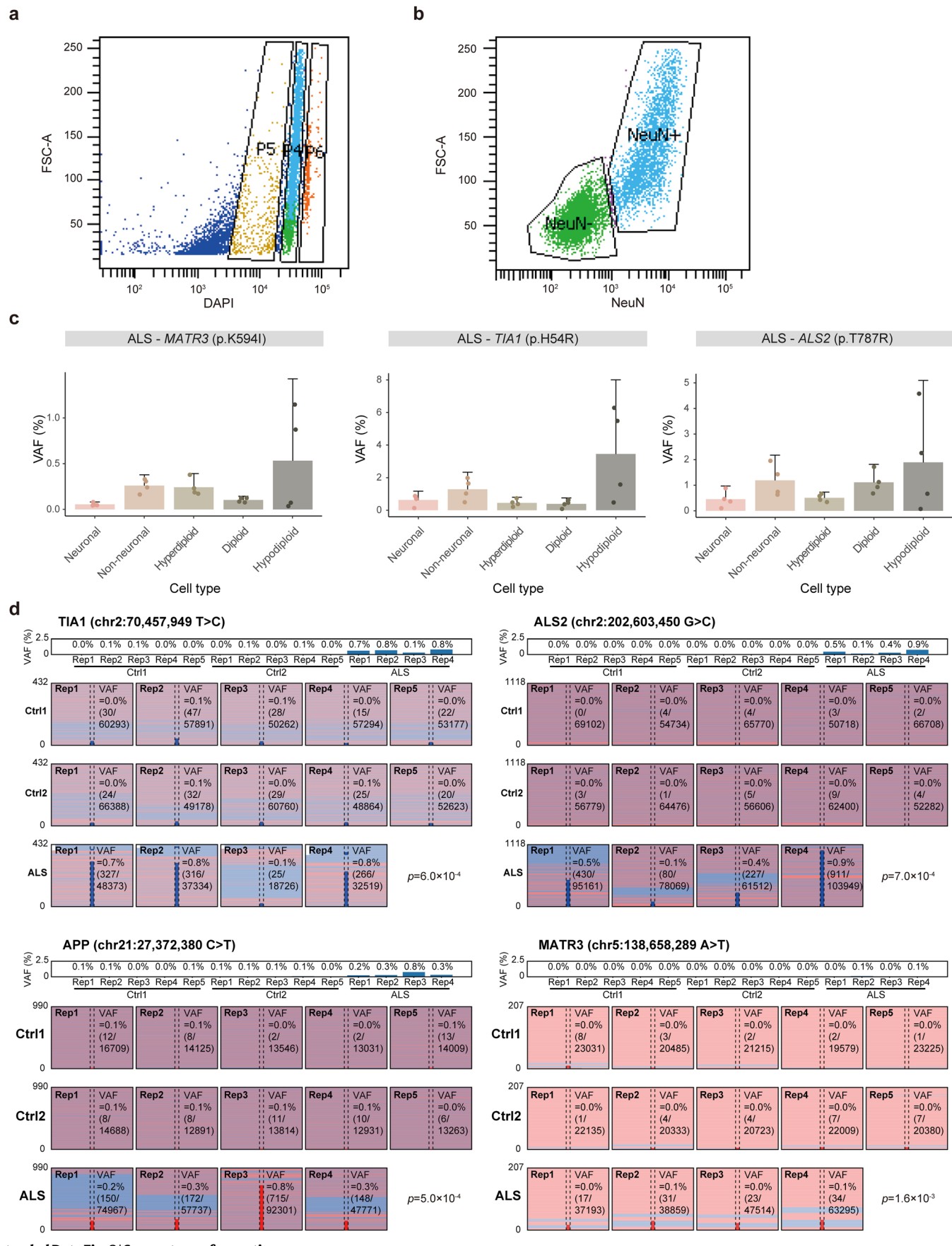

**Extended Data Fig. 8 | See next page for caption.**

**Extended Data Fig. 8 | Predicted deleterious somatic variants are enriched in hypodiploid cells.** All nuclei were stained with DAPI and AF488-conjugated anti-NeuN antibodies to distinguish neuronal and non-neuronal nuclei. **a**, Nuclei with diploid DNA content (P4), hypodiploid DNA content (P5), and hyperdiploid DNA content (P6). Cell debris and doublets were removed by prior gates (forward-scatter (FSC-H) versus FSC-A and side-scatter (SSC-H) versus SSH-W). **b**, Neuronal and non-neuronal nuclei selected from NeuN staining. Neuronal nuclei typically had larger sizes as indicated by the FSC-A values. **c**, VAFs of somatic variants in FANS sorted cell types. Five hundred neuronal (NeuN+), non-neuronal (NeuN−), diploid (DAPI), hyperdiploid (high DAPI) and hypodiploid (low DAPI) cells were each sorted for amplicon sequencing with four replicates. Bar graph, mean ± 95% CI. **d**, Visualization of variant-supporting reads at four somatic variant sites from validation sequencing of sorted neurons. Each subpanel shows a zoomed-in view of variant-supporting reads, with a conpochromasistent scale applied per variant. Pink and light blue horizontal lines represent the reads from positive and negative strands, respectively. Empirical two-sided $P$-values were calculated using a group label permutation test (10,000 iterations) comparing the mean VAF between variant carrier and control groups.

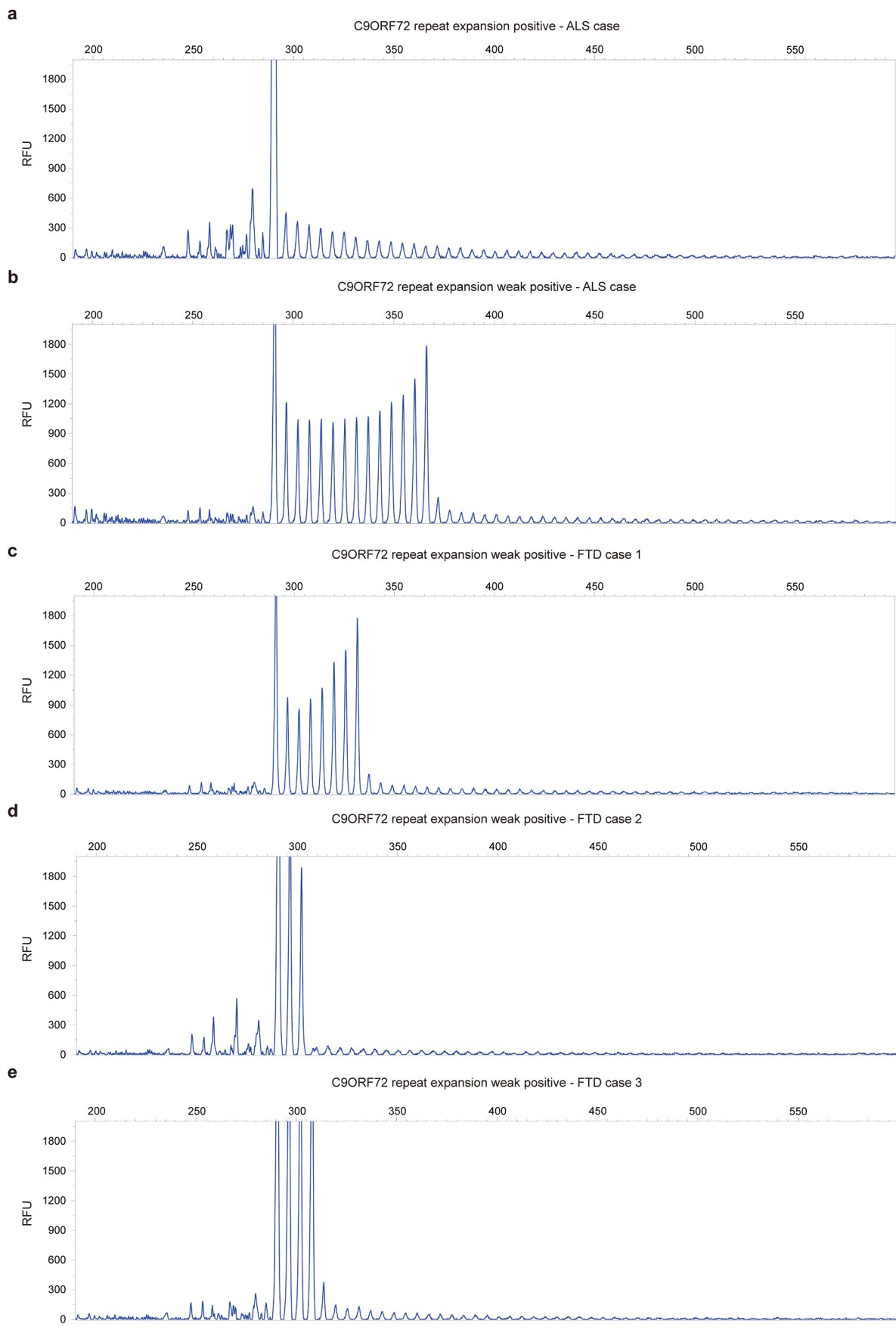

**Extended Data Fig. 9 | See next page for caption.**

**Extended Data Fig. 9 | Detection of somatic *C9orf72* repeat expansion using repeat-primed PCR.** Electropherograms of cerebellar samples from five ALS and FTD cases. **a**, An ALS case exhibited a major peak, indicating a wild-type allele with two repeats. The repeat expansions in this sample showed peaks with high peak heights (reproduced from Supplementary Fig. 2c). **b**–**e**, Four ALS and FTD cases showed major peaks indicating wild-type alleles with various repeat sizes (15, 9, 4, and 5 repeats, respectively). The repeat expansions in these cases showed reduced peak heights compared to the sample shown in **a**, suggesting the presence of somatic *C9orf72* repeat expansion. RFU, relative fluorescence units. *x*-axis denotes the fragment size (bp).

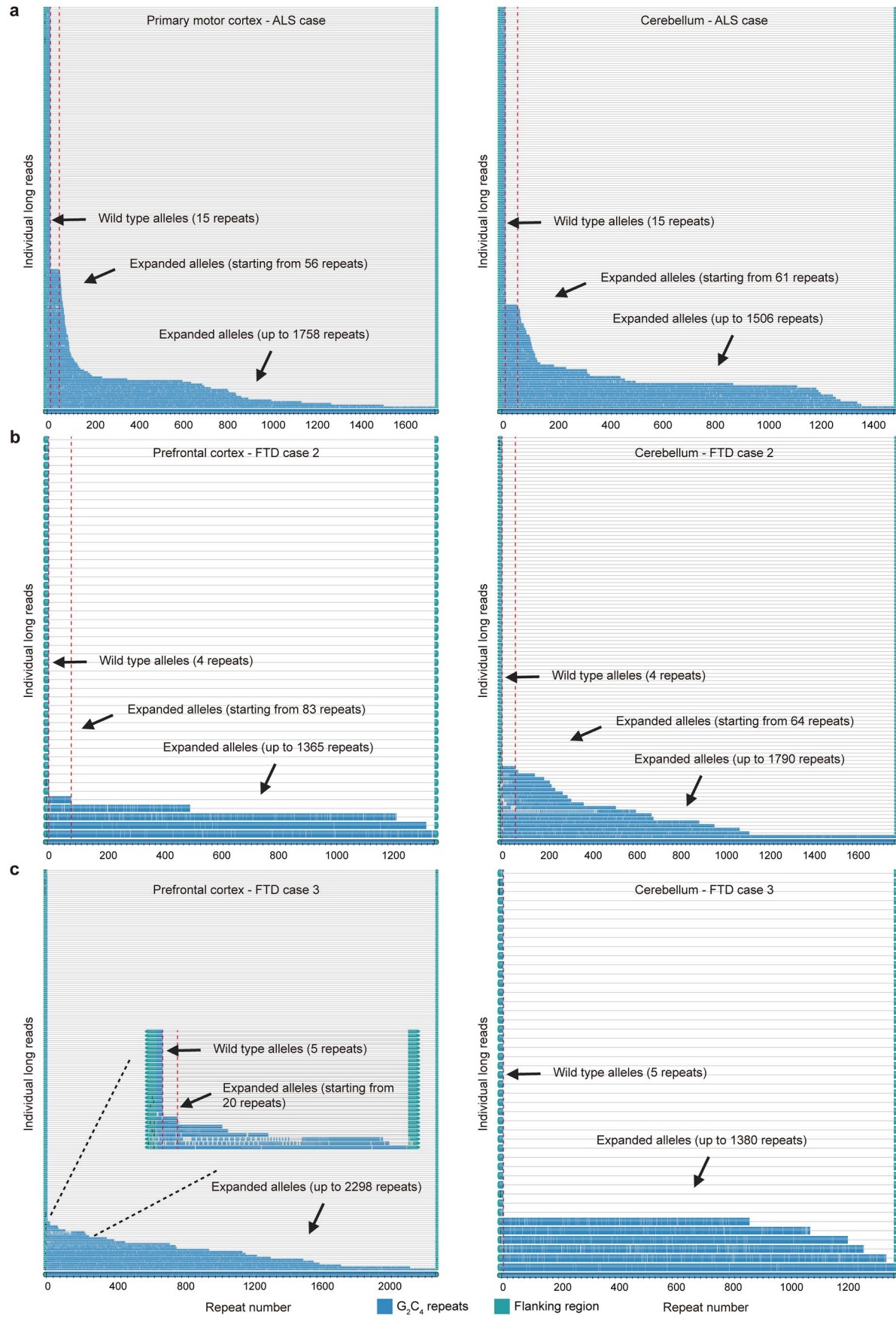

**Extended Data Fig. 10 | See next page for caption.**

**Extended Data Fig. 10 | Detection of somatic *C9orf72* repeat expansion using targeted long-read sequencing.** Waterfall plots from targeted long-read sequencing results of brain tissues from ALS and FTD cases revealed the presence of both wild-type alleles and expanded alleles. **a**, In an ALS case, the primary motor cortex and cerebellum samples showed a wild-type allele with 15 repeats and expanded alleles raging from 56 to 1,758 repeats. **b**, In an FTD case, the prefrontal cortex and cerebellum samples showed a wild-type allele with 4 repeats and expanded alleles ranging from 64 to 1,790 repeats. **c**, In an FTD case, the prefrontal cortex and cerebellum samples showed a wild-type allele with 5 repeats and expanded alleles ranging from 20 to 2,298 repeats. Red dashed lines indicate the sizes of the wild-type alleles and the shortest expanded alleles. Each row in the waterfall plots represents an individual sequencing read, with flanking regions shown in green and the *C9orf72* repeat expansions in blue. The *x*-axis indicates the number of repeats.

| --- | --- |

# Reporting Summary

## Statistics

For all statistical analyses, confirm that the following items are present in the figure legend, table legend, main text, or Methods section.

| n/a | Confirmed | |
| --- | --- | --- |
| ☐ | ☒ | The exact sample size (*n*) for each experimental group/condition, given as a discrete number and unit of measurement |
| ☒ | ☐ | A statement on whether measurements were taken from distinct samples or whether the same sample was measured repeatedly |
| ☐ | ☒ | The statistical test(s) used AND whether they are one- or two-sided  *Only common tests should be described solely by name; describe more complex techniques in the Methods section.* |
| ☐ | ☒ | A description of all covariates tested |
| ☐ | ☒ | A description of any assumptions or corrections, such as tests of normality and adjustment for multiple comparisons |
| ☐ | ☒ | A full description of the statistical parameters including central tendency (e.g. means) or other basic estimates (e.g. regression coefficient) AND variation (e.g. standard deviation) or associated estimates of uncertainty (e.g. confidence intervals) |
| ☐ | ☒ | For null hypothesis testing, the test statistic (e.g. *F*, *t*, *r*) with confidence intervals, effect sizes, degrees of freedom and *P* value noted  *Give P values as exact values whenever suitable.* |
| ☒ | ☐ | For Bayesian analysis, information on the choice of priors and Markov chain Monte Carlo settings |
| ☒ | ☐ | For hierarchical and complex designs, identification of the appropriate level for tests and full reporting of outcomes |
| ☐ | ☒ | Estimates of effect sizes (e.g. Cohen's *d*, Pearson's *r*), indicating how they were calculated |

*Our web collection on statistics for biologists contains articles on many of the points above.*

## Software and code

Policy information about availability of computer code

| Data collection | No collection software was used. |
| --- | --- |
| Data analysis | (1) MIP targeted gene panel sequencing analysis: Sequencing data was processed to generate analysis-ready BAM using BWA-mem (0.7.15), samtools (1.3.1), and GATK (3.7) as described in Methods section. Cutadapt (2.4), BAMClipper (1.0.1), and UMI-tools (1.0.0) were also used to perform preprocessing steps including adapter trimming, read masking, and UMI deduplication. GATK (3.7), RePlow (1.1.0), Mutect2 (4.1.5), and Pisces (5.2.11) were used to identify SNV and indel candidates from MIP sequencing data. ANNOVAR (2017Jul17) was used to deliver computational information of multiple databases for variant pathogenicity prediction. All data visualization and statistical tests including linear mixed-effect regression modelling were performed using R software (4.0.1). The implemented codes for data preprocessing, statistical test, and visualization will be available before publication.  (2) Bulk RNA-seq analysis: STAR (v2.5.0a) was used for read alignment. Picard (1.138) and GATK (3.6) were used for duplicates removal, SplitNCigarReads, indel realignment, and base quality recalibration. RNA-MosaicHunter was used to call somatic SNVs, and its source code and default configuration file are available at https://gitlab.aleelab.net/august/rna-mosaichunter.git.  (3) Targeted long-read sequencing analysis: Raw HiFi reads were aligned to the human reference genome (GRCh38) using minimap2 (v2.28), and sorting and indexing were performed using samtools (v1.3.1). The Tandem Repeat Genotyping Tool (TRGT, v1.1.1) was used to determine the C9ORF72 repeat counts for each sample (chr9:27573528-27573546, (GGGGCC)n). A waterfall plot was generated using TRVZ to visualize repeat counts with mosaicism.  (4) Repeat-primed PCR assay: The RP-PCR products were separated by the SeqStudio Genetic Analyzer (Thermo Fisher) with the GeneScan™ 600 LIZ™ Dye Size Standard (Thermo Fisher). Results of fragment sizes were analyzed by Peak Scanner™ Software v1.0 (Thermo Fisher). |

For manuscripts utilizing custom algorithms or software that are central to the research but not yet described in published literature, software must be made available to editors and reviewers. We strongly encourage code deposition in a community repository (e.g. GitHub). See the Nature Portfolio guidelines for submitting code & software for further information.

## Data

Policy information about availability of data

All manuscripts must include a data availability statement. This statement should provide the following information, where applicable:

- Accession codes, unique identifiers, or web links for publicly available datasets
- A description of any restrictions on data availability
- For clinical datasets or third party data, please ensure that the statement adheres to our policy

The bulk RNA-seq data generated by the NYGC ALS Consortium are available through controlled access via the Target ALS Data Portal (https://dataengine.targetals.org/). Access requires acceptance of the Target ALS Data Use Agreement and submission of a data access request through the portal application process. Additional details are provided in the Target ALS Data Portal User Manual. The MIP-based targeted sequencing data and long-read sequencing data generated in this study have been deposited in dbGaP under accession number phs003530, with access governed by human subject privacy regulations. Germline and somatic variants identified and validated in this study are listed in the supplementary tables.

## Research involving human participants, their data, or biological material

Policy information about studies with human participants or human data. See also policy information about sex, gender (identity/presentation), and sexual orientation and race, ethnicity and racism.

| | |
|---|---|
| Reporting on sex and gender | The cases involved in this study were selected regardless of the sex and gender. The sex and gender were unintentionally balanced. We did not perform specific analyses to compare differences between genders. |
| Reporting on race, ethnicity, or other socially relevant groupings | Race and ethnicity information was not available for all cases involved in this study. Cases were selected regardless of the race and ethnicity. No related analyses were performed. |
| Population characteristics | Sporadic ALS and FTD cases were selected based on available clinical records provided by brain banks. ALS and FTD cases without clear recording of family histories were also selected if the age of death was above 45 years old. Normal cases did not have dementia or other neurological diseases based on available clinical records. |
| Recruitment | The study did not involve live human participants. Case selection is described above. |
| Ethics oversight | Tissue collection and distribution for research and publication was conducted according to protocols approved by the the Massachusetts Alzheimer's Disease Research Center, Oxford Brain Bank, Target ALS Foundation affiliated brain banks , and NIH NeuroBioBank affiliated brain banks. Research on these deidentified specimens and data was performed at Boston Children's Hospital with approval from the Committee on Clinical Investigation. |

Note that full information on the approval of the study protocol must also be provided in the manuscript.

# Field-specific reporting

Please select the one below that is the best fit for your research. If you are not sure, read the appropriate sections before making your selection.

☒ Life sciences  ☐ Behavioural & social sciences  ☐ Ecological, evolutionary & environmental sciences

For a reference copy of the document with all sections, see nature.com/documents/nr-reporting-summary-flat.pdf

# Life sciences study design

All studies must disclose on these points even when the disclosure is negative.

| | |
|---|---|
| Sample size | No sample-size calculation was performed. We tried to collect as many sporadic ALS, FTD and control cases as possible. Our results indicate that the sample size is enough to make statistically significant conclusions. |
| Data exclusions | No data was excluded. |
| Replication | Germline mutations identified in this study were shared by multiple tissue samples from the same individuals, which validates the germline mutations. Forty one somatic mutations identified in the MIP panel sequencing data were further validated by amplicon sequencing. Thirty four of these somatic mutations were also validated by ddPCR. The DYNC1H1 and LMNA somatic mutations identified in bulk RNA-seq data were also validated by amplicon sequencing. |
| Randomization | Cases were classified into ALS, FTD and control groups based on their clinical records and diagnoses. Clinical conditions and covariates of interest (e.g. age, gender, sequencing depth) were modeled as fixed effects and the batch and individual (donor) information were modeled as random effects, considering the uncertainty caused by sample clusters from the same origin (donor or batch). |
| Blinding | No blinding was performed. Blinding was not relevant because samples were postmortem tissues obtained from established brain banks, and downstream sequencing, variant calling, and validation were performed using automated pipelines with predefined criteria independent of disease status. |

# Reporting for specific materials, systems and methods

We require information from authors about some types of materials, experimental systems and methods used in many studies. Here, indicate whether each material, system or method listed is relevant to your study. If you are not sure if a list item applies to your research, read the appropriate section before selecting a response.

## Materials & experimental systems

| n/a | Involved in the study |
|-----|----------------------|
| ☐ | ☒ Antibodies |
| ☐ | ☐ Eukaryotic cell lines |
| ☐ | ☐ Palaeontology and archaeology |
| ☐ | ☐ Animals and other organisms |
| ☐ | ☐ Clinical data |
| ☐ | ☐ Dual use research of concern |
| ☐ | ☐ Plants |

## Methods

| n/a | Involved in the study |
|-----|----------------------|
| ☐ | ☐ ChIP-seq |
| ☐ | ☒ Flow cytometry |
| ☐ | ☐ MRI-based neuroimaging |

## Antibodies

| | |
|---|---|
| Antibodies used | NeuN (Millipore, MAB377, clone A60, 1:1,500) pTDP-43 (CosmoBio, CAC-TIP-PTD-P03, polyclonal, 1:10,000) |
| Validation | The NeuN antibody has been validated for use in FC, IC, IF, IH, IH(P), IP and WB. Its species reactivity has been validated for Av, Ch, Ft, H, M, Po, R, and Sal. This antibody has been cited in more than one thousand publications. All the information is available on the manufacturer's website. The pTDP-43 antibody has been validated for use in WB, ELISA and IHC for detection of human TDP-43 phosphorylated on serine 409. |

## Eukaryotic cell lines

Policy information about cell lines and Sex and Gender in Research

| | |
|---|---|
| Cell line source(s) | n/a |
| Authentication | n/a |
| Mycoplasma contamination | n/a |
| Commonly misidentified lines (See ICLAC register) | n/a |

## Palaeontology and Archaeology

| | |
|---|---|
| Specimen provenance | n/a |
| Specimen deposition | n/a |
| Dating methods | n/a |

☐ Tick this box to confirm that the raw and calibrated dates are available in the paper or in Supplementary Information.

| | |
|---|---|
| Ethics oversight | n/a |

Note that full information on the approval of the study protocol must also be provided in the manuscript.

## Animals and other research organisms

Policy information about studies involving animals; ARRIVE guidelines recommended for reporting animal research, and Sex and Gender in Research

| | |
|---|---|
| Laboratory animals | n/a |
| Wild animals | n/a |
| Reporting on sex | n/a |

| Field-collected samples | n/a |
| Ethics oversight | n/a |

Note that full information on the approval of the study protocol must also be provided in the manuscript.

# Clinical data

Policy information about clinical studies

All manuscripts should comply with the ICMJE guidelines for publication of clinical research and a completed CONSORT checklist must be included with all submissions.

| Clinical trial registration | n/a |
| Study protocol | n/a |
| Data collection | n/a |
| Outcomes | n/a |

# Dual use research of concern

Policy information about dual use research of concern

## Hazards

Could the accidental, deliberate or reckless misuse of agents or technologies generated in the work, or the application of information presented in the manuscript, pose a threat to:

No | Yes
☒ ☐ Public health
☒ ☐ National security
☒ ☐ Crops and/or livestock
☒ ☐ Ecosystems
☒ ☐ Any other significant area

## Experiments of concern

Does the work involve any of these experiments of concern:

No | Yes
☒ ☐ Demonstrate how to render a vaccine ineffective
☒ ☐ Confer resistance to therapeutically useful antibiotics or antiviral agents
☒ ☐ Enhance the virulence of a pathogen or render a nonpathogen virulent
☒ ☐ Increase transmissibility of a pathogen
☒ ☐ Alter the host range of a pathogen
☒ ☐ Enable evasion of diagnostic/detection modalities
☒ ☐ Enable the weaponization of a biological agent or toxin
☒ ☐ Any other potentially harmful combination of experiments and agents

# Plants

| Seed stocks | n/a |
| Novel plant genotypes | n/a |
| Authentication | n/a |

# ChIP-seq

## Data deposition

☐ Confirm that both raw and final processed data have been deposited in a public database such as GEO.

☐ Confirm that you have deposited or provided access to graph files (e.g. BED files) for the called peaks.

| | |
|---|---|
| Data access links<br>*May remain private before publication.* | n/a |
| Files in database submission | n/a |
| Genome browser session<br>(e.g. UCSC) | n/a |

## Methodology

| | |
|---|---|
| Replicates | n/a |
| Sequencing depth | n/a |
| Antibodies | n/a |
| Peak calling parameters | n/a |
| Data quality | n/a |
| Software | n/a |

# Flow Cytometry

## Plots

Confirm that:

☒ The axis labels state the marker and fluorochrome used (e.g. CD4-FITC).

☒ The axis scales are clearly visible. Include numbers along axes only for bottom left plot of group (a 'group' is an analysis of identical markers).

☒ All plots are contour plots with outliers or pseudocolor plots.

☒ A numerical value for number of cells or percentage (with statistics) is provided.

## Methodology

| | |
|---|---|
| Sample preparation | Postmortem brain tissues were homogenized and lysed followed by sucrose gradient based nuclei isolation. Nuclei were stained with NeuN antibody and DAPI before FACS sorting. |
| Instrument | BD FACSAria II |
| Software | BD FACSDiva |
| Cell population abundance | The purify of NeuN+ neurons was not examined in this study. Our previous studies using the same gating strategy showed >96% purity of neurons. |
| Gating strategy | The gating strategy is shown in Extended Data Fig. 6. |

☒ Tick this box to confirm that a figure exemplifying the gating strategy is provided in the Supplementary Information.

# Magnetic resonance imaging

## Experimental design

| | |
|---|---|
| Design type | n/a |
| Design specifications | n/a |
| Behavioral performance measures | n/a |

## Acquisition

| | |
|---|---|
| Imaging type(s) | n/a |
| Field strength | n/a |
| Sequence & imaging parameters | n/a |
| Area of acquisition | n/a |

Diffusion MRI ☐ Used ☒ Not used

## Preprocessing

| | |
|---|---|
| Preprocessing software | n/a |
| Normalization | n/a |
| Normalization template | n/a |
| Noise and artifact removal | n/a |
| Volume censoring | n/a |

## Statistical modeling & inference

| | |
|---|---|
| Model type and settings | n/a |
| Effect(s) tested | n/a |

Specify type of analysis: ☐ Whole brain ☐ ROI-based ☐ Both

| | |
|---|---|
| Statistic type for inference | n/a |

(See Eklund et al. 2016)

| | |
|---|---|
| Correction | n/a |

## Models & analysis

| n/a | Involved in the study |
|---|---|
| ☒ | ☐ Functional and/or effective connectivity |
| ☒ | ☐ Graph analysis |
| ☒ | ☐ Multivariate modeling or predictive analysis |

