## [Peer Review File · Nature Genetics]

Somatic Mosaicism in Amyotrophic Lateral Sclerosis and Frontotemporal Dementia Identifies Focal Mutations Associated with Widespread Degeneration

Corresponding Author: Professor Christopher Walsh

A version of this paper was originally rejected for publication by Nature Genetics, however that decision was reconsidered after appeal by the authors.

Version 0:

Decision Letter:

6th December 2023

Dear Chris,

Your Article "Somatic Mosaicism in Amyotrophic Lateral Sclerosis and Frontotemporal Dementia Reveals Widespread Degeneration from Focal Mutations" has been seen by three referees. You will see from their comments below that, while they find your work of potential interest, they have raised substantial concerns that must be addressed. In light of these comments, we cannot accept the manuscript for publication at this time, but we would be very interested in considering a suitably revised version that addresses the referees' concerns.

We hope you will find the referees' comments useful as you decide how to proceed. If you wish to submit a substantially revised manuscript, please bear in mind that we will be reluctant to approach the referees again in the absence of major revisions.

To guide the scope of the revisions, the editors discuss the referee reports in detail within the team, including with the chief editor, with a view to identifying key priorities that should be addressed in revision, and sometimes overruling referee requests that are deemed beyond the scope of the current study. In this case, we ask that you address all technical concerns related to the methodologies used for variant detection and classification, performing additional validation experiments where feasible and revising the analyses and interpretations where needed, and provide additional clinical phenotyping data where available to aid interpretation of the findings. We hope you will find this prioritized set of referee points to be useful when revising your study. Please do not hesitate to get in touch if you would like to discuss these issues further.

If you choose to revise your manuscript taking into account all reviewer and editor comments, please highlight all changes in the manuscript text file. At this stage, we will need you to upload a copy of the manuscript in MS Word .docx or similar editable format.

*2) If you have not done so already please begin to revise your manuscript so that it conforms to our Article format instructions, available [here](http://www.nature.com/ng/authors/article_types/index.html). Refer also to any guidelines provided in this letter.

Link Redacted

If you wish to submit a suitably revised manuscript, we hope to receive it within 3-6 months. If you cannot send it within this time, please let us know. We will be happy to consider your revision so long as nothing similar has been accepted for publication at Nature Genetics or published elsewhere. Should your manuscript be substantially delayed without notifying us in advance and your article is eventually published, the received date would be that of the revised, not the original, version.

Nature Genetics is committed to improving transparency in authorship. As part of our efforts in this direction, we are now requesting that all authors identified as 'corresponding author' on published papers create and link their Open Researcher and Contributor Identifier (ORCID) with their account on the Manuscript Tracking System (MTS), prior to acceptance. ORCID helps the scientific community achieve unambiguous attribution of all scholarly contributions. You can create and link your ORCID from the home page of the MTS by clicking on 'Modify my Springer Nature account'. For more information, please visit www.springernature.com/orcid.

Thank you for the opportunity to review your work.

Sincerely,
Kyle

Kyle Vogan, PhD
Senior Editor
Nature Genetics
<https://orcid.org/0000-0001-9565-9665>

Referee expertise:

Referee #1: Genetics, neurodegeneration, ALS

Referee #2: Genetics, neurodegeneration, somatic variation

Referee #3: Genetics, neurodegeneration, somatic variation

Reviewers' Comments:

Reviewer #1:
Remarks to the Author:

In the current manuscript by Zhou et al., the authors perform ultra-deep molecular inversion probe sequencing (MIPS) to identify potential somatic variants in amyotrophic lateral sclerosis (ALS) and frontotemporal dementia (FTD) cases. A role of somatic mutations in ALS and FTD has not yet been performed so this represents a novel advance.

They report that cases without a presumably pathogenic inherited variant are more likely to have somatic variants in relevant nervous system tissues, and that as high as 2.7% of this subset of cases could be considered genetically explained if the variants are causal. The findings that somatic variants may be enriched in the motor cortex of ALS cases and the prefrontal cortex of FTD cases, and that some somatic variants could be enriched in non-neuronal populations, are particularly interesting and present an exciting avenue of research if well validated and replicated.

However, there are some concerns with the methods used to discover and validate the candidate somatic variants in this study. The use of MIPS to screen large numbers of samples, while efficient and amenable to scale, comes with the necessity

to confirm observed variants using an orthogonal method. The authors used amplification-based targeted re-sequencing of the candidate variants, and do not propose a secondary method without amplification. Experiments using either amplification-free sequencing or in situ hybridization in isolated nuclei or gene-edited cell lines will be necessary to validate the current conclusions properly.

The manuscript itself is well written, and the figures are well designed. While there are concerns about the reproducibility of the results, I believe this manuscript could be suitable to publish if the concerns are addressed with further experimentation.

Major Comments

My greatest concern with the results of this study is the use of amplified DNA products to study somatic variation. While some of the variants might be true positives, the results currently are not sufficient to rule out errors induced by amplification. A method that does not rely on amplification would be a better validation of the potential somatic variants. Direct DNA sequencing or in-situ allele-specific hybridization would give a better estimate of the variant frequency in a given sample or tissue. Phusion Taq was used for both MIPS and amplicon validation, and somatic variants could be a reproducible artifact inherent to amplification. While Phusion has one of the lower error rates of commercial DNA polymerase enzymes, it would be prudent to generate an estimation of the expected false positive rate in these experiments. Figure 3a demonstrates that the discovery sequencing and the amplicon sequencing have highly correlated VAF estimates – however, the strong correlation also could imply that the error rate is similar between the two methods. Further, the correlation R^2 is only given for the full dataset, whereas the zoomed insert of figure 3a does not appear to have the same R^2 value. Is the number of somatic variants observed significantly different from that expected based on predicted errors inherent in DNA amplification? Was there a correlation between amplification cycle number and VAF? What process was used to optimize the protocol to avoid generating errors?

Because the study makes strong conclusions about the effects of acquired somatic variants, a necessary experiment would be to generate isogenic cell lines that carry a given somatic variant. If a variant is sufficient to instigate pathology through non-cell autonomous mechanisms, it should be demonstrable that the presence of a candidate somatic variant in a cell culture results in ALS-relevant pathology. One suggestion is to perform an experiment with the TARDBP p.L248F variant and test its potential pathogenicity by engineering this variant into an iPSC-derived cell type (e.g., neuron) and seeing if it affects TDP-43 function (e.g., STMN2 cryptic splicing) like other known TDP-43 mutant forms do (e.g., Melamed et al., 2019).

It is not sufficient to conclude that somatic variants are restricted to the site of the initial observation; lack of evidence is not evidence per se. In particular, the TARDBP p.L248F variant should be observable in the original DNA sample that it was derived from, in that 0.5% VAF should be reproducible across a pool of DNA molecules from the original tissue sampling. If this is not the case, it could be either that the original variant call is overestimating the VAF, or that the call is a false positive.

The fluorescence-activated nuclei sorting (FANS) experiment was used to demonstrate that certain variants have higher variant allele fraction (VAF) in non-neuronal populations compared to neuronal. However, that the variant was observed across developmentally different cell lineages suggests that the mutational event occurred early in development. If these variants are of strong effect and impart a non-cell autonomous phenotype, it is necessary to determine experimentally the relationship between VAF and pathogenicity. At the current state, this experiment is preliminary and is not strongly validated to support the claims of VAF estimation and cell-type specificity.

Minor Comments

The introductory paragraph concerning penetrance estimates of ALS/FTD-associated variants would benefit from citing specific examples from the literature (for example, PMID: 37861203).

Line 190: "Ninety-five unique somatic mutations in neurodegeneration-related genes". Are these 95 variants a subset of the 167 variants identified in the prior paragraph? While I interpret this as only the variants observed in the cases without variants in neurodegenerative-associated genes, it is possible to misinterpret this. Further, the 95 variants were observed in large sample numbers from the individuals, which would imply either that the variants are repeatedly observed across tissues from the same individual, or that a given variant is observed across many individuals. However, the results then mention that 76/95 variants were focal and in a single individual. These results would benefit from a more clear explanation of the data.

Line 203: "Indeed, just one protein-altering somatic mutation was observed among all controls". This sounds as though all controls carried the same variant, while I believe the intention is to say that among all controls tested, only one variant was observed (in a certain number of the samples). Please clarify.

Line 725: "MIPs were rebalanced in the pool based on the percent of GC content within the regions". Could the authors precise the rebalancing procedure? Is this to say that probes were increased by an empirically-derived proportion based on pilot sequencing results?

Line 761: "From the aligned BAM file, off-target reads were removed by checking the overlaps with the target regions using bedtools". Why were off-target reads removed from the analysis? Did these reads have any effect on variant calling if they were not assigned a genomic region in the BAM file? What is the rationale for removing reads from a sequencing result?

If the VAF of a variant is < 0.02 and the average read depth is 1800X coverage, this would imply that 36 UMIs were observed

on average per variant call. However, it is unclear in the results whether this is the case, or whether the VAF is composed of multiple reads from the same UMI. The methods mention briefly that UMI-tools was used to deduplicate the mapped reads, but more precision in the methods would help to understand how UMIs were incorporated in this study. Specifically, are UMIs used to distinguish samples, initial capture events within samples, or both?

Variants were designated as pathogenic without supporting evidence. A variant observed in a gene previously associated with ALS does not automatically implicate the variant in ALS risk. Based on the filters and criteria used, the final list of germline variants have a higher probability to be pathogenic, but caution is urged when describing variants that are novel (not previously described in ALS studies or absent in population databases). Please reword "pathogenic" variants to a more conservative wording, such as "candidate" or simply "predicted deleterious" variants.

Line 828: "We additionally found that low-level contamination of DNA from another sample occurred in a few samples." This is particularly concerning in a somatic variant analysis. If this is the case, could the authors be more exact about the quantity of samples that were affected, and the estimated percentage of contaminated reads? If these samples were excluded from the analysis altogether, would that alter the results of the study?

Line 838: "Pathogenic somatic variants were further annotated with similar criteria for selecting pathogenic germline variants." This is circular reasoning. Somatic variants are not a priori pathogenic, and care should be taken to describe them as variants that may be pathogenic based on the current evidence but not pathogenic until proven otherwise.

Could the authors provide an explicit equation for the linear mixed-effect regression model to demonstrate how the covariates (such as age, gender, and sequencing depth) were incorporated into the analysis?

It is unclear what Figure 4b is showing. While it is understood that the variants are subset by column into all, exonic, or protein-altering, and that each row is a different brain region, what is unclear is what disease effect size is describing. The estimates in each row are the odds ratios of how common it is to observe a variant in a given case or control group. "Disease effect size" is therefore likely not the correct descriptor here, and could be simply "odds ratio" or "beta value" depending on the statistical test employed. It may also be interesting (but not necessary) to include a grouped analysis of ALS/FTD instead of splitting the groups.

The error bars extend into negative VAF in Figure 5 for TIA1 p.H54R and ALS2 p.T787R, is this statistically sound on a fixed 0-100 scale?

Statistically speaking, several of the p-values generated in the study approach 0.05. Given the number of statistical analyses described in the text, the likelihood of incurring a type I error is moderately high in this study. While the results are not necessarily incorrect, it would be prudent to include limitations in the discussion section. Currently, the limitations imply that the study should have found more somatic variants rather than providing actual limitations to the results themselves.

Reviewer #2:
Remarks to the Author:

This is a timely, well-conducted, well-presented, original, and very important piece of work. It provides the first systematic evidence for a role of somatic mutations in relevant genes in the aetiology of the ALS/FTD neurodegeneration spectrum. There are some particularly nice aspects which strengthen the conclusions and are highly relevant to the broader field of somatic mutations and neurodegeneration.

(1) Region affected by disease subtype (considering ALS/FTD as a continuum). The different focality in FTD and ALS is exactly as one would expect if the regional distribution of somatic mutations of a given gene/pathway modulates the precise phenotype. This is similar in principle to a recent report of somatic SNCA CNVs in a rare synucleinopathy, multiple system atrophy, being found at different levels in a region differentially affected in disease subtypes, but a similar level in a region affected by both subtypes. DOI: 10.1002/mds.29291

(2) Cell type with mutation. The increased VAF in non-neurons compared to neurons seems counter-intuitive but does indeed suggest that neurons with mutations are more vulnerable and therefore likely dead, so non-neurons serve as a surrogate for neuronal somatic mutation discovery.

(3) The "double hit" in the ALS2 gene (inherited and somatic in a recessive gene) is fascinating, and could trigger a reassessment of the relevance of heterozygous recessive mutations in neurodegenerative diseases. I am not aware of a somatic second hit in other neurodegenerative disorders having been detected.

(4) The possible existence of genes where somatic mutations lead to ALS, but germline are so severe that they would not be classed as ALS genes, may also open new avenues. It is slightly analogous to a case of a somatic PSEN1 mutation causing early AD, which was transmitted to the daughter, who was therefore germline heterozygous and had a completely different very severe phenotype, which would not have suggested that this gene is relevant to Alzheimer's.
<https://doi.org/10.1093/hmg/ddh134>

The authors provide excellent detail to interpret the results using Extended data.

Extended Data Figure 3 is important for benchmarking the approach.

(a) I am intrigued by the separate plotting of Hom and Het mutations and VAF. Of course, 20% DNA spike will give 10% VAF for Het and 20% for Hom. As the plots are, I think the x axis should NOT be labelled target VAF, but with the actual % spike in. The result of plotting Hom and Het together is that the violin plots look a bit messy, since there are two observed peaks. An alternative strategy would be a plot which really represents VAF, so the Hom samples at 5% spike in should be shown together with the Het samples at 10% spike in - since the AF will be 5% in both cases. I think the plot would look much nicer in this way. However, if the authors prefer the current depiction, they are more than welcome to keep it, but they could change the name of the x axis.

(b) I am surprised that even for 10% AF, the sensitivity is ~80%. Am I missing something? I thought UMI with 1800x coverage should have almost 100% sensitivity for 10% AF. It also dramatically declines below 2.5% VAF. Are the filters too stringent? Did the authors look into what is being missed? Also, is this really all 10% AF, or is it a mixture of 10% (het) and 20% (hom) based on the spike in violin plots? I think it would be important to not show Hom and Het together here (as the actual VAF for a given spike in will be double for Hom compared to Het). The "10% target VAF" should correspond to Hom variants of DNA spiked in at 10% and Het variants of DNA spiked in at 20%. If the "10%" actually includes the 20%, as in the violin plot in (a), this is possibly inflating sensitivity for each actual VAF.

In Extended Data Figure 4, there is a sharp peak around 2% - I would expect a more broad low VAF distribution. Is there a reason for this being so different in disease vs controls? Is there a logical explanation for the peak ~18%? I am slightly concerned by the higher VAF % peaks. Could the ~30% AF relate to heterozygous duplication of the other allele? If so, there should also be a peak ~0.7 VAF (I appreciate this would be 0.33/0.67 in theory, but it does seem close to 0.33 in ALS). This could also be checked by seeing if the ~0.3 VAF is consistent in different regions from the same patient - this would be quite a coincidence for a somatic mutation. Also, the upper VAF cut off is necessarily arbitrary for somatic mutations, and 40% seems a good start. Looking at the plots, however, the FTD has an actual VAF peak at 40%. I worry this could be a Het AF, and perhaps the cut off here should be ~35% looking at graph.

I am very surprised by the reported presence of ALS/FTD "novel pathogenic mutations" in 5.6% of controls. I wonder if there are other studies investigating this (not from a mosaic point of view - looking for ALS mutations in cases v controls). I wondered if the pathogenicity is exaggerated, although the authors do suggest that it would require validation. I note that, excluding the GRN mutations, the others are reported as VUS in ClinVar, and the MAPT mutation as benign (I assume it is the same one, although the amino acid notation is different). Perhaps this is worth mentioning.

In the germline variants, I note PSEN1 and ATP13A2. PSEN1 is an Alzheimer gene, and I note the case carrying it also had C9ORF72. It is a minor point in the greater scheme of things, but I don't think it's correct to infer that the PSEN1 is relevant here, if the diagnosis is truly FTD. I guess it's impossible without detailed pathology, or at least detailed clinic assessment, but this individual could have two unrelated neurogenetic conditions. I am not sure how relevant the two carriers of ATP13A2 mutations are (I assume they are heterozygous). This gene was best known as a recessive parkinsonian syndrome, although there are now some cases of complicated HSP reported - perhaps there are also ALS cases, although I have not seen these reports. As far as I know, however, there are no cases reported in heterozygotes - happy to be corrected. So, overall, if these are carriers of mutations which causes a different disease, and only when biallelic/homozygous, I suggest they are removed from the %, and mentioned as of unclear relevance separately - unless one were to invoke that there is also a second somatic hit, but below detection threshold. This is possible but seems a bit too speculative.

Similarly, in Figure 3, there are some somatic mutations in disease cases which appear to me unlikely to be relevant. These include VPS35, a Parkinson's gene, GCH1, a dystonia gene (although this can be so severe it is misdiagnosed as spastic paraplegia <https://www.sciencedirect.com/science/article/pii/S1353802023000330?via=ihub>), and genes which to my knowledge are only pathogenic in a recessive way (biallelic). These include SPG11, which cause complicated HSP, arguably similar to ALS, and PINK1 and PLA2G6, which cause parkinsonism, and would struggle to assume they are relevant. One more gene, UCH-L1, is presumably included because of a single PD report a long time ago.

As the authors reasonably aimed to capture a wide range of genes, it becomes crucial to disentangle which might be relevant to the actual disease presentation. It may not always be clear, but I do not feel it is correct to assume that all these are relevant. If anything, some of the examples above I could see as "control genes" - i.e., genes known to be involved in neurodegeneration (are all such genes prone to somatic mutations?), but likely NOT relevant to these cases. At the very least, I believe figures and % scores need to be separated into "high" and "low" chance of relevance - where "low" would include those involved in other diseases AND/OR known to be pathogenic only when biallelic.

After mutation detection, nuclear sorting is used in a very welcome way, to resolve underlying cell type. It is interesting that most mutation levels are higher in non-neuronal. As the authors have also used Sox10 sorting to enrich for oligodendrocytes, it would have been nice to see this here, but perhaps this can be deferred. The most puzzling is the question of hypodiploid nuclei (I presume not resolved by cell type). Apoptosis is a plausible explanation, but the wide discrepancy between replicates is worrying. Is there an explanation? Were the high VAF in different mutations found in the same sorts? Are there only two replicates in the APP mutation analysis?

The authors are very wise to carefully screen for contamination, which could result in a germline SNP in one sample appearing as a mosaic variant in another one or more. The strategy of excluding any variant "present in less than 0.1% of the

population in any ethnic group of 827 public databases" could even be seen as too stringent, if one accepts that a somatic SNV could by chance be identical to a germline SNP. Is it possible to report the number of variants that were excluded this way, and select a small number for brief discussion - e.g. did samples carrying these also have many other apparent contaminant SNPs at similar AF? If not, are they plausible somatic candidates? On the contrary, accepting only those "observed in < 5 different individuals" could be seen as too lenient. Were there any that were validated, but seen in more than one individual? Would then these be somatic mutation hotspots?

In the succinct but excellent discussion, the authors very reasonably propose (l.374) that "degeneration may spread from a site containing mutant cells to eventually cause loss of neurons in regions that do not carry the mutation". This is indeed a very plausible explanation for sALS, with TDP43 spread. This idea is not novel, e.g. <https://doi.org/10.1016/j.mehy.2011.06.027> [10.1073/pnas.0909343106](https://doi.org/10.1073/pnas.0909343106). The one question not addressed at all here is whether there may be somatic mutation (expansion) of the C9ORF72 tandem repeat, which would not have been detected in the present methods. This needs to be mentioned as a limitation, although I am not aware of any suggestion that a normal germline repeat size can expand somatically. The only work on this I found is a small negative study <https://doi.org/10.1212/NXG.0000000000000317>. The authors encourage the genetic classification of all ALS, regardless of family history, in line with a very recent relevant guideline to offer testing to all; doi: 10.1002/acn3.51895

NOTE:

(1) I am not necessarily expecting the authors to cite all the studies I mention, especially if there are space limitations, but they are very welcome to, if they agree that they further help understand the background and implications.

(2) As far as I can see, there is no preprint. I would encourage the authors to release such important work as a preprint, rather than after the long delays involved in publishing, to benefit the community.

Minor points

Could I ask about the 5 mutations not confirmed by amplicon sequencing - any pattern that could help further curation of low level somatic mutations?

I note whole genome amplification by PTA is mentioned in methods. I am aware of its successful use by the authors for single cell WGS. I am not sure I understand what happened here - were the 500 nuclei of each type per mutation / experiment sorted in bulk, but then each pool of 500 was then amplified by PTA prior to amplicon sequencing?

Please ensure that "µl" and not "ul" is used in methods.

Reviewer #3:

Remarks to the Author:

The authors studied a list of 88 genes using a targeted approach (single-molecule molecular inversion probes) followed by sequencing with an average depth of coverage of 1800x in multiple CNS DNA samples from patients with sporadic Amyotrophic Lateral Sclerosis (ALS), sporadic Frontotemporal Dementia (FTD) (n = 404 patients) and 144 controls.

After detection of likely germline pathogenic variants and secondary exclusion of carriers for further analyses, a significant enrichment in protein-altering variants was observed in ALS/FTD cases as compared to controls, with a specific enrichment in tissues primarily affected by each disease. This result is rather exciting and suggests that at least some of these variants may have contributed to the disease pathophysiology.

Then, the authors attempted to replicate these results in an ALS case - control series (RNAseq data) albeit with only 23 controls.

Overall, I found the paper difficult to judge at this stage for three major reasons: variant classification, variant detection/selection, and patient selection/phenotyping i.e. inclusion/exclusion criteria. Please see specific comments below.

Overall, I think that the authors should clarify what is pathogenic and what is predicted deleterious. They should rely on international recommendations, such as the ACMG-AMP recommendations, to classify variants as (likely) pathogenic or not. It seems that supplementary tables split variants that are pathogenic in patient-specific databases from candidate variants that are predicted deleterious in their algorithm, but in the text we can read "known and novel pathogenic variants" as a whole, which I feel is misleading, especially because the criteria for classification briefly reported in the methods do not allow the replication of the classification of the variants (see examples below with Clinvar benign variants classified as pathogenic).

Mixing such concepts (pathogenicity / in silico predictions) leads to a misevaluation of the rate of germline pathogenic variants, first, and, most importantly, this makes the main message of the paper, i.e. the contribution of somatic variants, questionable.

Interpretation using ACMG-AMP guidelines by a geneticist expert in neurodegenerative diseases should be provided, for (likely) pathogenic variants. This does not preclude a secondary analysis with a burden test allowing the search for an enrichment in coding, predicted deleterious variants, but such a study should rely on homogeneous/unbiased criteria for variant selection.

Doing clinical classification of variants would avoid, for example, seeing a nonsense variant in an exon that is not expressed in the adult brain in the MAPT gene to be rated as probably pathogenic (a gene where the haploinsufficiency mechanism is not causal of FTD, in addition) – although it is in a control sample (and the variant in “benign” in Clinvar), or a missense APP variant outside any functionally relevant domain being rated as probably pathogenic as well.

Many of the variants discussed in the paper are actually variants of uncertain significance and it is unclear how they were selected, given (i) a questioning list of genes, (ii) unclear methods for variants selection.

One of the other main weaknesses is that the authors do not mention the underlying pathology / precise diagnosis in most cases. This study is based on brain/spine banks and I am surprised to read that precise neuropathological diagnoses are not available. Having the subtype of FTD, for example Tau-related, TDP-43 or FUS, seems very important (at least ubiquitin +/-), as well as co-pathologies (Alzheimer's disease). This would avoid seeing a PSEN1 pathogenic variant carrier rated as “FTD” while it is much more probable that the patient actually suffers from frontal variant of AD, I assume. The information on the underlying pathology and copathology seems mandatory to me, for the interpretation of such cases.

Specific queries and comments

1. Comments on sequencing and detection of variants

1.1. What were the criteria for selecting somatic candidate variants for amplicon-based validation? It is unclear how we start from 167 candidate variants to a selection of 41 variants for confirmation, 87.8% of which are true positives, and then 95 unique variants were analyzed (how many are true positives?). Where do the 95 variants come from? I count 105 unique variants in supplementary table 5. What are the “neurodegeneration-related genes”, are these the 88 genes or a selection of it? This is a very important point as it could introduce a bias in all the subsequent analyses and main conclusions of the paper. As some of the variants were detected in only one sample from a specific region, orthogonal validation is required for all variants in the statistical test.

1.2. Regarding the detection of germline variants, in my own experience of smMIPs, the GATK haplotype caller performed quite poorly. I understood that it was likely due to the design of MIPs/amplicon-like reads, so that variants are all located in the same position of a given MIP-based read and this introduces a bias during variant calling (specifically for variants at the extremity of reads). Could the authors comment on this and double check or validate the presence of germline variants?

1.3. Are graphs and numbers on read depth presented after or before deduplication (I guess, after?) Was the sequencing performed in paired-end mode and with which length of reads (F+R reading of same molecules or not?) and what were the rates of duplicates and initial sequencing depth (if results are presented after dedup)?

1.4. Small detail, I would not qualify as “ultra-deep”, sequencing in the order of magnitude of 1000-3000x, but rather for >10,000, although there is no definition of what is “ultra” deep or not. I would not call it ultrasensitive either, as the validation using spike-in shows a maximum of 75% sensitivity for 5-10% mosaics, which is surprisingly low.

1.5. This is not clear if the tools used for somatic variant detection calculate the average error rate (noise) per position in the sequencing run and then the probability for a candidate variant to significantly deviate from the average noise after multiple testing. Could at least the authors add some information on the principle of variant calling in the methods section using the three tools?

1.6. The PPV seems very high (extended data fig 3b) but still only 87.8% of selected somatic variants were confirmed after amplicon-based sequencing, could the authors comment on that?

1.7. It seems useful to compute the theoretical minimal VAF that is detectable using the sequencing method (N UMIs, depth) and the criteria used for the detection of mosaics, i.e. 15 supporting reads, to better figure out the power of the study in terms of VAF resolution.

1.8. Why selecting variant candidates if observed only in disease or control groups but not in both?

1.9. Was there any additional samples with DNA contamination than the “few ones” cited? How many were identified overall? (“few” is not very accurate)

2. I could not find the information on the underlying pathology of these cases. They all come from neuropath biobanks, why the TDP-43, MAPT, FUS pathology would not be available? At least ubiquitin + or - information should be available?

The information on co-pathology with Alzheimer's would also be very useful, as well as Braak&Braak stages. For example, there is a pathogenic germline PSEN1 variant. The patient must have Alzheimer's disease, if the variant is actually pathogenic. I guess that the patient may have frontal variant of AD rather than FTD. The information on neuropath should solve this important issue, and if that is the case, the patient should be excluded from the screened cases as well as all

patients with a final diagnosis of Alzheimer's disease (whatever the clinical presentation upon presentation). Interpretation of additional cases may require a precise diagnosis.

3. It is not clear why patients with unclear family history were selected if age at death above 45 years?

4. The list of genes is rather original and not justified. For example, looking at Alzheimer's disease genes, I see APP, PSEN1, PSEN2 genes which are the only three known autosomal dominant genes. Then, I also see MARK4; I could find only one de novo germline mutation in a single case in the literature in a sporadic AD patient, and MARK2 which has never been associated with AD. In addition, ABCA7 is mentioned along with APOE, which both are risk factor genes and are rated as "autosomal dominant", while the stronger risk factor gene SORL1 – sometimes considered as 4th autosomal dominant gene – is not listed, etc. I understand that some other genes are here as part of differential diagnoses (e.g., SPG4/SPAST). I think that the list should be justified in the methods. Maybe even removing non-relevant genes from the somatic mutation analysis, which should focus on genes that are relevant for the given pathology (e.g. MAPT for Tau-FTD, etc.) as in the second analysis of the somatic variants results. I believe that the global burden does not make much sense given the list of 88 genes.

5. HGVS nomenclature should be used throughout the text and supp tables (e.g., NM_000454:exon1:c.C14T:p.A5V is not an accepted nomenclature)

6. Comments on variant interpretation

6.1. In which database the SOD1:NM_000454:exon4:c.G304A:p.D102N variant is considered as pathogenic? I could not find it in Clinvar and it is in supp table 3 as previously reported to be pathogenic. I guess it is in an ALS-specific database, no doubt it is pathogenic (another missense change at this position is pathogenic in Clinvar), but it would be helpful to have the reference. In the main text, "multiple clinical databases" are mentioned but I could not find them in the methods.

6.2. Variant VAPB:NM004738:exon5:c.G510A:p.M170I is benign in Clinvar (2*, multiple submitters) but still in the reported pathogenic germline variants in Table S3.

6.3. The emphasis on the APP somatic variant is very difficult to understand. The rationale of including APP in a FTD/ALS based study is very questioning as there is no evidence that APP derivatives (including Abeta peptides) may play a role in the pathophysiology of FTD/ALS. If the gene was included as part of a differential diagnosis (which is very questioning based on the fact that authors used a neuropath-based cohort, i.e. the gold standard for diagnostics, thus not requiring genetics for the classification of disease), then the only variants that may cause Alzheimer's disease should be interpreted. They all map to exons 16 and 17 of NM_000484.4 in AD. Here, the missense variant with emphasis in the main text is in exon 7. I am not aware of any hypothesis on how such a variant could have a potential effect on alpha/beta/gamma secretase cleavage of APP or aggregation/neurodegeneration. These kinds of variants of uncertain significance can be found in many controls or patients and do not justify a specific attention in clinical practice. This raises more questions about the interpretation of the distribution of the variant, as the authors seem to put some weight on this latter argument, but for this specific variant, it is clearly not relevant. In addition, in the main text, the authors seem to have included Alzheimer's disease genes, not only as part as differential diagnoses, but also as putative providers of Tau pathology. This is again very questioning given the inclusion criteria: was Alzheimer's disease excluded based on neuropath information?

6.4. In the methods section, it is not clear how pathogenic variants were identified. It seems that benign variants from Clinvar/HGMD were excluded (but from my previous comment, we can see that it is not the case) and the "pathogenic prediction module" was used, but it does not mention Clinvar/other patient-specific databases classification as pathogenic. However, the paper is mainly based on the classification as pathogenic (already reported) vs predicted. This must be clarified.

"missense variant was selected to be pathogenic if at least three different algorithms predicted damaging effects (deleterious for SIFT, LRT, FATHMM, PROVEAN, MetaSVM and MetaLR; probably damaging for PolyPhen2; disease_causing for MutationTaster)": pathogenicity cannot only rely on in silico predictions, even more with so lenient criteria ($\geq 3/8$ tools). They only predict possible deleteriousness on protein products. Pathogenicity likelihood should be rated following international recommendations (ACMG-AMP) and then remaining variants remain missense variants that are predicted deleterious or not (most of them being variants of uncertain significance based on ACMG-AMP recommendations)

It is not acceptable to qualify as "pathogenic" variants only because three bioinformatics tools predict an effect on the protein.

The interpretation of the paper mainly relies on the fact that so-called predicted pathogenic variants are identified as a mosaic. Looking at Table S7 (titled "pathogenic somatic mutations"), I found the above-mentioned of APP, which is everything but pathogenic, I also see a variant of uncertain significance in TARDBP and variants in other genes. I did not re-interpret every variant but I have the feeling that most – if not all - are actually variants of uncertain significance.

6.5. Among the germline variants found in controls, how a nonsense MAPT variant can be considered as predicted pathogenic? Haploinsufficiency of MAPT does not cause FTD, this variant is not present in a transcript expressed in adult brain, and it is already known as benign in Clinvar (table S4)

7. Minor comment, not sure that "oligogenic" well applies to the carrier status for multiple pathogenic variants in autosomal dominant genes

8. Did the author phase ALS2 variants in the patient with a germline heterozygous variant and a somatic variant?
9. How was performed the SMN1 analysis by short-read whole genome sequencing? Is that really reliable, given the difficulty in differentiating SMN1 from SMN2 in short-read data?
10. I did not find very convincing the claims regarding enrichment of somatic variants in hypodiploid cells given low numbers, there are controls or replicates missing to make this result significant (too few variants studied)
11. In the RNA-seq based analysis, were the C9ORF72 expansion carriers excluded from the case-control analysis (it seems that there are a few based on a supp table)? The methods for this part of the analysis are not described in detail. How was the case-control analysis performed? Were all variants included? Any threshold on VAF or coding/non-coding status? I am skeptical that using only 23 controls would be enough to reach significance.
12. Did the patient with the pathogenic somatic DYNC1H1 present any neurodevelopmental phenotype? How was it assessed before stating that the patient does not show the classical phenotype? Some adults can survive with this disease.
13. The pathogenic variant in LMNA (H566Y) is not reported in the Clinvar database, what are the arguments for pathogenicity?
14. If the results are eventually replicated in other datasets and if at least part of these somatic variants are confirmed pathogenic, that would make the pathophysiology of FTD/ALS different from that of AD from that prion-like hypothesis perspective. Indeed, a similar work has already been performed in AD using a similar approach (MIPs), although with a relatively lower number of brain tissues (around 100), but no pathogenic variant was identified. Could the authors comment on why such conditions with prion-like mechanism would differentiate from that perspective? That is also another reason to make sure that there are/ there are no Alzheimer's disease cases in the current series.

Version 1:

Decision Letter:

16th January 2025

Dear Chris,

Your revised Article entitled "Somatic Mosaicism in Amyotrophic Lateral Sclerosis and Frontotemporal Dementia Reveals Widespread Degeneration from Focal Mutations" has been seen by the original referees, whose comments are included below. In the light of the advice of Reviewers #1 and #3, we have decided that we cannot offer to publish your manuscript in Nature Genetics.

In particular, while Reviewer #2 is supportive, Reviewers #1 and #3 raise ongoing concerns about the strength of the evidence supporting the study's claims, and we are persuaded that these reservations are sufficiently important as to preclude publication of this study in Nature Genetics.

I am sorry we cannot be more positive on this occasion, but we hope you will find our referees' comments helpful when preparing your paper for submission elsewhere.

Sincerely,
Kyle

Kyle Vogan, PhD
Senior Editor
Nature Genetics
<https://orcid.org/0000-0001-9565-9665>

Referee expertise:

Referee #1: Genetics, neurodegeneration, ALS

Referee #2: Genetics, neurodegeneration, somatic variation

Referee #3: Genetics, neurodegeneration, somatic variation

Reviewers' Comments:

Reviewer #1 (Remarks to the Author):

I appreciate the efforts that the authors have put in to clarifying the aspects of the initial manuscript that were unclear. The clarifications in variant validation techniques and analysis procedures have made this manuscript easier to understand. I do have a few concerns with the revised manuscript.

The somatic variant in TARDBP (p.L248F) was validated by two orthogonal sequencing methods, supporting that the variant is indeed real and confined to a narrow region of the cortex. However, as this variant is novel and therefore of unknown consequence in ALS, further experimentation is required to claim that this variant is detrimental. In a study of this magnitude, it would be beneficial to see that any of the reported somatic variants result in observable cellular phenotypes. There remains the possibility that the variants observed in this study are those that are tolerated by cells, as acquired somatic variants that result in strong degenerative phenotypes may have selected out in cells carrying these variants and would therefore not be observed. The TARDBP p.L248F variant is located in the previously annotated Nuclear Export Sequence (NES); however, there are reports disputing the necessity or ability of this coding region to instigate TDP-43 nuclear export (<https://doi.org/10.1038/s41598-018-25008-4>, <https://doi.org/10.1371/journal.pbio.3002527>, and <https://doi.org/10.1038/s41598-018-22858-w>). In vitro experimentation to show that this variant either affects export, splicing, TDP-43 aggregation, or some other perturbation would be beneficial to support the potential pathogenicity of this variant. Or, if the claim is that this localized variant results in pathology of nearby brain regions, histological experimentation could show that spread of TDP-43 aggregates begins in this region and lessens over distance from the variant-containing sample.

The choice to alter the regional variant distribution results is acceptable, but as the main result is stated to be that predicted deleterious somatic variants are enriched in hypodiploid cells, it would be helpful to have an explanation as to what these cells could be. The only proposed answer currently is that they are likely apoptotic cells, and while this may be the case, this suggestion should be cited with previous literature.

The use of digital droplet PCR (ddPCR) as an orthogonal validation method is an adequate means to ensure that the variants are real and at the frequency observed. While I am satisfied with the statistical improbability of having amplification errors due to polymerase error rate and UMI inclusion, this also presupposes that there are no unknown sources of error in the protocol. One remaining question on this aspect would be whether the primers used in ddPCR validation are the same as those in MIPS?

Reviewer #2 (Remarks to the Author):

The authors have responded extensively to all my comments, minor and major, and provided considerable additional evidence by using ddPCR and also targeted c9orf72 long read seq. This is now an excellent, comprehensive, well-presented and timely body of work. Christos Proukakis

Reviewer #3 (Remarks to the Author):

The authors have revised the paper, which is much clearer. Overall, the manuscript has been improved. However, now that the methods have been clarified, I see major limitations that make the impact of the study more limited than what the authors claim, from my point of view.

1. Interpretation of the results.

Overall, while some of the somatic variants could be pathogenic, as exemplified in the manuscript with interesting examples and candidates (e.g., the TARDBP variant), the claims are too strong as they suggest that somatic mutations are a not-so-rare disease mechanism. From the data in the paper, I would interpret it as a handful of variants have a potential to be possibly pathogenic, but the vast majority of patients do not show any candidate. Thus, the somatic variant hypothesis would explain a very limited number of cases.

Importantly, the so-called enrichment of somatic variants in ALS cases in ALS-affected regions (resp. FTD) might still be largely driven by unspecific, neurodegeneration-driven processes. Pathogenicity/etiology is not proven here, my personal interpretation is that it could remain that patients with neurodegenerative diseases have a higher burden of somatic variants overall, irrespective of the genes, and some arguments are in favor in this study.

Here, only candidate genes were sequenced in the first part of the study. Hence, the claim that they could be pathogenic is too straightforward as there are no comparisons with other genes. The only available comparisons that would favor specificity are the ones of ALS genes versus FTD-AD genes, but the p-values are close to 0.05 and see also further queries that I have on the statistical aspects that might bias the results towards p-values below 0.05 (see below). I would also be curious to know what happens if Alzheimer genes were removed from FTD/Tau genes (as there is no tauopathy without amyloid in pathogenic variant carriers in these genes, these genes are not relevant to this hypothesis).

Consistent with the hypothesis of increased somatic mutations related to and secondary to neurodegeneration, there is an enrichment of somatic variants, irrespective of the genes, in the bulk RNA analysis, supporting a non-causal effect for most variants and further supporting non-specificity to ALS genes.

In addition, the enrichment in hypodiploid cells is also in favor of a secondary rather than a primary effect. Indeed, re-entry in cell cycle/apoptosis of post-mitotic neurons during neurodegeneration could result in novel somatic variants in these cells, under oxidative stress/neuroinflammation. This does not indicate that they are causal, most of them could be secondary effects of neurodegeneration, whatever the cause.

In addition, one could assume, as an alternate hypothesis, that the loss of neurons in the regions affected by neurodegeneration results in an increased proportion of glial cells in an inflammatory state, both associated with mitosis and oxidative stress, potentially leading to somatic mutations in any gene (and not only explaining the paradox of having more somatic variants in glial cells, as the authors interpret). The reduction of the amount of neurons could also unmask previously present glial variants that arose during mitoses. This would be consistent with the SBS1 signature and enrichment in glial cells in the cell-type analysis.

The fact that there are not necessarily more variants among non-synonymous vs non-coding could also be an argument of non-specificity/causality. This aspect is not so clear, as the comparisons are mostly exonic versus protein altering, but it is not clear how it compares to the non-coding variants, beyond the few synonymous variants. I acknowledge that it remains difficult to interpret the coding/non-coding variant ratios, as non-coding regions were not directly targeted, thus reducing the power to perform such an analysis. However, given the rarity of the somatic synonymous variants, it would remain useful to know whether there is any enrichment of predicted deleterious variants compared to all noncoding variants, and not only synonymous ones?

2. Statistics

I understand that there is a tendency of an enrichment of somatic variants in ALS genes in ALS and in FTD genes in FTD, which is an important argument, but p-values remain close to 5% with lower end of CI very close to 1.

However, I am not sure that the statistics translate what the authors meant. From my understanding of the methods, paragraph "Burden analysis of somatic mutations using linear mixed model" and the rebuttal letter, there is no "individual" effect per se, beyond the fixed effects. Is that correct? Is the α the PMI fixed effect?

If so, that would mean that a somatic variant found in 5 samples from a single individual would count 5 times? As there are more cases than controls, this would artificially increase N in a biased way towards cases. Given the p-values close to 0.05, it becomes likely that there are no significant results if reasoning at the donor level, if I understood the statistics correctly. It would be more logical to me to compare the number of donors with at least one somatic variant (nonsynonymous vs non-coding and synonymous) rather than the number of samples. However, there are more chances to find at least one variant in a case with 5 samples than when 2 samples are available, then a correction based on the number of samples should also be applied.

For these reasons, I feel that the impact of the paper remains limited and that the study lacks evidence of pathogenicity/specificity to support their claims.

Additional comments:

- results on germline are not novel and should probably be reduced in size in the results section.
- in the list of genes (supp table), some are falsely reported as dominant, as they are risk factors, e.g. APOE, ABCA7, TREM2 (the latter is AR for Nasu-Hakola disease, risk factor for AD)
- in the supp table reporting germline variants, the het/hom status would be useful
- analysis of Alzheimer genes along with FTD genes with the argument that AD genes cause tauopathy is misleading, as these genes are related to amyloid, not directly to tau, there is no tau pathology without amyloid in patients carrying these variants. Thus, grouping these genes with the FTD genes makes no sense to me.
- everywhere "deleterious" is used should be "predicted deleterious"
- results, germline paragraph, when referring to germline variants, the term mutation should be avoided, if not demonstrated as de novo, prefer the ACMG class (i.e. pathogenic) and the word variant. Overall, the term mutation is not preferred as it can be misinterpreted as pathogenic by readers.
- SMN analysis : deletions can be missed even by long read seq as they do not easily show differences between SMN1 and 2, and remain the major cause of SMA
- there is a control with Braak stage 5, this is not the best control although it does not alter the results
- supp tables are still in the format TARDBP:NM_007375:exon6:c.G744C instead of TARDBP(NM_007375):c.744G>C

**Although we cannot publish your paper, it may be appropriate for another journal in the Nature Portfolio pending further revision. If you wish to explore other journals and transfer your manuscript, please use our Link Redacted>manuscript transfer portal. You will not have to re-supply manuscript metadata and files, unless you wish to make modifications. For more information, please see our http://www.nature.com/authors/author_resources/transfer_manuscripts.html?WT.mc_id=EMI_NPG_1511_AUTHORTRANSF&WT.ec_id=AUTHOR>manuscript transfer FAQ page.

Version 2:

Decision Letter:

IMPORTANT: Please note the reference number: NG-A63880R1-Z Walsh. This number must be quoted whenever you communicate with us regarding this paper.

21st May 2025

Dear Chris,

Thank you for asking us to reconsider our decision on your manuscript "Somatic Mosaicism in Amyotrophic Lateral Sclerosis and Frontotemporal Dementia Reveals Widespread Degeneration from Focal Mutations". I have discussed your proposed resubmission with my editorial colleagues, and we invite you to revise your manuscript along the lines that you propose for further consideration and peer review.

When preparing a revision, please ensure that it fully complies with our editorial requirements for format and style; details can be found in the Guide to Authors on our website (<http://www.nature.com/ng/>).

Please be sure that your manuscript is accompanied by a separate point-by-point response detailing the changes you have made and your response to the points raised. At this stage, we will need you to upload:

- 1) A copy of the manuscript in MS Word .docx format.
- 2) The Editorial Policy Checklist:
<https://www.nature.com/documents/nr-editorial-policy-checklist.pdf>
- 3) The Reporting Summary:
<https://www.nature.com/documents/nr-reporting-summary.pdf>

(Here you can read about the role of the Reporting Summary in reproducible science:
<https://www.nature.com/news/announcement-towards-greater-reproducibility-for-life-sciences-research-in-nature-1.22062>)

Please use the link below to be taken directly to the site and view and revise your manuscript:

Link Redacted

With best wishes,
Kyle

Kyle Vogan, PhD
Senior Editor
Nature Genetics
<https://orcid.org/0000-0001-9565-9665>

Version 3:

Decision Letter:

1st October 2025

Dear Chris,

Your revised Article "Somatic Mosaicism in Amyotrophic Lateral Sclerosis and Frontotemporal Dementia Reveals Widespread Degeneration from Focal Mutations" has been seen by two of the original reviewers. You will see from their comments below that, while they find the manuscript improved, they have raised a few ongoing concerns. We remain interested in the possibility of publishing your study in Nature Genetics, but we would like to consider your response to these ongoing concerns in the form of a further revision before we make a final decision on publication.

We therefore invite you to revise your manuscript again taking into account all reviewer comments. Please highlight all changes in the manuscript text file. At this stage, we will need you to upload a copy of the manuscript in MS Word .docx or similar editable format.

*2) If you have not done so already, please begin to revise your manuscript so that it conforms to our Article format instructions, available

http://www.nature.com/ng/authors/article_types/index.html >here.

*3) Include a revised version of your Reporting Summary: <https://www.nature.com/documents/nr-reporting-summary.pdf> It will be available to referees (and, potentially, statisticians) to aid in their evaluation if the manuscript goes back for peer review.

Please be aware of our [guidelines](https://www.nature.com/nature-research/editorial-policies/image-integrity) on digital image standards.

EXTENDED DATA FIGURES

Link Redacted

We hope to receive your revised manuscript within 4-8 weeks. If you cannot send it within this time, please let us know.

Nature Genetics is committed to improving transparency in authorship. As part of our efforts in this direction, we are now requesting that all authors identified as 'corresponding author' on published papers create and link their Open Researcher and Contributor Identifier (ORCID) with their account on the Manuscript Tracking System (MTS), prior to acceptance. ORCID helps the scientific community achieve unambiguous attribution of all scholarly contributions. You can create and link your ORCID from the home page of the MTS by clicking on 'Modify my Springer Nature account'. For more information, please visit www.springernature.com/orcid.

Sincerely,
Kyle

Kyle Vogan, PhD
Senior Editor
Nature Genetics
<https://orcid.org/0000-0001-9565-9665>

Referee expertise:

Referee #1: Genetics, neurodegeneration, ALS

Referee #3: Genetics, neurodegeneration, somatic variation

Reviewers' Comments:

Reviewer #1 (Remarks to the Author):

The authors have thoroughly revised their manuscript and have addressed my comments and suggestions. I think this paper is important and will be of interest to the field and I recommend publication in Nature Genetics. I just have one minor comment that I think they should consider addressing as an edit to the text.

Regarding the analysis of the TARDBP p.L248F variant, the data presented are sufficient to support that the variant exists in the tissue sample, but the interpretation of the phospho-TDP-43 (pTDP-43) results seems to be stronger than what the results demonstrate. As different brain regions arise from neurodevelopmental processes and are not linearly correlated with physical distance from the motor cortex, it would be more correct to simply state that pTDP-43 is highest in the region in which the somatic variant is observed. However, as pTDP-43 levels likely correlate with ALS pathology, and ALS pathology is generally localized to the motor cortex and spinal cord, I hesitate to draw any firm conclusions from this analysis. Without further functional data on this variant, it is difficult to accept a claim that the variant is causal, even if it is within the TARDBP gene. I think for the purposes of this manuscript that it would be acceptable to state that the individual has a somatic variant in TARDBP that is potentially pathogenic, but that causality will require further investigation. It is possible that focal somatic mutation and focal pTDP-43 are functionally correlated, without more evidence of focal spread this experiment is more of a case study than a conclusion.

Please consider editing the text to only state that the various brain regions tested displayed different levels of pTDP-43, and that the highest levels of pTDP-43 were observed in the same tissue sample as the somatic variant in TARDBP. Representative images for IHC of pTDP-43 in various brain regions would be informative, if available.

Reviewer #3 (Remarks to the Author):

The authors have greatly improved the manuscript and I am satisfied with most of the answers. To me, as the article stands now, the main conclusions from this article are: (i) replication study on rates of pathogenic germline variants in FTD/ALS genes, (ii) examples of somatic variants that appear as good candidates to explain the disease (e.g. DYNC1H1) and (iii) identification of somatic C9 expansions.

Regarding the main message on the rates of somatic variants in selected genes in ALS/FTD cases compared to controls, this is undoubtedly a very interesting input, but I am still not fully convinced. If statistics are now much clearer and greatly reduce risks of biases, I am not fully convinced that most of these somatic variants are not secondary to neurodegeneration instead of causal. The major argument for causality is certainly the presence in different cell types, neuronal and non-neuronal. However, this has been assessed in a very limited number of individuals, and I would like to ask whether the few dots in the neuronal populations actually separate significantly from the background noise in the sequencing data, despite the fact that the authors state that many cells should carry this variant to observe such a VAF. This experiment is missing a control from other ALS/FTD cases. There is a lot of emphasis on this experiment and a lot of discussion, including the fact that neuronal cells may have died, explaining their low proportion compared to glial cells, which is totally meaningful, if true, but can we make sure that neuronal cells do carry these variants, along the glial cells? That would decrease my concerns by a lot, if the authors can show it. Indeed, the rest of the data remains compatible with a cellular death/apoptosis signal or revealing low-level glia-specific mutations (if they would be glia-specific), including the mutational signatures, the presence of an enrichment in non ALS/FTD genes somatic variants, the only borderline significance of the association of ALS gene somatic variants in ALS cases (are p-values presented after correction for multiple testing?), while the regional distribution of somatic variants does not argue in favor of one or another hypothesis, as somatic variants may be enriched in ALS/FTD affected regions because they are causal, or because they are secondary to neurodegeneration. If this is not technically possible, I would suggest tempering the message related to this part, or, at least, showing the BAM extracts supporting the variants in both directions. If neuronal and non neuronal cells actually show some proportion of true variants, then I agree with the conclusions of the authors.

I was also interested and surprised to read that C9ORF72 expansions might arise from a small normal allele. However, while the authors are very affirmative in the results section, they acknowledge that haplotyping was not clear, in the discussion. As this result is surprising, compared to what we know from this gene and other expansions, this limitation should appear along the results, not only in the discussion section.

Other comments:

- It seems that there is one control with a germline pathogenic variant (C9 expansion?), but I could not find this control individual in supp tables (S4/S1).
- C9 genotyping results of controls are not present in supp files.
- Germline variants: It is not clear how variants of unknown significance are in the "known pathogenic" variants supp tables.
- It seems like NEK1 truncating variant are considered pathogenic, although they are risk factors, not fully penetrant; the ACMG criteria do not apply to risk factors.
- Germline variants part: the classification of variants as known pathogenic versus novel is not very relevant in the whole paragraph, it should be pathogenic/likely pathogenic compared to VUS. Whatever a truncating GRN variant is novel or not, it is likely pathogenic (if predicted to trigger NMD), knowing that it is novel or not is not relevant as lab scientists would rate them as likely pathogenic whatever their known/novel status. These pathogenic variants among the novel variants are

biasing these tests. These tests are interesting as they show what might be remaining, the potential of reclassification, but as it includes variants that are already clearly likely pathogenic, it is not very meaningful. The potential of reclassification is actually in some proportion of the VUS, i.e. among the remaining predicted deleterious variants.

- Germlines variants part: "Consistent with previous reports, multiple cases showed evidence for possible oligogenic inheritance, including several with both C9ORF72 expansions and other variants (Fig. 2d and Supplementary Table 3)": as there are not so many examples here, it would be useful to specify how many cases have multiple pathogenic variants, instead of "multiple cases". In addition, counting only the (likely) pathogenic variants would be more appropriate, as we don't know the potential of all the VUS. Counting the VUS overestimates this proportion of cases with "double" variants.

- Clarification of subgroups of genes is required. Sometimes we can read dementia genes and sometimes, neurodegeneration (e.g. "Our MIP panel contained not only ALS/FTD genes but also genes involved in other dementias. We first focused on somatic variants in all the targeted neurodegeneration genes."). CADASIL (NOTCH3) is not a neurodegenerative disease. Neurodevelopmental diseases associated with EP300 are not neurodegenerative at all, I don't understand why this gene is in the list.

- It still reads strange that the authors tried to exclude spinal muscular amyotrophy by long read sequencing, this method can miss the recurrent pathogenic deletions and this case is not SMA as it shows TDP43 pathology, indeed, maybe this sentence should be skipped in the main text?

- It is difficult to read figure 3c, could the authors apply colors that are easier to separate visually? The different levels of yellow are difficult to read.

- I would suggest to temper the message based on the LMNA variant, as there is no evidence that it is pathogenic, although the case is interesting.

Version 4:

Decision Letter:

Our ref: NG-A63880R3

6th December 2025

Dear Chris,

Your revised manuscript "Somatic Mosaicism in Amyotrophic Lateral Sclerosis and Frontotemporal Dementia Reveals Widespread Degeneration from Focal Mutations" (NG-A63880R3) has been seen by Reviewer #3. As you will see from the comments below, Reviewer #3 is satisfied and has no remaining requests, and therefore we will be happy in principle to publish your study in Nature Genetics as an Article pending final revisions to comply with our editorial and formatting guidelines.

We are now performing detailed checks on your paper, and we will send you a checklist detailing our editorial and formatting requirements soon. Please do not upload the final materials or make any revisions until you receive this additional information from us.

Thank you again for your interest in Nature Genetics. Please do not hesitate to contact me if you have any questions.

Sincerely,
Kyle

Kyle Vogan, PhD
Senior Editor
Nature Genetics
<https://orcid.org/0000-0001-9565-9665>

Reviewer #3 (Remarks to the Author):

The authors have revised the paper according to my comments. In particular, they are providing evidence of the presence of some somatic variants among neurons, which was the main point I raised.

Reviewers' Comments:

Reviewer #1:

Remarks to the Author:

In the current manuscript by Zhou et al., the authors perform ultra-deep molecular inversion probe sequencing (MIPS) to identify potential somatic variants in amyotrophic lateral sclerosis (ALS) and frontotemporal dementia (FTD) cases. A role of somatic mutations in ALS and FTD has not yet been performed so this represents a novel advance.

They report that cases without a presumably pathogenic inherited variant are more likely to have somatic variants in relevant nervous system tissues, and that as high as 2.7% of this subset of cases could be considered genetically explained if the variants are causal. The findings that somatic variants may be enriched in the motor cortex of ALS cases and the prefrontal cortex of FTD cases, and that some somatic variants could be enriched in non-neuronal populations, are particularly interesting and present an exciting avenue of research if well validated and replicated.

However, there are some concerns with the methods used to discover and validate the candidate somatic variants in this study. The use of MIPS to screen large numbers of samples, while efficient and amenable to scale, comes with the necessity to confirm observed variants using an orthogonal method. The authors used amplification-based targeted re-sequencing of the candidate variants, and do not propose a secondary method without amplification. Experiments using either amplification-free sequencing or in situ hybridization in isolated nuclei or gene-edited cell lines will be necessary to validate the current conclusions properly.

The manuscript itself is well written, and the figures are well designed. While there are concerns about the reproducibility of the results, I believe this manuscript could be suitable to publish if the concerns are addressed with further experimentation.

Major Comments

My greatest concern with the results of this study is the use of amplified DNA products to study somatic variation. While some of the variants might be true positives, the results currently are not sufficient to rule out errors induced by amplification. A method that does not rely on amplification would be a better validation of the potential somatic variants. Direct DNA sequencing or in-situ allele-specific hybridization would give a better estimate of the variant frequency in a given sample or tissue. Phusion Taq was

used for both MIPs and amplicon validation, and somatic variants could be a reproducible artifact inherent to amplification. While Phusion has one of the lower error rates of commercial DNA polymerase enzymes, it would be prudent to generate an estimation of the expected false positive rate in these experiments. Figure 3a demonstrates that the discovery sequencing and the amplicon sequencing have highly correlated VAF estimates – however, the strong correlation also could imply that the error rate is similar between the two methods. Further, the correlation R2 is only given for the full dataset, whereas the zoomed insert of figure 3a does not appear to have the same R2 value. Is the number of somatic variants observed significantly different from that expected based on predicted errors inherent in DNA amplification? Was there a correlation between amplification cycle number and VAF? What process was used to optimize the protocol to avoid generating errors?

We thank Reviewer #1 for the insightful comment. To address the concern regarding potential errors introduced by DNA amplification, we first estimated the theoretical number of false positive candidates that could originate from polymerase errors, based on the error rate of the DNA polymerase we utilized (Phusion DNA Polymerase, 4.4×10^{-7}). Since our MIP sequencing incorporates a UMI-based deduplication step, each read after UMI deduplication corresponds to a unique original DNA molecule, meaning any polymerase errors originating from the DNA molecules are independent at the read level. For a given genomic site, assuming a polymerase error rate e and a read depth n , the probability of observing an error with k supporting reads can be modeled using a binomial distribution $Binom(k; n, e)$. For our data, we had an average read depth of 1800X after UMI deduplication, covered by two independent sets of amplicons, each with a depth of 900X. Our variant calling criteria required a minimum of 15 variant-supporting reads, a VAF of 0.1%, and support from two independent amplicon sets. Based on these, the probability of falsely calling a mutation due to amplification error with ≥ 15 supporting reads across two different amplicons can be estimated as follows: $p = (1 - P(X \leq 7))^2$ where $P(X=k) = Binom(k; 900, 4.4 \times 10^{-7})$

This estimation yields a nearly zero chance, even after accounting for the genomic size of the target regions ($\sim 5.8 \times 10^5$ bp), indicating that the likelihood of amplification errors causing the observed results is very low.

However, we acknowledge that theoretical estimates don't necessarily address all concern. Therefore, we further performed droplet digital PCR (ddPCR) as an orthogonal validation, using a pair of fluorescently labeled probes to distinguish wild-type and mutant alleles through hybridization of amplicons of the targeted region. While ddPCR still relies on PCR amplification, it mitigates amplification errors by partitioning the sample into thousands of droplets, enabling digital counting of individual DNA molecules

through binary distinction between the wild-type and mutant allele. Among the 34 tested candidates, 29 were validated as somatic (85.3%) along with two validated as germline and three as negative, supporting a high true positive rate. These results have been updated in the Supplemental Table 6.

Because the study makes strong conclusions about the effects of acquired somatic variants, a necessary experiment would be to generate isogenic cell lines that carry a given somatic variant. If a variant is sufficient to instigate pathology through non-cell autonomous mechanisms, it should be demonstrable that the presence of a candidate somatic variant in a cell culture results in ALS-relevant pathology. One suggestion is to perform an experiment with the TARDBP p.L248F variant and test its potential pathogenicity by engineering this variant into an iPSC-derived cell type (e.g., neuron) and seeing if it affects TDP-43 function (e.g., STMN2 cryptic splicing) like other known TDP-43 mutant forms do (e.g., Melamed et al., 2019).

We thank the reviewer for suggesting the potential experiment for functional validation. However, it is beyond the scope of the current study. The specific TARDBP variant is within the Nuclear Export Signal (NES) sequence of the RRM2 domain of TDP-43, and previous studies have shown that mutations of the NES sequence leads to decreased solubility of TDP-43 and formation of TDP-43 aggregates (PMID: 29728608, 18305110).

It is not sufficient to conclude that somatic variants are restricted to the site of the initial observation; lack of evidence is not evidence per se. In particular, the TARDBP p.L248F variant should be observable in the original DNA sample that it was derived from, in that 0.5% VAF should be reproducible across a pool of DNA molecules from the original tissue sampling. If this is not the case, it could be either that the original variant call is overestimating the VAF, or that the call is a false positive.

We thank the reviewer for bringing this to our attention and regret any confusion caused by the wording in our initial manuscript. The specific variant had been validated in the original DNA sample extracted from the initial sampling site by two independent amplicon sequencing experiments, as well as our newly performed ddPCR experiment. However, it was absent in a second tissue sampling taken several millimeters away from the initial sampling site, which prevented us from conducting cell-type analyses. These results suggest that its distribution is limited to the original sampling site. Similarly, the MATR3 p.K594I and APP p.R328Q variants were also validated by both amplicon-seq and ddPCR. Unlike the first variant, these variants were present in a second sampling adjacent to the initial sampling site, which allowed us to conduct cell-type analyses.

The fluorescence-activated nuclei sorting (FANS) experiment was used to demonstrate that certain variants have higher variant allele fraction (VAF) in non-neuronal populations compared to neuronal. However, that the variant was observed across developmentally different cell lineages suggests that the mutational event occurred early in development. If these variants are of strong effect and impart a non-cell autonomous phenotype, it is necessary to determine experimentally the relationship between VAF and pathogenicity. At the current state, this experiment is preliminary and is not strongly validated to support the claims of VAF estimation and cell-type specificity.

We thank the reviewer for raising this concern. In the revised manuscript, we have removed the somatic *APP* variant and also the proposed non-cell autonomous mechanism related to it, as the specific variant is not definitively disease-relevant.

Minor Comments

The introductory paragraph concerning penetrance estimates of ALS/FTD-associated variants would benefit from citing specific examples from the literature (for example, PMID: 37861203).

We appreciate the reviewer's suggestion and have added several references related to low penetrance, including the one recommended by the reviewer.

Line 190: "Ninety-five unique somatic mutations in neurodegeneration-related genes". Are these 95 variants a subset of the 167 variants identified in the prior paragraph? While I interpret this as only the variants observed in the cases without variants in neurodegenerative-associated genes, it is possible to misinterpret this. Further, the 95 variants were observed in large sample numbers from the individuals, which would imply either that the variants are repeatedly observed across tissues from the same individual, or that a given variant is observed across many individuals. However, the results then mention that 76/95 variants were focal and in a single individual. These results would benefit from a more clear explanation of the data.

We apologize for this confusion. The 95 variants are indeed a subset of the 167 variants. The numbers mentioned by the reviewer are hierarchically related, with the latter counts being subgroups of the former. We agree with the reviewer that the way we presented the numbers in the manuscript may cause confusion, especially since some variants were repeatedly observed across multiple tissues from the same individual. To clarify, we revised the manuscript to report only the number of unique somatic variants (i.e., each variant is counted once, even if observed multiple times across different

tissues). We also updated the Supplementary Table 5 by adding an index to represent each unique variant and merging identical variants from the same individual, which helps recognize variants observed multiple times across tissues.

Below are the updated descriptions and numbers for the variant sets described in the manuscript (please note that all counts have been additionally updated after thorough validation experiments during this revision):

- The total number of unique somatic candidates identified in all individuals in this study: 98 (149 candidates, including repetitive counts across multiple tissues)
- The total number of validated somatic variants in all individuals included in this study: 64
- The number of validated somatic variants observed in individuals who do not carry pathogenic germline variants (germline-free cases and normal controls): 55
- The number of validated focal somatic variants (i.e., observed in only one brain region from a given individual) in individuals who do not carry pathogenic germline variants: 43

Line 203: "Indeed, just one protein-altering somatic mutation was observed among all controls". This sounds as though all controls carried the same variant, while I believe the intention is to say that among all controls tested, only one variant was observed (in a certain number of the samples). Please clarify.

We thank the reviewer for this comment. We revised this sentence as the reviewer suggested: "only one protein-altering somatic mutation was observed among all control cases, whereas 15 and 7 protein-altering mutations were observed in ALS and FTD cases, respectively."

Line 725: "MIPs were rebalanced in the pool based on the percent of GC content within the regions". Could the authors precise the rebalancing procedure? Is this to say that probes were increased by an empirically-derived proportion based on pilot sequencing results?

The rebalancing was not based on the sequencing result of the panel. Instead, we rebalanced the pool of MIP probes based on GC content, which has been observed to reduce the coverage of regions when using MIPs in other projects conducted by ours and other groups (PMID: 39019033). To account for this, we duplicated MIPs that bind to higher GC content regions to attempt to increase coverage. Specifically, we calculated the GC content for the extension and ligation arms of each MIP. If the GC content is less than 60%, we added only one copy of the MIP. However, for higher GC content, we added the following number of copies: 60-70% = 2 copies, 70-80% = 5 copies, 80-90% = 8 copies, and more than 90% = 10 copies. We have included this

information in the Methods section of the revised manuscript.

Line 761: “From the aligned BAM file, off-target reads were removed by checking the overlaps with the target regions using bedtools”. Why were off-target reads removed from the analysis? Did these reads have any effect on variant calling if they were not assigned a genomic region in the BAM file? What is the rationale for removing reads from a sequencing result?

We thank the reviewer for this comment. Off-target reads are those that have been aligned to genomic regions outside the intended target regions. These can occur due to non-specific binding of probes, PCR artifacts, library preparation artifacts, and sequencing artifacts. We believe removing these reads helps to increase the specificity and accuracy of variant detection by eliminating low-confidence calls located outside the region of interest. In addition, we confirmed that there is no difference between the variant call sets with and without removing off-target reads. Our average off-target read ratio was approximately 12%, which is comparable to or even lower than that of typical targeted sequencing approaches.

If the VAF of a variant is < 0.02 and the average read depth is 1800X coverage, this would imply that 36 UMIs were observed on average per variant call. However, it is unclear in the results whether this is the case, or whether the VAF is composed of multiple reads from the same UMI. The methods mention briefly that UMI-tools was used to deduplicate the mapped reads, but more precision in the methods would help to understand how UMI were incorporated in this study. Specifically, are UMI used to distinguish samples, initial capture events within samples, or both?

It is correct that a variant with a VAF of 0.02 would get approximately 36 UMIs at an average depth of 1800X. In our study, UMIs were utilized to tag each DNA molecule to distinguish the initial capture events. All sequencing depths reported in the manuscript reflect the counts after UMI deduplication. In addition to the UMIs, we used sample barcodes during library preparation to distinguish samples. We have updated the Methods section to specify that the average sequencing depth is obtained after UMI deduplication. We have also provided a figure showing read depth before and after UMI deduplication in Extended Data Fig. 1.

Variants were designated as pathogenic without supporting evidence. A variant observed in a gene previously associated with ALS does not automatically implicate the variant in ALS risk. Based on the filters and criteria used, the final list of germline variants have a higher probability to be pathogenic, but caution is urged when describing variants that are novel (not previously described in ALS studies or absent in

population databases). Please reword “pathogenic” variants to a more conservative wording, such as “candidate” or simply “predicted deleterious” variants.

We thank the reviewer for raising this concern. We agree and have replaced the term “pathogenic” with “predicted deleterious” for novel variants.

Line 828: “We additionally found that low-level contamination of DNA from another sample occurred in a few samples.” This is particularly concerning in a somatic variant analysis. If this is the case, could the authors be more exact about the quantity of samples that were affected, and the estimated percentage of contaminated reads? If these samples were excluded from the analysis altogether, would that alter the results of the study?

We apologize for the insufficient description regarding the contamination issue in our samples. We checked and confirmed contaminated samples and their contaminant sources by comparing the somatic candidate sets against the germline variant sets of other individuals, as described in the Methods section. Among the 1,787 samples in our study, we identified potential contamination in 29 samples (13 ALS, 10 FTD, and 6 control samples), which represents 1.6% of the total sample pool. Of these, 23 samples did not have any somatic candidates besides the contaminated sites, five had one candidate, and one had two candidates. The average proportion of reads estimated to originate from the contamination source was 2.8%, inferred from the average VAF of the contaminating germline variants.

We conducted a thorough analysis to ensure that the identified somatic variants were not artifacts of contamination. We checked the discordance between the observed VAF of these somatic candidates and the estimated rate of contamination, as well as verified the absence of corresponding variants within the contaminant sources, confirming they are independent candidates.

None of the somatic variants from the contaminated samples were located within the exonic regions of ALS or FTD genes. Therefore, excluding these contaminated samples would not alter the results in our study. We have revised the manuscript to include detailed descriptions of our observations and the methods to address this issue.

Line 838: “Pathogenic somatic variants were further annotated with similar criteria for selecting pathogenic germline variants.” This is circular reasoning. Somatic variants are not a priori pathogenic, and care should be taken to describe them as variants that may be pathogenic based on the current evidence but not pathogenic until proven otherwise.

We appreciate the reviewer's thoughtful comment and apologize for creating this confusion. This part was intended to describe the annotation process for somatic variant candidates, which is similar to the process we used for germline variant candidates. We have revised this paragraph to explain this process in detail: "Somatic variants were further annotated with similar criteria for selecting deleterious germline variants. Among the final candidates, variants that are 1) observed only in disease or control groups but not in both, 2) possible protein-altering variants, and 3) affecting ALS- and FTD-related genes were selected and applied for the deleterious effect prediction module. ANNOVAR and SpliceAI were utilized to annotate variants with various genomic information and detect additional splice-altering variants, respectively." In addition, we also replaced the term "pathogenic somatic variants" with "predicted deleterious somatic variants" throughout our manuscript.

Could the authors provide an explicit equation for the linear mixed-effect regression model to demonstrate how the covariates (such as age, gender, and sequencing depth) were incorporated into the analysis?

We appreciate the reviewer's thoughtful comment and have added the equation of the linear mixed-effect model as follows: $y_{ij} = \mu + \alpha_i + \beta_i + \gamma_i + \delta_{ij} + U_{ij} + \varepsilon_{ij}$, where y_{ij} is the somatic mutation burden in sample j from donor i , μ is average mutation burden in normal condition, β_i is the fixed effect of disease status relative to normal condition (ALS, FTD) from donor i , γ_i is the fixed effect of the sex of donor i , and δ_{ij} is the fixed effect of the average sequencing depth of sample j from donor i . $U_{ij} \sim N(0, \sigma_r^2)$ is the random effect of the sequencing batch, and $\varepsilon_{ij} \sim N(0, \sigma^2)$ is the measurement error of each sample. In the analysis, covariates with a p-value < 0.05 were considered to be significant based on a t-test using the Satterthwaite approximation of degrees of freedom. We decided not to include age in our model because most of the low-VAF somatic variants observed in this study should arise during the late stages of cortical development (rather than clonally expanding later in life, as seen in cancer); therefore, their burden should not be influenced by an individual's age.

It is unclear what Figure 4b is showing. While it is understood that the variants are subset by column into all, exonic, or protein-altering, and that each row is a different brain region, what is unclear is what disease effect size is describing. The estimates in each row are the odds ratios of how common it is to observe a variant in a given case or control group. "Disease effect size" is therefore likely not the correct descriptor here, and could be simply "odds ratio" or "beta value" depending on the statistical test employed. It may also be interesting (but not necessary) to include a grouped analysis of ALS/FTD instead of splitting the groups.

We apologize for the confusion caused by the descriptors in Fig. 4b. In this figure, we conducted a mutation burden analysis for each brain region using a linear mixed model described in Methods. The estimates shown specifically relate to the clinical condition (after correcting for other covariates in the model, though their estimates are not displayed), which is why we originally referred to them as “disease effect size”. However, we recognize that this term might have been unclear in conveying how the values were derived. Therefore, following the reviewer’s suggestion, we have added “ β estimate” to the descriptor to more accurately reflect the statistical output of the model. The key result illustrated by this figure is the distinct enrichment patterns of somatic variants in disease-affected regions between ALS and FTD groups. Therefore, we decided not to include a combined analysis of ALS and FTD here.

The error bars extend into negative VAF in Figure 5 for TIA1 p.H54R and ALS2 p.T787R, is this statistically sound on a fixed 0-100 scale?

We appreciate the reviewer’s comment regarding the error bars for the confidence intervals in Fig. 5. The data points for the *TIA1* and *ALS2* variants showed large variance, leading to wide confidence intervals that extended below zero. To address this, we initially considered applying a logit transformation to constrain the confidence intervals within the 0-1 range, but this approach resulted in excessively large confidence intervals for data points near zero, even when the data points were clustered. Instead, we decided to display only the upper limits of the confidence intervals using the t-distribution for these variants. Additionally, we applied jitter to better visualize the actual data points. These results are currently moved to Extended Data Fig. 7.

Statistically speaking, several of the p-values generated in the study approach 0.05. Given the number of statistical analyses described in the text, the likelihood of incurring a type I error is moderately high in this study. While the results are not necessarily incorrect, it would be prudent to include limitations in the discussion section. Currently, the limitations imply that the study should have found more somatic variants rather than providing actual limitations to the results themselves.

We appreciate the reviewer’s insightful comments regarding the limitation of the statistical analyses in our study. We have included the following description in the discussion section to address this: “On the other hand, our analysis also harbors inherent statistical limitations. Several p-values generated in the study approach the 0.05 threshold, indicating a moderate risk of type I errors given the number of statistical tests performed. This underscores the need for future studies with larger sample sizes and more sensitive approaches.”

Reviewer #2:

Remarks to the Author:

This is a timely, well-conducted, well-presented, original, and very important piece of work. It provides the first systematic evidence for a role of somatic mutations in relevant genes in the aetiology of the ALS/FTD neurodegeneration spectrum. There are some particularly nice aspects which strengthen the conclusions and are highly relevant to the broader field of somatic mutations and neurodegeneration.

(1) Region affected by disease subtype (considering ALS/FTD as a continuum). The different focality in FTD and ALS is exactly as one would expect if the regional distribution of somatic mutations of a given gene/pathway modulates the precise phenotype. This is similar in principle to a recent report of somatic SNCA CNVs in a rare synucleinopathy, multiple system atrophy, being found at different levels in a region differentially affected in disease subtypes, but a similar level in a region affected by both subtypes. DOI: 10.1002/mds.29291

(2) Cell type with mutation. The increased VAF in non-neurons compared to neurons seems counter-intuitive but does indeed suggest that neurons with mutations are more vulnerable and therefore likely dead, so non-neurons serve as a surrogate for neuronal somatic mutation discovery.

(3) The "double hit" in the ALS2 gene (inherited and somatic in a recessive gene) is fascinating, and could trigger a reassessment of the relevance of heterozygous recessive mutations in neurodegenerative diseases. I am not aware of a somatic second hit in other neurodegenerative disorders having been detected.

(4) The possible existence of genes where somatic mutations lead to ALS, but germline are so severe that they would not be classed as ALS genes, may also open new avenues. It is slightly analogous to a case of a somatic PSEN1 mutation causing early AD, which was transmitted to the daughter, who was therefore germline heterozygous and had a completely different very severe phenotype, which would not have suggested that this gene is relevant to Alzheimer's. <https://doi.org/10.1093/hmg/ddh134>

The authors provide excellent detail to interpret the results using Extended data.

Extended Data Figure 3 is important for benchmarking the approach.

(a) I am intrigued by the separate plotting of Hom and Het mutations and VAF. Of

course, 20% DNA spike will give 10% VAF for Het and 20% for Hom. As the plots are, I think the x axis should NOT be labeled target VAF, but with the actual % spike in. The result of plotting Hom and Het together is that the violin plots look a bit messy, since there are two observed peaks. An alternative strategy would be a plot which really represents VAF, so the Hom samples at 5% spike in should be shown together with the Het samples at 10% spike in - since the AF will be 5% in both cases. I think the plot would look much nicer in this way. However, if the authors prefer the current depiction, they are more than welcome to keep it, but they could change the name of the x axis.

We appreciate the reviewer's thoughtful comments. We agree that labeling the x-axis as "spike-in level" rather than "target VAF" more accurately reflects the data presented, therefore we have updated it accordingly. We have decided to keep the original format of the violin plots, as one of our objectives with this figure is to illustrate the VAF distribution of Hom and Het variants within the same sample.

(b) I am surprised that even for 10% AF, the sensitivity is <80%. Am I missing something? I thought UMI with 1800x coverage should have almost 100% sensitivity for 10% AF. It also dramatically declines below 2.5% VAF. Are the filters too stringent? Did the authors look into what is being missed? Also, is this really all 10% AF, or is it a mixture of 10% (het) and 20% (hom) based on the spike in violin plots? I think it would be important to not show Hom and Het together here (as the actual VAF for a given spike in will be double for Hom compared to Het). The "10% target VAF" should correspond to Hom variants of DNA spiked in at 10% and Het variants of DNA spiked in at 20%. If the "10%" actually includes the 20%, as in the violin plot in (a), this is possibly inflating sensitivity for each actual VAF.

Somatic variant calling targeting low-VAF variants inevitably comes with many false positive candidates due to various aspects of sequencing and alignment errors. To address these false positives, we applied a series of filters, such as requiring a minimum number of variant-supporting read counts, read-depth, and base-call quality scores. One of the most effective criteria was requiring two distinct types of variant-supporting amplicons, which eliminated hundreds of low-VAF false positive candidates. However, due to the panel design, some regions cannot be covered by more than one amplicon. Many spike-in variants were located within such regions and thus could not be called as somatic candidates based on our filtering criteria, leading to a decrease in sensitivity even for targeting 10% AF. Initially, we reported the benchmarking results including these variants to be conservative, but we found that this may mislead readers to think our pipeline has low sensitivity, just as the reviewer pointed out. Therefore, we now exclude spike-in variants from benchmarking if 1) a given site could be covered by only one amplicon, or 2) had a read depth of less than 100, which were inherent

limitations of the panel design rather than the variant calling process. This adjustment resulted in a sensitivity of 96% for 10% AF.

We also agree with the reviewer that including homozygous spike-in variants could inflate the benchmarking results. Accordingly, we revised our benchmarking process to consider only heterozygous variants, providing a more accurate reflection of target VAFs. We have updated Extended Data Fig. 3b and manuscript to reflect these changes.

In Extended Data Figure 4, there is a sharp peak around 2% - I would expect a more broad low VAF distribution. Is there a reason for this being so different in disease vs controls? Is there a logical explanation for the peak ~18%? I am slightly concerned by the higher VAF % peaks. Could the ~30% AF relate to heterozygous duplication of the other allele? If so, there should also be a peak ~0.7 VAF (I appreciate this would be 0.33/0.67 in theory, but it does seem close to 0.33 in ALS). This could also be checked by seeing if the ~0.3 VAF is consistent in different regions from the same patient - this would be quite a coincidence for a somatic mutation. Also, the upper VAF cut off is necessarily arbitrary for somatic mutations, and 40% seems a good start. Looking at the plots, however, the FTD has an actual VAF peak at 40%. I worry this could be a Het AF, and perhaps the cut off here should be ~35% looking at graph.

We thank the reviewer for the thoughtful comments. In another study from our lab (PMID: 37986891), we showed that somatic variants with low VAFs are restricted to a small brain region, while those with high VAFs are typically shared across multiple regions. The majority of our somatic candidates were indeed focal, which led to the peak around 2%. The observed narrower VAF distribution, contrary to the reviewer's expectations, is primarily due to the small number of somatic candidates in our study, which led to a sparse distribution. The specific VAF peak around 18% was based on only a few observations and should not be considered to be a general trend. Unlike the ALS and FTD samples, which showed an enrichment of low-VAF variants, the control samples had only a few variants with uniformly distributed VAFs, resulting in a quite different VAF distribution. Histograms might better represent such a small dataset, but we decided to present them with density plots to provide a more illustrative view of the overall VAF distribution of somatic variants. During this revision, we conducted additional validation for all somatic candidates and further removed several false positives. The reduced number of candidates eliminated several peaks, including the one around 8% that the reviewer pointed out.

We also agree with the reviewer that the FTD variant with a VAF around 40% could be a germline variant. Considering this, we adjusted the cutoff for germline variants to 35% in Extended Data Fig. 4, as the reviewer suggested.

I am very surprised by the reported presence of ALS/FTD "novel pathogenic mutations" in 5.6% of controls. I wonder if there are other studies investigating this (not from a mosaic point of view - looking for ALS mutations in cases v controls). I wondered if the pathogenicity is exaggerated, although the authors do suggest that it would require validation. I note that, excluding the GRN mutations, the others are reported as VUS in ClinVar, and the MAPT mutation as benign (I assume it is the same one, although the amino acid notation is different). Perhaps this is worth mentioning.

We agree with the reviewer that the pathogenicity prediction could be exaggerated. Our intention was to remove as many cases carrying potentially damaging germline variants as possible, so that we could better evaluate the impact of somatic variants in true sporadic cases. In the current revision, we have reclassified the candidates following the ACMG Standards and Guidelines. Two germline variants were removed as they were classified as benign, including the *MAPT* variant mentioned by the reviewer. Following the reviewer's suggestion, we have also added the following sentence to further explain this: "Although our pathogenicity predictions for novel germline variants may overestimate their impact, this approach was taken to exclude as many cases with potentially deleterious germline variants as possible."

In the germline variants, I note PSEN1 and ATP13A2. PSEN1 is an Alzheimer gene, and I note the case carrying it also had C9ORF72. It is a minor point in the greater scheme of things, but I don't think it's correct to infer that the PSEN1 is relevant here, if the diagnosis is truly FTD. I guess it's impossible without detailed pathology, or at least detailed clinic assessment, but this individual could have two unrelated neurogenetic conditions. I am not sure how relevant the two carriers of ATP13A2 mutations are (I assume they are heterozygous). This gene was best known as a recessive parkinsonian syndrome, although there are now some cases of complicated HSP reported - perhaps there are also ALS cases, although I have not seen these reports. As far as I know, however, there are no cases reported in heterozygotes - happy to be corrected. So, overall, if these are carriers of mutations which causes a different disease, and only when biallelic/homozygous, I suggest they are removed from the %, and mentioned as of unclear relevance separately - unless one were to invoke that there is also a second somatic hit, but below detection threshold. This is possible but seems a bit too speculative.

We apologize for the confusion caused by the results presented in Fig. 2 and Supplementary Table 3. We aimed to report all germline variants that could be of interest, including the *ATP13A2* variants, as highlighted by the reviewer. However, we did not use these variants to classify individuals as having pathogenic germline variants. For example, as the reviewer correctly noted, *ATP13A2* is a recessive gene and all patients with *ATP13A2* germline variants in our study were heterozygous. Therefore, we considered these patients, along with those carrying heterozygous germline variants in *ALS2*, as germline-free cases. We classified an individual as a germline-risk carrier only if they harbored a known/predicted deleterious variant in an autosomal dominant gene previously reported to be associated with ALS/FTD, which is described in Supplementary Table 2. To clarify this point, we have added a new column labeled “Note” in Supplementary Table 3 to indicate recessive genes. Additionally, we have removed the *PSEN1* variant from the list as it is classified as VUS by the ACMG guidelines. We also revised the manuscript to clearly state that such patients are considered as germline-free cases, as follows: “Patients with germline deleterious mutations in non-ALS/FTD genes or with a heterozygous mutation in recessive genes (e.g. *ATP13A2*, *ALS2*) were also considered germline-free cases and included in the somatic analysis.”

Similarly, in Figure 3, there are some somatic mutations in disease cases which appear to me unlikely to be relevant. These include VPS35, a Parkinson's gene, GCH1, a dystonia gene (although this can be so severe it is misdiagnosed as spastic paraplegia <https://www.sciencedirect.com/science/article/pii/S1353802023000330?via=ihub>), and genes which to my knowledge are only pathogenic in a recessive way (biallelic). These include SPG11, which cause complicated HSP, arguably similar to ALS, and PINK1 and PLA2G6, which cause parkinsonism, and would struggle to assume they are relevant. One more gene, UCH-L1, is presumably included because of a single PD report a long time ago.

We thank the reviewer for this comment and apologize for any confusion. Fig. 3 represents a landscape of somatic variants in germline-free cases, which does not claim these variants as disease-relevant. We have added the following sentence to explain this: “We first focused on somatic mutations in all the targeted neurodegeneration genes.” In the revised manuscript, we have performed separate regional enrichment analyses for somatic variants in ALS/FTD genes alone, as well as across all targeted neurodegeneration genes.

As the authors reasonably aimed to capture a wide range of genes, it becomes crucial to disentangle which might be relevant to the actual disease presentation. It may not always be clear, but I do not feel it is correct to assume that all these are relevant. If anything, some of the examples above I could see as "control genes"- i.e., genes known

to be involved in neurodegeneration (are all such genes prone to somatic mutations?), but likely NOT relevant to these cases. At the very least, I believe figures and % scores need to be separated into "high" and "low" chance of relevance- where "low" would include those involved in other diseases AND/OR known to be pathogenic only when biallelic.

We appreciate the reviewer's comment regarding the need to disentangle the target genes based on their disease relevance. We have revised our analyses to stratify the target genes accordingly. For Fig. 3c, we categorized the genes into four groups—ALS/FTD-related dominant genes, ALS/FTD-related recessive genes, Tau proteinopathy-related genes, and other neurodegeneration-associated genes—as suggested by the reviewer. For the enrichment analysis across brain regions, we analyzed somatic variants separately for ALS/FTD-related genes and for all targeted neurodegenerative genes. The results are now presented in the main and extended data figures, respectively.

As we mentioned above, we did not include cases with variants in non-ALS/FTD genes or those with heterozygous variants in recessive genes for the % scores reported in the manuscript. We have added clarification in the main text to avoid any misunderstanding.

After mutation detection, nuclear sorting is used in a very welcome way, to resolve underlying cell type. It is interesting that most mutation levels are higher in non-neuronal. As the authors have also used Sox10 sorting to enrich for oligodendrocytes, it would have been nice to see this here, but perhaps this can be deferred. The most puzzling is the question of hypodiploid nuclei (I presume not resolved by cell type). Apoptosis is a plausible explanation, but the wide discrepancy between replicates is worrying. Is there an explanation? Were the high VAF in different mutations found in the same sorts? Are there only two replicates in the APP mutation analysis?

We thank the reviewer for raising this concern. For each variant and each brain region, four replicates of each cell type were sorted. We think the observed variation is caused by the small number of cells in each replicate. We sorted 500 cells for each replicate followed by PTA-based whole-genome amplification. Detection of variants at 1-2% VAFs in 500 cells can give big variation due to slightly more or less mutant nuclei in each sorted population. The whole-genome amplification might also cause allelic imbalance and variations of the mutant alleles. We have moved these results from Fig. 5 to Extended Data Fig. 7 and toned down our claims regarding the cell types.

The authors are very wise to carefully screen for contamination, which could result in a germline SNP in one sample appearing as a mosaic variant in another one or more. The

strategy of excluding any variant "present in less than 0.1% of the population in any ethnic group of 827 public databases" could even be seen as too stringent, if one accepts that a somatic SNV could by chance be identical to a germline SNP. Is it possible to report the number of variants that were excluded this way, and select a small number for brief discussion - e.g. did samples carrying these also have many other apparent contaminant SNPs at similar AF? If not, are they plausible somatic candidates? On the contrary, accepting only those "observed in < 5 different individuals" could be seen as too lenient. Were there any that were validated, but seen in more than one individual? Would then these be somatic mutation hotspots?

We apologize for the insufficient description regarding the contamination issue in our samples. As the reviewer pointed out, a somatic SNV could occur at a known SNP site. Therefore, we checked contamination by comparing the somatic candidate sets against the germline variant sets of other individuals, and identified potential contamination if a sample had ≥ 40 low-*VAF* somatic candidates that matched germline variants in another specific individual. Using this method, we identified potential contamination in 29 out of 1,787 samples (13 ALS, 10 FTD, and 6 control samples), which represents 1.6% of the total sample pool. Among these, 23 samples did not have any somatic variant candidates, five had one candidate, and one had two candidates. We conducted a thorough analysis to ensure that these identified somatic variants were not artifacts of contamination. This included checking for discordance between the observed *VAF* of the somatic candidates and the estimated contamination rate, as well as verifying the absence of corresponding variants within the contaminant sources. This analysis confirmed that the remaining candidates were independent and not contamination artifacts.

We also agree with the reviewer's concern that accepting variants observed in <5 different individuals might be too lenient. During this revision, we conducted validation sequencing for all identified variants. We found that no variant was validated if it was observed in more than two individuals. We therefore updated our criteria to only consider variants observed in no more than two individuals.

In the succinct but excellent discussion, the authors very reasonably propose (l.374) that "degeneration may spread from a site containing mutant cells to eventually cause loss of neurons in regions that do not carry the mutation". This is indeed a very plausible explanation for sALS, with TDP43 spread. This idea is not novel, e.g. <https://doi.org/10.1016/j.mehy.2011.06.027> 10.1073/pnas.0909343106. The one question not addressed at all here is whether there may be somatic mutation (expansion) of the C9ORF72 tandem repeat, which would not have been detected in the present methods. This needs to be mentioned as a limitation, although I am not aware

of any suggestion that a normal germline repeat size can expand somatically. The only work on this I found is a small negative study <https://doi.org/10.1212/NXG.0000000000000317>. The authors encourage the genetic classification of all ALS, regardless of family history, in line with a very recent relevant guideline to offer testing to all; doi: 10.1002/acn3.51895

We thank the reviewer for bringing these useful references to our attention. We have incorporated them into the Discussion section of the manuscript. We also thank the reviewer for mentioning the possibility of somatic *C9ORF72* repeat expansions in sporadic cases and the limitation of our current approaches. The publication mentioned by the reviewer demonstrated the potential of using reduced peak heights in repeat-primed PCR analysis to detect somatic *C9ORF72* repeat expansions (PMID: 31041398). We indeed found three ALS and FTD cases exhibiting reduced peak heights in repeat-primed PCR analysis, suggesting possible somatic *C9ORF72* repeat expansions. We performed targeted long-read sequencing of *C9ORF72* repeat expansions on these cases using the recently developed PureTarget Repeat Expansion Panel from PacBio. The targeted long-read sequencing identified wild-type alleles and expanded alleles with hundreds to thousands of repeats in these cases. In two cases, the smallest expanded alleles were short pathogenic alleles with 57 and 65 repeats, respectively. The smallest expanded alleles in the third case were intermediate alleles with 21 repeats. However, this approach does not provide accurate estimation of the VAFs for each allele and lacks sufficient coverage of large flanking regions outside the *C9ORF72* repeat expansions for phasing the alleles. We have included these findings and discussed the technical limitations in the manuscript and in Extended Data Fig. 9 and 10.

NOTE:

(1) I am not necessarily expecting the authors to cite all the studies I mention, especially if there are space limitations, but they are very welcome to, if they agree that they further help understand the background and implications.

We have revised the manuscript to include additional references as the reviewer suggested.

(2) As far as I can see, there is no preprint. I would encourage the authors to release such important work as a preprint, rather than after the long delays involved in publishing, to benefit the community.

We thank the reviewer for this suggestion. We have uploaded our original version of the manuscript to bioRxiv (<https://www.biorxiv.org/content/10.1101/2023.11.30.569436v1>) soon after initial submission.

Minor points

Could I ask about the 5 mutations not confirmed by amplicon sequencing - any pattern that could help further curation of low level somatic mutations?

During this revision, we performed additional amplicon sequencing on all remaining somatic candidates and identified additional false positive candidates. The false positives could be grouped into two categories: 1) germline variants and 2) low-*VAF* false positives. The first category was mainly observed in candidates with high *VAF* ($\geq 5\%$), which we suspect is caused by potential biases related to amplicon design or probe hybridization, leading to large deviation from the actual 50% *VAF* for germline variants. The second category primarily consists of candidates with *VAF* $< 3\%$, with about half of them being C>T (G>A) substitutions. This type of substitution is the most frequent both in true variants due to cytosine deamination and as an artifact from amplification errors (C>T transition). We couldn't find any other patterns discriminating against false positives. The complete results from the validation sequencing are now presented in Supplementary Table 6.

I note whole genome amplification by PTA is mentioned in methods. I am aware of its successful use by the authors for single cell WGS. I am not sure I understand what happened here - were the 500 nuclei of each type per mutation / experiment sorted in bulk, but then each pool of 500 was then amplified by PTA prior to amplicon sequencing?

The reviewer's understanding is correct.

Please ensure that " μ " and not "u" is used in methods.

We thank the reviewer for pointing this out. We have revised the manuscript accordingly.

Reviewer #3:

Remarks to the Author:

The authors studied a list of 88 genes using a targeted approach (single-molecule molecular inversion probes) followed by sequencing with an average depth of coverage

of 1800x in multiple CNS DNA samples from patients with sporadic Amyotrophic Lateral Sclerosis (ALS), sporadic Fronto-Temporal Dementia (FTD) (n = 404 patients) and 144 controls.

After detection of likely germline pathogenic variants and secondary exclusion of carriers for further analyses, a significant enrichment in protein-altering variants was observed in ALS/FTD cases as compared to controls, with a specific enrichment in tissues primarily affected by each disease. This result is rather exciting and suggests that at least some of these variants may have contributed to the disease pathophysiology.

Then, the authors attempted to replicate these results in an ALS case - control series (RNAseq data) albeit with only 23 controls.

Overall, I found the paper difficult to judge at this stage for three major reasons: variant classification, variant detection/selection, and patient selection/phenotyping i.e. inclusion/exclusion criteria. Please see specific comments below.

Overall, I think that the authors should clarify what is pathogenic and what is predicted deleterious. They should rely on international recommendations, such as the ACMG-AMP recommendations, to classify variants as (likely) pathogenic or not. It seems that supplementary tables split variants that are pathogenic in patient-specific databases from candidate variants that are predicted deleterious in their algorithm, but in the text we can read "known and novel pathogenic variants" as a whole, which I feel is misleading, especially because the criteria for classification briefly reported in the methods do not allow the replication of the classification of the variants (see examples below with Clinvar benign variants classified as pathogenic).

Mixing such concepts (pathogenicity / in silico predictions) leads to a misevaluation of the rate of germline pathogenic variants, first, and, most importantly, this makes the main message of the paper, i.e. the contribution of somatic variants, questionable.

Interpretation using ACMG-AMP guidelines by a geneticist expert in neurodegenerative diseases should be provided, for (likely) pathogenic variants. This does not preclude a secondary analysis with a burden test allowing the search for an enrichment in coding, predicted deleterious variants, but such a study should rely on homogeneous/unbiased criteria for variant selection.

Doing clinical classification of variants would avoid, for example, seeing a nonsense variant in an exon that is not expressed in the adult brain in the MAPT gene to be rated

as probably pathogenic (a gene where the haploinsufficiency mechanism is not causal of FTD, in addition) – although it is in a control sample (and the variant in “benign” in Clinvar), or a missense APP variant outside any functionally relevant domain being rated as probably pathogenic as well.

Many of the variants discussed in the paper are actually variants of uncertain significance and it is unclear how they were selected, given (i) a questioning list of genes, (ii) unclear methods for variants selection.

One of the other main weaknesses is that the authors do not mention the underlying pathology / precise diagnosis in most cases. This study is based on brain/spine banks and I am surprised to read that precise neuropathological diagnoses are not available. Having the subtype of FTD, for example Tau-related, TDP-43 or FUS, seems very important (at least ubiquitin +/-), as well as co-pathologies (Alzheimer's disease). This would avoid seeing a PSEN1 pathogenic variant carrier rated as “FTD” while it is much more probable that the patient actually suffers from frontal variant of AD, I assume. The information on the underlying pathology and copathology seems mandatory to me, for the interpretation of such cases.

Specific queries and comments

1. Comments on sequencing and detection of variants

1.1. What were the criteria for selecting somatic candidate variants for amplicon-based validation? It is unclear how we start from 167 candidate variants to a selection of 41 variants for confirmation, 87.8% of which are true positives, and then 95 unique variants were analyzed (how many are true positives?). Where do the 95 variants come from? I count 105 unique variants in supplementary table 5. What are the “neurodegeneration-related genes”, are these the 88 genes or a selection of it? This is a very important point as it could introduce a bias in all the subsequent analyses and main conclusions of the paper. As some of the variants were detected in only one sample from a specific region, orthogonal validation is required for all variants in the statistical test.

We apologize for the confusion regarding the reported number of somatic candidates. Initially, we identified 167 somatic candidates, including recurrent ones across multiple tissue regions within the same individuals (105 unique candidates). Of these, 133 candidates (95 unique candidates) were from germline-free cases, which were used in the somatic burden analysis. The term “neurodegeneration-related genes” refers to all 88 target genes included in our panel, and the candidate counts were from all these target regions. To improve clarity, we have revised the manuscript to report only the number of unique somatic variants. We have also updated Supplementary Table 5 by adding an index to represent each unique candidate and merging the same variants

from the same individual, which helps to recognize variants observed multiple times across tissues.

For validation sequencing, we previously targeted all exonic variants and randomly selected some non-exonic variants, resulting in a total of 41 variants chosen for validation. Among them, 36 variants were validated, as reported in Supplementary Table 6. We agree with the reviewer that validation should be performed for all variants included in the statistical tests. Therefore, during this revision, we conducted additional amplicon sequencing for the rest of identified variants and updated the results in Supplementary Table 6. We found additional false positives particularly among variants with a high VAF ($\geq 5\%$), many of which were confirmed as germline variants. After excluding all false positives, we finally confirmed 64 unique validated variants. All statistical tests were conducted using only these validated candidates. We have updated the manuscript under the section "Identification of somatic SNVs and indels from MIP sequencing data" to include detailed descriptions of the validation sequencing and its results. Please note that the candidate counts have been changed following these validation experiments and minor changes in variant-calling criteria made during this revision.

We performed droplet digital PCR (ddPCR) as an orthogonal validation for 34 of the 41 variants initially validated by amplicon sequencing. ddPCR probes could not be designed for the remaining 7 variants. We validated 29 of the 34 (85.3%) candidates, along with two confirmed germline variants and three negatives. For the rest of the somatic variants identified in the MIP sequencing data, we performed additional amplicon sequencing to validate them. Overall, we were able to validate 65.3% of all detected variants with amplicon sequencing. The slightly reduced validation rate was due to germline variants, which appeared as somatic variants with high VAFs. This issue was already mentioned by the reviewer in the next comment. We therefore only included validated variants in the subsequent analyses. We regret that we could not perform ddPCR as an orthogonal validation for all the identified somatic variants, as the experiments are very costly.

1.2. Regarding the detection of germline variants, in my own experience of smMIPs, the GATK haplotype caller performed quite poorly. I understood that it was likely due to the design of MIPs/amplicon-like reads, so that variants are all located in the same position of a given MIP-based read and this introduces a bias during variant calling (specifically for variants at the extremity of reads). Could the authors comment on this and double check or validate the presence of germline variants?

We appreciate the reviewer's thoughtful comment, and we agree that MIP probes can introduce biases potentially leading to discrepancies in VAFs, as observed in our validation sequencing. However, for known/predicted deleterious germline variants, we confirmed that all identified variants were consistently observed across all tissue samples from the same individual. Additionally, we tried our best to cover the target regions with more than one MIP probe. We believe these efforts have reduced the likelihood of these germline variants being artifacts.

1.3. Are graphs and numbers on read depth presented after or before deduplication (I guess, after?) Was the sequencing performed in paired-end mode and with which length of reads (F+R reading of same molecules or not?) and what were the rates of duplicates and initial sequencing depth (if results are presented after dedup)?

All the read-depth information presented in our manuscript reflects the numbers after deduplication, as the reviewer expected. The sequencing was performed in paired-end mode, with a read length of 150 bp. Each read in a pair sequenced opposite strands of the same DNA molecule.

The average read depth was 3,525.1X before deduplication and 1,795.7X after deduplication, representing an average of 1.96 reads per UMI. Given our relatively low read-per-UMI ratio, we implemented an additional criterion that required two different sets of variant-supporting MIP amplicons to call somatic variant candidates. This criterion proved to be one of the most effective in reducing false positives during benchmarking. We have additionally included a figure illustrating the read depth before and after UMI deduplication in Extended Data Fig. 1.

1.4. Small detail, I would not qualify as "ultra-deep", sequencing in the order of magnitude of 1000-3000x, but rather for >10,000, although there is no definition of what is "ultra" deep or not. I would not call it ultrasensitive either, as the validation using spike-in shows a maximum of 75% sensitivity for 5-10% mosaics, which is surprisingly low.

We agree with the reviewer that the term "ultra-deep" may not be appropriate given the current read depth. We have revised the term from "ultra-deep sequencing" to "deep sequencing" in the manuscript.

We also apologize for the insufficient description regarding the benchmarking sensitivity, particularly for the 5-10% VAF range. As we described above, one of the key criteria for somatic calling was requiring two distinct types of variant-supporting amplicons, which effectively eliminated many low-VAF false positives in our benchmarking. However, due

to the limitation of panel design, certain regions could not be covered by more than one amplicon. Many spike-in variants were located within such regions and, as a result, were not called as somatic candidates based on our filtering criteria, leading to a lower sensitivity even for targeting 10% AF. Initially, we included these variants in benchmarking results to be conservative, but we found that this may mislead readers to have an impression that our variant calling pipeline has low sensitivity. To address this, we have revised the benchmarking process by excluding spike-in variants if 1) a given site could be covered by only one amplicon, or 2) had a read depth of less than 100, which are inherent limitations of the panel design rather than the variant calling process. This adjustment improved the sensitivity to 96% for 10% AF. We have updated Extended Data Fig. 3b and the manuscript to reflect these changes. We did not use the term “ultrasensitive” in our manuscript, so no changes were made in this regard.

1.5. This is not clear if the tools used for somatic variant detection calculate the average error rate (noise) per position in the sequencing run and then the probability for a candidate variant to significantly deviate from the average noise after multiple testing. Could at least the authors add some information on the principle of variant calling in the methods section using the three tools?

We apologize for the insufficient description regarding the somatic variant callers used in our study. RePlow is optimized for detecting low-VAF somatic variants from deep targeted sequencing data. It generates a profile of background errors per substitution type for a given sample and utilizes these distributions as priors in a Bayesian model to estimate the probability of a variant candidate to be an error, similar to the approach mentioned by the reviewer. MuTect2 is one of the most widely used somatic variant callers, particularly sensitive to detecting low-VAF variants in impure and heterogeneous samples. MuTect2 employs a Bayesian classifier to evaluate the likelihood of a variant being genuine versus a sequencing error, and applies multiple filters to reduce false positives. Pisces is specifically designed for detecting somatic variants from amplicon sequencing data, particularly in cases where no matched control sample is available. This tool stitches paired-end reads into consensus reads and recalibrates variant quality scores specifically tuned to address amplification-related errors such as thermal damage or deamination, often observed in FFPE samples. We have added these descriptions to the “Somatic variant calling from MIP sequencing data” section of the Methods to provide more detailed information.

1.6. The PPV seems very high (extended data fig 3b) but still only 87.8% of selected somatic variants were confirmed after amplicon-based sequencing, could the authors comment on that?

For benchmarking, we generated spike-in data using commercial gDNA from two Coriell cell lines (GM12787 and GM24695), which are widely used as reference materials in many studies. These commercial samples undergo strict quality control, providing us with excellent benchmarking results. However, our main sequencing data were generated from fresh frozen tissue samples, which can contain various artifacts arising during sample handling and library preparation processes. The high quality of spike-in data does not fully account for these potential background errors, which we believe contributed to the discrepancy between the high PPV in benchmarking and the validation experiment results.

1.7. It seems useful to compute the theoretical minimal VAF that is detectable using the sequencing method (N UMIs, depth) and the criteria used for the detection of mosaics, i.e. 15 supporting reads, to better figure out the power of the study in terms of VAF resolution.

We appreciate the reviewer's insightful comment on assessing the power of our mutation detection method. To address this, we simulated 100 datasets for each combination of VAF level and sequencing depth, covering a total of five different VAF levels (0.5%, 1%, 2.5%, 5%, 10%) and 50 different sequencing depths (ranging from 100 to 5000). The number of variant-supporting reads was modeled using a binomial distribution across these sequencing depths, and the corresponding sensitivities were calculated based on our current calling criteria (≥ 15 variant-supporting reads, $0.001 \leq \text{VAF} \leq 0.4$). The theoretical sensitivity across these conditions is depicted in the figure below and now added in Extended Data Fig. 3. Please note that this simulation does not account for the multiple filtering steps we additionally employ to reduce false positives, which leads to an overestimation of sensitivity compared to real data.

1.8. Why selecting variant candidates if observed only in disease or control groups but not in both?

Somatic variant calling targeting low-VAF variants inevitably involves a large number of false positives (FPs), so our top priority was to minimize these FPs. We assumed that variants observed in both disease and control samples were more likely to be FPs, and even if they were true variants, they would likely have minimal impact on the disease. Therefore, excluding these variants from the candidate set was worth doing to maintain the overall quality of our call sets. Our recent validation during this revision further supported this approach, as no variants were validated if they were observed in more than two individuals.

1.9. Was there any additional samples with DNA contamination than the “few ones” cited? How many were identified overall? (“few” is not very accurate)

We apologize for the insufficient description regarding the contamination issue in our samples. We checked and confirmed contaminated samples and their contaminant sources by comparing the somatic candidate sets against the germline variant sets of other individuals, as described in the Methods section. Among the 1,787 samples in our

study, we identified potential contamination in 29 samples (13 ALS, 10 FTD, and 6 control samples), which represents 1.6% of the total sample pool. Of these, 23 samples did not have any somatic variant candidates, five had one candidate, and one had two candidates. We conducted a thorough analysis to ensure that these identified somatic variants were not artifacts of contamination. We checked the discordance between the observed VAF of these somatic candidates and the estimated rate of contamination, as well as verified the absence of corresponding variants within the contaminant sources, confirming they are independent candidates. We have revised the manuscript to include detailed descriptions of the contamination issue in the “Identification of somatic SNVs and indels from MIP sequencing data” section.

2. I could not find the information on the underlying pathology of these cases. They all come from neuropath biobanks, why the TDP-43, MAPT, FUS pathology would not be available? At least ubiquitin + or- information should be available?

The information on co-pathology with Alzheimer's would also be very useful, as well as Braak&Braak stages. For example, there is a pathogenic germline PSEN1 variant. The patient must have Alzheimer's disease, if the variant is actually pathogenic. I guess that the patient may have frontal variant of AD rather than FTD. The information on neuropath should solve this important issue, and if that is the case, the patient should be excluded from the screened cases as well as all patients with a final diagnosis of Alzheimer's disease (whatever the clinical presentation upon presentation). Interpretation of additional cases may require a precise diagnosis.

We thank the reviewer for bringing up this important question. We have acquired pathological information for all the cases included in this study and summarized details on TDP-43 and Tau pathology in the updated Supplemental Table 1. Some cases predated the discovery of TDP-43 pathology, so this information is not available for those cases. While we provided Braak & Braak stages for a large portion of the cases, many still lacked this information. The specific case with the *PSEN1* variant exhibited only mild Tau pathology, with a Braak & Braak stage of 2, which is normal for the patient's age. Therefore, this case was not classified as an AD case. In the revised manuscript, the *PSEN1* variant was classified as VUS by ACMG guidelines. Therefore, we have removed this variant from Fig. 2.

3. It is not clear why patients with unclear family history were selected if age at death above 45 years?

We included cases without a clear family history to increase the sample size of our study. We relied on the identification of pathogenic and predicted deleterious germline

variants in the MIP sequencing data to exclude both familial cases without a clear family history and sporadic cases with *de novo* germline variants. In addition, we selected cases above the age of 45 to avoid juvenile ALS cases.

4. The list of genes is rather original and not justified. For example, looking at Alzheimer's disease genes, I see APP, PSEN1, PSEN2 genes which are the only three known autosomal dominant genes. Then, I also see MARK4; I could find only one de novo germline mutation in a single case in the literature in a sporadic AD patient, and MARK2 which has never been associated with AD. In addition, ABCA7 is mentioned along with APOE, which both are risk factor genes and are rated as "autosomal dominant", while the stronger risk factor gene SORL1 – sometimes considered as 4th autosomal dominant gene – is not listed, etc. I understand that some other genes are here as part of differential diagnoses (e.g., SPG4/SPAST). I think that the list should be justified in the methods. Maybe even removing non-relevant genes from the somatic mutation analysis, which should focus on genes that are relevant for the given pathology (e.g. MAPT for Tau-FTD, etc.) as in the second analysis of the somatic variants results. I believe that the global burden does not make much sense given the list of 88 genes.

We thank the reviewer for this comment. Our list of AD genes was based on a previous study (PMID: 24951455), which summarizes the genetic risk factors for AD. We further excluded genes originally identified through GWAS studies, during which SORL1 was unfortunately removed. Our panel design was intentionally inclusive, as we aimed to evaluate somatic mutations in genes not previously linked to ALS/FTD. This hypothesis was supported by the enrichment of somatic mutations in non-ALS/FTD genes identified in bulk RNA-seq data. In addition, potential AD risk-factor genes are often included in panel sequencing studies for AD, including the specific MIP panel sequencing study on AD cases mentioned by the reviewer in a later comment (PMID: 30114415). For the enrichment analysis across brain regions, we analyzed somatic variants separately for ALS/FTD-related genes and for all targeted neurodegenerative genes during this revision. The results are now presented in the main and extended data figures, respectively.

5. HGVS nomenclature should be used throughout the text and supp tables (e.g., NM_000454:exon1:c.C14T:p.A5V is not an accepted nomenclature)

We appreciate the reviewer's thoughtful comment. We have added a column for HGVS annotations in all the relevant supplementary tables.

6. Comments on variant interpretation

6.1. In which database the SOD1:NM_000454:exon4:c.G304A:p.D102N variant is considered as pathogenic? I could not find it in Clinvar and it is in supp table 3 as previously reported to be pathogenic. I guess it is in an ALS-specific database, no doubt it is pathogenic (another missense change at this position is pathogenic in Clinvar), but it would be helpful to have the reference. In the main text, “multiple clinical databases” are mentioned but I could not find them in the methods.

6.2. Variant VAPB:NM004738:exon5:c.G510A:p.M170I is benign in Clinvar (2, multiple submitters) but still in the reported pathogenic germline variants in Table S3.*

We thank the reviewer for pointing this out. In the revised manuscript, these two variants are now classified as benign based on the ACMG guidelines. We have updated Supplementary Table 3 to reflect the changes.

6.3. The emphasis on the APP somatic variant is very difficult to understand. The rationale of including APP in a FTD/ALS based study is very questioning as there is no evidence that APP derivatives (including Abeta peptides) may play a role in the pathophysiology of FTD/ALS. If the gene was included as part of a differential diagnosis (which is very questioning based on the fact that authors used a neuropath-based cohort, i.e. the gold standard for diagnostics, thus not requiring genetics for the classification of disease), then the only variants that may cause Alzheimer's disease should be interpreted. They all map to exons 16 and 17 of NM_000484.4 in AD. Here, the missense variant with emphasis in the main text is in exon 7. I am not aware of any hypothesis on how such a variant could have a potential effect on alpha/beta/gamma secretase cleavage of APP or aggregation/neurodegeneration. These kinds of variants of uncertain significance can be found in many controls or patients and do not justify a specific attention in clinical practice. This raises more questions about the interpretation of the distribution of the variant, as the authors seem to put some weight on this latter argument, but for this specific variant, it is clearly not relevant. In addition, in the main text, the authors seem to have included Alzheimer's disease genes, not only as part as differential diagnoses, but also as putative providers of Tau pathology. This is again very questioning given the inclusion criteria: was Alzheimer's disease excluded based on neuropath information?

We thank the reviewer for the insightful comment. We indeed included AD genes as potential providers of Tau pathology. We have collected TDP-43 and Tau pathology information of cases included in this study, which is now incorporated into Supplementary Table 1. We agree with the reviewer that the somatic *APP* variant may not be related to the disease pathology and removed this variant from Fig. 5 and Supplementary Table 7.

6.4. In the methods section, it is not clear how pathogenic variants were identified. It seems that benign variants from Clinvar/HGMD were excluded (but from my previous comment, we can see that it is not the case) and the “pathogenic prediction module” was used, but it does not mention Clinvar/other patient-specific databases classification as pathogenic. However, the paper is mainly based on the classification as pathogenic (already reported) vs predicted. This must be clarified.

“missense variant was selected to be pathogenic if at least three different algorithms predicted damaging effects (deleterious for SIFT, LRT, FATHMM, PROVEAN, MetaSVM and MetaLR; probably damaging for PolyPhen2; disease_causing for MutationTaster)”: pathogenicity cannot only rely on in silico predictions, even more with so lenient criteria ($\geq 3/8$ tools). They only predict possible deleteriousness on protein products. Pathogenicity likelihood should be rated following international recommendations (ACMG-AMP) and then remaining variants remain missense variants that are predicted deleterious or not (most of them being variants of uncertain significance based on ACMG-AMP recommendations)

It is not acceptable to qualify as “pathogenic” variants only because three bioinformatics tools predict an effect on the protein.

The interpretation of the paper mainly relies on the fact that so-called predicted pathogenic variants are identified as a mosaic. Looking at Table S7 (titled “pathogenic somatic mutations”), I found the above-mentioned of APP, which is everything but pathogenic, I also see a variant of uncertain significance in TARDBP and variants in other genes. I did not re-interpret every variant but I have the feeling that most – if not all - are actually variants of uncertain significance.

We thank the reviewer for this insightful comment. In the revised manuscript, we have classified these somatic variants based on the ACMG guidelines. All of these variants are now classified as VUS. The somatic APP variant is now classified as benign and removed from Fig. 5 and Supplementary Table 7. We have modified Supplementary Table 7 and the manuscript accordingly.

6.5. Among the germline variants found in controls, how a nonsense MAPT variant can be considered as predicted pathogenic? Haploinsufficiency of MAPT does not cause FTD, this variant is not present in a transcript expressed in adult brain, and it is already known as benign in Clinvar (table S4)

We appreciate the reviewer for pointing this out. In the revised manuscript, this variant is now classified as likely benign following the ACMG guidelines. We have updated Supplemental Table 4 and the associated text in the manuscript accordingly.

7. Minor comment, not sure that “oligogenic” well applies to the carrier status for multiple pathogenic variants in autosomal dominant genes.

We understand the concern that oligogenic inheritance typically describes a trait influenced by a few genes, while pathogenic variants in autosomal dominant genes are usually sufficient to cause the disease on their own. However, recent studies have demonstrated the incomplete penetrance of variants in dominant ALS/FTD genes, including the *C9ORF72* repeat expansion (PMID: 35620137). Furthermore, the co-occurrence of multiple variants in autosomal dominant ALS/FTD genes in ALS cases has been reported in several recent studies (PMID: 22645277, 25182743, doi: 10.1101/2024.03.21.24304693). In these studies, this phenomenon has been referred to as the “oligogenic inheritance” of ALS. We hope this explanation clarifies our usage of the term "oligogenic" in the context of our study.

8. Did the author phase ALS2 variants in the patient with a germline heterozygous variant and a somatic variant?

We thank the reviewer for this comment. The somatic *ALS2* variant was located 16.6 kb away from the nearest germline heterozygous SNP, making it challenging to phase them even with long-read sequencing. Due to the low VAF of the *ALS2* variant (~1%), a high sequencing depth was mandatory to capture reads harboring the variant and therefore conventional long-read WGS was not feasible for this purpose. We reached out to PacBio for their CRISPR-Cas9-based targeted long-read sequencing service. Unfortunately, this service was discontinued by PacBio earlier this year.

9. How was performed the SMN1 analysis by short-read whole genome sequencing? Is that really reliable, given the difficulty in differentiating SMN1 from SMN2 in short-read data?

To address this concern, we performed additional long-read WGS for these two cases. We did not identify any pathogenic germline variants in *SMN1* and *SMN2* in these two cases. We have summarized the germline variants from the long-read WGS data in Supplementary Table 11.

10. I did not find very convincing the claims regarding enrichment of somatic variants in

hypodiploid cells given low numbers, there are controls or replicates missing to make this result significant (too few variants studied).

We thank the reviewer for raising this concern. We have toned down our claims and moved these results to Extended Data Fig 7.

11. In the RNA-seq based analysis, were the C9ORF72 expansion carriers excluded from the case-control analysis (it seems that there are a few based on a supp table)? The methods for this part of the analysis are not described in detail. How was the case-control analysis performed? Were all variants included? Any threshold on VAF or coding/non-coding status? I am skeptical that using only 23 controls would be enough to reach significance.

In our case-control analysis, we excluded individuals with known germline pathogenic variants, including the C9ORF72 expansion. We considered all exonic calls that passed through the RNA-MosaicHunter pipeline (as detailed in “Somatic variant calling from RNA-seq data” in the Methods section) without further filtration on VAF. Utilizing a linear regression model, we tested whether the count of all variants or deleterious variants per sample is significantly associated with clinical condition (ALS or control), while controlling for potential confounding factors such as age, gender, sequencing depth, and PMI. Although our RNA-seq analysis comprised only 108 ALS and 23 control individuals, we typically have multiple tissue samples per individual. This resulted in a total of 539 samples for ALS and 107 for control, providing sufficient statistical power for burden analysis. We have provided additional details about our statistical analysis in the “Burden analysis of somatic mutations using linear mixed model” section of the Methods.

12. Did the patient with the pathogenic somatic DYNC1H1 present any neurodevelopmental phenotype? How was it assessed before stating that the patient does not show the classical phenotype? Some adults can survive with this disease.

We thank the reviewer for this comment. We reviewed the clinical record of this individual and didn't find indications of cortical developmental malformations. However, we cannot rule the presence of subtle changes due to the lack of MRI images.

13. The pathogenic variant in LMNA (H566Y) is not reported in the Clinvar database, what are the arguments for pathogenicity?

In the revised manuscript, this variant is now classified as a VUS based on ACMG guidelines. The specific variant has not been reported before, but a variant at the same

locus, *LMNA* (H566D) has been found in a patient with type 1 Charcot-Marie-Tooth disease. It is predicted to be deleterious by our *in silico* prediction. We have clarified this in the main text.

14. If the results are eventually replicated in other datasets and if at least part of these somatic variants are confirmed pathogenic, that would make the pathophysiology of FTD/ALS different from that of AD from that prion-like hypothesis perspective. Indeed, a similar work has already been performed in AD using a similar approach (MIPs), although with a relatively lower number of brain tissues (around 100), but no pathogenic variant was identified. Could the authors comment on why such conditions with prion-like mechanism would differentiate from that perspective? That is also another reason to make sure that there are/ there are no Alzheimer's disease cases in the current series.

We thank the reviewer for bringing up the comparison of our study to a similar study on AD (PMID: 30114415). In that earlier study, the authors performed MIP panel sequencing on 3 autosomal dominant AD genes (*APP*, *PSEN1*, and *PSEN2*) as well as 8 other AD risk-factor genes in a collection of blood (355 samples) and brain tissue samples (100 samples) from sporadic AD and control cases. Only one predicted damaging somatic variant in *SORL1* was identified in the blood of an AD individual. The authors concluded that pathogenic somatic variants are rare in AD. However, they also noted in the discussion that their study was limited by not including multiple brain regions. Including multiple brain regions is a major strength of our study, as it demonstrates that the identified somatic variants tend to be focal and enriched in disease-related brain regions. The aforementioned study only included four tissue samples from the hippocampus, a region often involved in the onset of AD, which limited its ability to identify disease-related somatic variants.

Summary of revisions and major responses

In this revision, we have performed new computational and experimental analyses that directly address the remaining reviewer concerns. Below are key updates:

1. Functional role of the TARDBP variant:

As suggested by Reviewer 1, we performed phospho-TDP-43 (pTDP-43) immunohistochemistry on multiple brain regions from the TARDBP (p.L248F) case. We found the highest pTDP-43 levels in the motor cortex (where the variant resides), with levels decreasing with distance—supporting a pathogenic role for this variant.

2. Statistical approach:

Per Reviewer 3's suggestion, we reanalyzed our data using individual-wise statistical tests. These results were consistent with our prior findings, as most somatic variants were present in only one tissue sample per individual.

3. Specificity of somatic variant enrichment:

a) We extended enrichment analyses to include intronic, non-coding, and non-functional variants. Only exonic and protein-altering variants were enriched in ALS and FTD brains; intronic and non-coding variants were not. The increased ratio of non-synonymous to non-coding variants further supports the specificity of protein-altering variants in disease.

b) Individual-wise analysis of bulk RNA-seq data no longer showed enrichment in ALS, highlighting the specificity of our targeted panel findings.

c) We clarified that the identified somatic variants are clonal events, present in both postmitotic neurons and mitotic glial cells, and are likely to have originated during development rather than arising as a consequence of neurodegeneration.

4. New discovery of *de novo* somatic *C9ORF72* expansion:

We identified *de novo* somatic *C9ORF72* repeat expansions in an FTD case—an event not previously reported. Although present technology—both in our hands and those of PacBio, with whom we worked during the revision process—does not allow successful analysis of potential *C9ORF72* mosaic expansions in all postmortem cases—this example case further supports the contribution of somatic mutations to sporadic ALS and FTD.

We believe these new results and clarifications fully address the reviewers' main concerns and further strengthen the manuscript.

Reviewers' Comments:

Reviewer #1 (Remarks to the Author):

I appreciate the efforts that the authors have put in to clarifying the aspects of the initial manuscript that were unclear. The clarifications in variant validation techniques and analysis procedures have made this manuscript easier to understand. I do have a few concerns with the revised manuscript.

The somatic variant in TARDBP (p.L248F) was validated by two orthogonal sequencing methods, supporting that the variant is indeed real and confined to a narrow region of the cortex. However, as this variant is novel and therefore of unknown consequence in ALS, further experimentation is required to claim that this variant is detrimental. In a study of this magnitude, it would be beneficial to see that any of the reported somatic variants result in observable cellular phenotypes. There remains the possibility that the variants observed in this study are those that are tolerated by cells, as acquired somatic variants that result in strong degenerative phenotypes may have selected out in cells carrying these variants and would therefore not be observed. The TARDBP p.L248F variant is located in the previously annotated Nuclear Export Sequence (NES); however, there are reports disputing the necessity or ability of this coding region to instigate TDP-43 nuclear export (<https://doi.org/10.1038/s41598-018-25008-4>, <https://doi.org/10.1371/journal.pbio.3002527>, and <https://doi.org/10.1038/s41598-018-22858-w>). In vitro experimentation to show that this variant either affects export, splicing, TDP-43 aggregation, or some other perturbation would be beneficial to support the potential pathogenicity of this variant. Or, if the claim is that this localized variant results in pathology of nearby brain regions, histological experimentation could show that spread of TDP-43 aggregates begins in this region and lessens over distance from the variant-containing sample.

→ We appreciate the reviewer's suggestion regarding the functional impact of the TARDBP p.L248F variant. The in vitro experiments proposed by the reviewer would require a longer timeframe than is feasible for the current rebuttal. However, following the reviewer's other suggestion, we have performed immunohistochemistry staining for phospho-TDP-43 (pTDP-43) on tissue sections from multiple brain regions in this case. The motor cortex exhibited higher levels of pTDP-43 compared to other brain regions

Figure 1. Quantification of phospho-TDP43 staining in brain tissue sections from the sALS case with the TARDBP p.L248F somatic variant. (a) Bar graph showing pTDP-43 levels across multiple brain regions. Each dot represents the quantification of pTDP-43 in a non-overlapping tissue section. Error bars represent 95% confidence intervals (CI). (b) Regional distribution of mean pTDP-43 levels across brain regions. Motor cortex exhibiting significantly higher pathology compared to the hippocampus ($p = 0.0042$), middle temporal gyrus ($p = 0.040$), and occipital cortex ($p = 0.032$) in post hoc Tukey's HSD tests following one-way ANOVA.

(Figure 1). Moreover, pTDP-43 levels followed a gradient, decreasing with distance from the motor cortex, which aligns with the hypothesis that TDP-43 aggregation spreads from the region harboring the variant. We have included these results in Extended Data Fig. 9 of the manuscript.

The choice to alter the regional variant distribution results is acceptable, but as the main result is stated to be that predicted deleterious somatic variants are enriched in hypodiploid cells, it would be helpful to have an explanation as to what these cells could be. The only proposed answer currently is that they are likely apoptotic cells, and while this may be the case, this suggestion should be cited with previous literature.

→ We thank the reviewer for the suggestion that we support this claim with relevant literature. We have cited a review article (PMID: 1333943), which provides a comprehensive overview of studies showing that reduced cellular DNA content observed by flow cytometry is a hallmark of late-stage apoptosis. Furthermore, we have included a brief explanation in the Discussion to clarify the link between hypodiploid cells and apoptosis.

The use of digital droplet PCR (ddPCR) as an orthogonal validation method is an adequate means to ensure that the variants are real and at the frequency observed. While I am satisfied with the statistical improbability of having amplification errors due to polymerase error rate and UMI inclusion, this also presupposes that there are no unknown sources of error in the protocol. One remaining question on this aspect would be whether the primers used in ddPCR validation are the same as those in MIPS?

→ The ddPCR primers were designed using Bio-Rad's online tool and their exact sequences were not disclosed by the company. However, it is very unlikely that the sequences of ddPCR primers are the same as the MIPS probes, given the differences in design algorithms and intended applications. ddPCR assays prefer shorter amplicons, with amplicon sizes ranging from 60 to 200 bp, while the MIPS probes in our panel target regions between 120 and 250bp, with an average of 209 bp.

In addition, almost all of the identified variants were validated using three different primer sets in amplicon sequencing, except for five variants where only two sets could be designed (Supplementary Table 12). In all cases, variants were validated by at least two primer sets (when three were available) or at least one set (when only two were available). These amplicon-seq PCR primers have an average amplicon size of ~200 bp, also differentiating them from the ddPCR assays. Together, these distinctions reduce the likelihood that the observed variants arose from PCR amplification artifacts.

Reviewer #2 (Remarks to the Author):

The authors have responded extensively to all my comments, minor and major, and provided considerable additional evidence by using ddPCR and also targeted c9orf72 long read seq. This is now an excellent, comprehensive, well-presented and timely body of work. Christos Proukakis

→ We thank the reviewer for the guidance through the review process.

Reviewer #3 (Remarks to the Author):

The authors have revised the paper, which is much clearer. Overall, the manuscript has been improved. However, now that the methods have been clarified, I see major limitations that make the impact of the study more limited than what the authors claim, from my point of view.

1. Interpretation of the results.

Overall, while some of the somatic variants could be pathogenic, as exemplified in the manuscript with interesting examples and candidates (e.g., the TARDBP variant), the claims are too strong as they suggest that somatic mutations are a not-so-rare disease mechanism. From the data in the paper, I would interpret it as a handful of variants have a potential to be possibly pathogenic, but the vast majority of patients do not show any candidate. Thus, the somatic variant hypothesis would explain a very limited number of cases.

→ We would like to clarify that we do not suggest somatic variants as a common mechanism of sporadic ALS and FTD. Rather, we acknowledge in our manuscript that somatic variants potentially explain a small proportion of cases (~2%). Our study highlights the discovery of somatic variants as a previously unexplored etiological factor in sporadic ALS and FTD, providing new insights into their pathogenesis. The finding that a portion of the cases arise from a point source of mutation indicates that disease pathology must spread beyond its point of initiation by non-cell autonomous mechanisms. It is this aspect of the work that provides new insight into disease pathogenesis, even if somatic variants are causative for a small portion of cases.

Figure 2. Enrichment of somatic variants in different genomic regions of germline-free ALS and FTD cases compared to normal controls

Importantly, the so-called enrichment of somatic variants in ALS cases in ALS-affected regions (resp. FTD) might still be largely driven by unspecific, neurodegeneration-driven processes. Pathogenicity/etiology is not proven here, my personal interpretation is that it could remain that patients with neurodegenerative diseases have a higher burden of somatic variants overall, irrespective of the genes, and some arguments are in favor in this study.

→ We would like to clarify that all somatic variants identified in this study are clonal variants. Our cell type analysis of several potentially deleterious somatic variants revealed their presence in both postmitotic neurons and mitotic glial cells. These results strongly support that these variants arose during late brain development rather than during disease progression. In contrast, somatic variants arising during the process of neurodegeneration (which is not a known, but rather a theoretical possibility) in any event cannot be shared by neurons and non-neuronal cells, since neurons do not divide. This definitively

identifies the variants as arising clonally during development and not secondary to some degenerative process acting upon single cells.

To further evaluate the reviewer's concern that the enrichment of somatic variants could be a secondary consequence of disease pathogenesis, leading to an overall increase in somatic variants irrespective of specific genes or genomic regions, we also expanded our analysis to include additional variant categories. Following the reviewer's suggestion, we conducted this analysis using individual-wise tests, with a detailed explanation provided below (See below in Statistics).

Our revised analysis showed a significant enrichment of exonic and protein-altering variants in sporadic ALS and FTD brains, but not for intronic, non-coding, or non-functional variants (i.e., all variants except protein-altering ones (missense, nonsense, and frameshift variants)) (Figure 2). These findings suggest that the observed enrichment is primarily driven by protein-altering variants rather than by random, non-specific variants. We have incorporated these results into Fig. 4a of the manuscript.

Here, only candidate genes were sequenced in the first part of the study. Hence, the claim that they could be pathogenic is too straightforward as there are no comparisons with other genes. The only available comparisons that would favor specificity are the ones of ALS genes versus FTD-AD genes, but the p-values are close to 0.05 and see also further queries that I have on the statistical aspects that might bias the results towards p-values below 0.05 (see below). I would also be curious to know what happens if Alzheimer genes were removed from FTD/Tau genes (as there is no tauopathy without amyloid in pathogenic variant carriers in these genes, these genes are not relevant to this hypothesis).

→ We understand the reviewer's concern. However, targeted panel sequencing is a well-established approach for investigating somatic variants in genes with strong prior evidence of disease involvement. Compared to whole-genome sequencing, this method provides higher sensitivity to detect low-frequency somatic variants. Our gene selection was based on their known relevance to neurodegenerative diseases, including ALS and FTD. The observed enrichment of somatic variants in a disease-specific manner strongly supports their role in disease, especially given that this enrichment was not observed in controls.

To address the reviewer's concerns regarding p-values and the statistical model, we reanalyzed the data using individual-wise tests as the reviewer suggested. This reanalysis confirmed the disease-specific enrichment of somatic variants (see below in Statistics). Regarding the involvement of AD genes in the FTD brain analysis, our intention was to explore the overlap of these two conditions on Tauopathy. However, we acknowledge that this may complicate the interpretation of the results. After excluding Alzheimer's-related genes, the enrichment of somatic variants in FTD genes in sporadic FTD brains was

Figure 3. Enrichment of exonic and protein-altering somatic variants in different categories of genes in ALS and FTD compared to normal controls.

no longer statistically significant (Figure. 3). Nonetheless, FTD brains were enriched for somatic variants in other neurodegeneration-related genes. We have revised Fig. 4b of the manuscript accordingly and

added Extended Data Fig. 7 to reflect these changes. Notably, the enrichment of somatic variants in ALS genes in sporadic ALS brains remains significant, which provides valuable insights into disease etiology.

Consistent with the hypothesis of increased somatic mutations related to and secondary to neurodegeneration, there is an enrichment of somatic variants, irrespective of the genes, in the bulk RNA analysis, supporting a non-causal effect for most variants and further supporting non-specificity to ALS genes.

→ We thank the reviewer for this comment. The bulk RNAseq analysis has sensitivity only to detect somatic variants present in about 10% or cells or more, and thus would not detect at all variants limited to a single cell; therefore variants detected in the bulk RNAseq analysis must be clonally shared. Using individual-wise tests as the reviewer suggested (see below in Statistics), we no longer observe a statistically significant enrichment of somatic variants in the bulk RNAseq analysis (Figure 4). We have revised these results in Extended Data Fig. 11. However, the revised model continues to demonstrate consistent results in the panel sequencing data (see below in Statistics), which has been incorporated into Fig. 4a of the manuscript. Together, these findings suggest that the enrichment of potentially functional somatic variants observed in our panel sequencing data is not a secondary effect of neurodegeneration.

In addition, the enrichment in hypodiploid cells is also in favor of a secondary rather than a primary effect. Indeed, re-entry in cell cycle/apoptosis of post-mitotic neurons during neurodegeneration could result in novel somatic variants in these cells, under oxidative stress/neuroinflammation. This does not indicate that they are causal, most of them could be secondary effects of neurodegeneration, whatever the cause.

→ We would like to clarify that variants included in the cell-type analysis (Extended Data Fig. 10) were shared among multiple cell types, indicating that they arose during development, and had variant allele frequencies >0.53%, indicating that they would be shared by at least 414 cells—based on the 250 ng of input DNA used for panel sequencing library preparation and the assumption of 6.4 pg of DNA per cell. While post-mitotic neurons can, in rare cases, re-enter the cell cycle, they are not expected to differentiate into glial cell types, so that we are on safe ground in concluding that somatic variants shared between neurons and non-neuronal cells cannot have arisen during neurodegeneration, and can only have arisen during development.

In addition, one could assume, as an alternate hypothesis, that the loss of neurons in the regions affected by neurodegeneration results in an increased proportion of glial cells in an inflammatory state, both associated with mitosis and oxidative stress, potentially leading to somatic mutations in any gene (and not only explaining the paradox of having more somatic variants in glial cells, as the authors interpret). The reduction of the amount of neurons could also unmask previously present glial variants that arose during mitoses. This would be consistent with the SBS1 signature and enrichment in glial cells in the cell-type analysis.

→ We would like to reiterate that the variants mentioned by the reviewer are shared between neurons and glial cells. While a neurodegeneration-associated inflammatory state could potentially induce somatic variants in glial cells, these variants would not be expected to be shared with neurons. The presence of these variants in both neurons and glial cells, along with their SBS1 signature, suggest that they originated during development. The fact that our reported somatic variants are also present at VAF's $\geq 0.23\%$, corresponding to sharing by ≥ 182 cells, also rules out this argument, which would require a mechanism not shown to occur in normal or degenerating brain.

The fact that there are not necessarily more variants among non-synonymous vs non-coding could also be an argument of non-specificity/causality. This aspect is not so clear, as the comparisons are mostly exonic versus protein altering, but it is not clear how it compares to the non-coding variants, beyond the few synonymous variants. I acknowledge that it remains difficult to interpret the coding/non-coding variant

ratios, as non-coding regions were not directly targeted, thus reducing the power to perform such an analysis. However, given the rarity of the somatic synonymous variants, it would remain useful to know whether there is any enrichment of predicted deleterious variants compared to all noncoding variants, and not only synonymous ones?

→ We believe that the Figure 2 above clearly demonstrates the specificity of the enrichment of non-synonymous variants compared to other genomic regions, including non-coding regions.

As suggested by the reviewer, we also examined the ratio of non-synonymous to non-coding variants. We first combined all variant candidates from each clinical categories (ALS, FTD, control) and calculated the odds ratios between the disease and control groups (Figure 5a). Both disease groups demonstrated greatly elevated odds ratios (6.6 for ALS vs. Control and 15.4 for FTD vs. Control).

Additionally, we performed an individual-level analysis of the ratios of non-synonymous to non-coding variants (Figure 5b). To avoid the issue of division by zero, we added 0.01 to both the numerator and denominator. The 95% confidence intervals were generated through 1,000 bootstrap resampling iterations. This analysis also demonstrated an increased burden of non-synonymous variants in the disease groups, further supporting our main findings. We have included these results in Extended Data Fig. 6 of the manuscript.

Figure 5. Increased burden of non-synonymous variants compared to non-coding variants in disease groups.

2. Statistics

I understand that there is a tendency of an enrichment of somatic variants in ALS genes in ALS and in FTD genes in FTD, which is an important argument, but p-values remain close to 5% with lower end of CI very close to 1.

However, I am not sure that the statistics translate what the authors meant. From my understanding of the methods, paragraph “Burden analysis of somatic mutations using linear mixed model” and the rebuttal letter, there is no “individual” effect per se, beyond the fixed effects. Is that correct? Is the α_i the PMI fixed effect?

If so, that would mean that a somatic variant found in 5 samples from a single individual would count 5 times? As there are more cases than controls, this would artificially increase N in a biased way towards cases. Given the p-values close to 0.05, it becomes likely that there are no significant results if reasoning at the donor level, if I understood the statistics correctly. It would be more logical to me to compare the number of donors with at least one somatic variant (nonsynonymous vs non-coding and synonymous)

rather than the number of samples. However, there are more chances to find at least one variant in a case with 5 samples than when 2 samples are available, then a correction based on the number of samples should also be applied.

For these reasons, I feel that the impact of the paper remains limited and that the study lacks evidence of pathogenicity/specificity to support their claims.

→ We apologize for the incomplete description. Yes, α_i represents the PMI fixed effect in our model. We also considered sequencing batch as a random effect in the linear mixed model, alongside the fixed effects.

As the reviewer correctly understood, our initial analysis was conducted at the sample level. We initially chose this approach because it allowed us to include a larger number of samples and better account for sample-specific confounding factors, such as sequencing depth. However, we acknowledge the reviewer's concern regarding potential biases and we have now revised the model to an individual-level analysis as follows:

$$y_i = \mu + \alpha_i + \beta_i + \gamma_i + \delta_i + U_i + n_i + \varepsilon_i$$

To perform this analysis, we created individual-wise variant lists by merging all candidate variants from different tissue samples of the same individuals. Any overlapping variants were counted as a single variant to prevent overcounting. In this model:

- y_i is the somatic variant burden of donor i
- μ is the average variant burden in the normal condition
- α_i is the fixed effect of PMI
- β_i is the fixed effect accounting for the disease status (ALS and FTD) of donor i relative to the normal condition
- γ_i is the fixed effect accounting for the sex of donor i
- δ_i is the fixed effect accounting for the average sequencing depth across all samples of donor i
- n_i is the newly added fixed effect accounting for the number of samples from donor i , as suggested by the reviewer.

This change substantially reduced N in statistical tests, from 1,455 germline-free samples (696 ALS, 243 FTD and 516 control samples) to 438 germline-free individuals (216 ALS, 78 FTD and 144 control cases). The inclusion of n_i as a fixed effect also made the revised tests more conservative.

Despite these changes, the individual-level tests consistently demonstrated the same overall results: individuals with ALS and FTD showed significant increases in the variant burden of exonic and protein-altering variants (Figure 2 above), while no significant enrichments were observed for intronic, non-coding, or non-functional variants.

Figure 6. Enrichment of exonic and protein-altering somatic variants in two different groups of disease-related genes (ALS genes and FTD/AD genes) compared to normal controls.

Regarding disease-related genes, we found that ALS patients specifically showed significant enrichment in ALS genes, while FTD patients showed similar enrichment in FTD/AD genes, consistent with our previous results (Figure 6; we included AD genes here to directly compare results using the previous and revised models. We did not include this result (FTD/AD genes) in the revised manuscript based on an earlier reviewer comment). The inclusion of n_i in the model had a significance on variant burden in most tests. However, even after accounting for this effect, the p-values remained at similar levels, despite the reduced N. This suggests that the observed significant enrichments are not artifacts of the sample-wise analysis but are genuinely driven by disease status. Additionally, most functional variants are singletons (observed in only one tissue sample), further supporting the observed genomic and brain-region-specific enrichments.

We believe that these clarifications and revisions address the reviewer's concerns.

Additional comments:

- results on germline are not novel and should probably be reduced in size in the results section.

→ We have significantly reduced the section on germline variant analysis in this revision.

- in the list of genes (supp table), some are falsely reported as dominant, as they are risk factors, e.g. APOE, ABCA7, TREM2 (the latter is AR for Nasu-Hakola disease, risk factor for AD)

→ We appreciate the reviewer's comment. We have corrected them in the supplementary table in this revision.

- in the supp table reporting germline variants, the het/hom status would be useful

→ We appreciate this suggestion. We have added the genotype status (het/hom) to the supplementary table in this revision.

- analysis of Alzheimer genes along with FTD genes with the argument that AD genes cause tauopathy is misleading, as these genes are related to amyloid, not directly to tau, there is no tau pathology without amyloid in patients carrying these variants. Thus, grouping these genes with the FTD genes makes no sense to me.

→ As mentioned above, we have performed this analysis by excluding AD genes, which resulted in a loss of statistical significance for somatic variants in FTD genes in FTD brains (Figure 3). Therefore, we have changed this result in this revision.

- everywhere "deleterious" is used should be "predicted deleterious"

→ We have revised this terminology throughout the manuscript.

- results, germline paragraph, when referring to germline variants, the term mutation should be avoided, if not demonstrated as de novo, prefer the ACMG class (i.e. pathogenic) and the word variant. Overall, the term mutation is not preferred as it can be misinterpreted as pathogenic by readers.

→ We have replaced all instances of the term "mutation" with "variant" in the manuscript.

- SMN analysis: deletions can be missed even by long read seq as they do not easily show differences between SMN1 and 2, and remain the major cause of SMA

→ We acknowledge this limitation. However, these two patients exhibit TDP-43 pathology, which strongly argues against SMA, as TDP-43 pathology is not typically observed in SMA patients. Combined with our long-read WGS data, our results suggest that these two cases are unlikely to have SMA.

- there is a control with Braak stage 5, this is not the best control although it does not alter the results

→ We agree that this individual may not be an ideal control. However, since no somatic variant candidates were identified from this individual, we have decided to retain it in the study.

- supp tables are still in the format TARDBP:NM_007375:exon6:c.G744C instead of TARDBP(NM_007375):c.744G>C

→ We appreciate the reviewer's comment and have updated the format in the supplementary tables accordingly.

Reviewers' Comments:

Reviewer #1 (Remarks to the Author):

The authors have thoroughly revised their manuscript and have addressed my comments and suggestions. I think this paper is important and will be of interest to the field and I recommend publication in Nature Genetics. I just have one minor comment that I think they should consider addressing as an edit to the text.

Regarding the analysis of the TARDBP p.L248F variant, the data presented are sufficient to support that the variant exists in the tissue sample, but the interpretation of the phospho-TDP-43 (pTDP-43) results seems to be stronger than what the results demonstrate. As different brain regions arise from neurodevelopmental processes and are not linearly correlated with physical distance from the motor cortex, it would be more correct to simply state that pTDP-43 is highest in the region in which the somatic variant is observed. However, as pTDP-43 levels likely correlate with ALS pathology, and ALS pathology is generally localized to the motor cortex and spinal cord, I hesitate to draw any firm conclusions from this analysis. Without further functional data on this variant, it is difficult to accept a claim that the variant is causal, even if it is within the TARDBP gene. I think for the purposes of this manuscript that it would be acceptable to state that the individual has a somatic variant in TARDBP that is potentially pathogenic, but that causality will require further investigation. It is possible that focal somatic mutation and focal pTDP-43 are functionally correlated, without more evidence of focal spread this experiment is more of a case study than a conclusion.

Please consider editing the text to only state that the various brain regions tested displayed different levels of pTDP-43, and that the highest levels of pTDP-43 were observed in the same tissue sample as the somatic variant in TARDBP. Representative images for IHC of pTDP-43 in various brain regions would be informative, if available.

We thank the reviewer for these suggestions. We have revised the main text to reflect that different brain regions displayed varying levels of pTDP-43, with the highest levels observed in the region harboring the somatic *TARDBP* variant. Representative IHC images have been included in Extended Data Fig 2 as suggested.

Reviewer #3 (Remarks to the Author):

The authors have greatly improved the manuscript and I am satisfied with most of the answers. To me, as the article stands now, the main conclusions from this article are: (i) replication study on rates of pathogenic germline variants in FTD/ALS genes, (ii) examples of somatic variants that appear as good candidates to explain the disease (e.g. DYNC1H1) and (iii) identification of somatic C9 expansions.

Regarding the main message on the rates of somatic variants in selected genes in ALS/FTD cases compared to controls, this is undoubtedly a very interesting input, but I am still not fully

convinced. If statistics are now much clearer and greatly reduce risks of biases, I am not fully convinced that most of these somatic variants are not secondary to neurodegeneration instead of causal. The major argument for causality is certainly the presence in different cell types, neuronal and non-neuronal. However, this has been assessed in a very limited number of individuals, and I would like to ask whether the few dots in the neuronal populations actually separate significantly from the background noise in the sequencing data, despite the fact that the authors state that many cells should carry this variant to observe such a VAF. This experiment is missing a control from other ALS/FTD cases. There is a lot of emphasis on this experiment and a lot of discussion, including the fact that neuronal cells may have died, explaining their low proportion compared to glial cells, which is totally meaningful, if true, but can we make sure that neuronal cells do carry these variants, along the glial cells? That would decrease my concerns by a lot, if the authors can show it. Indeed, the rest of the data remains compatible with a cellular death/apoptosis signal or revealing low-level glia-specific mutations (if they would be glia-specific), including the mutational signatures, the presence of an enrichment in non ALS/FTD genes somatic variants, the only borderline significance of the association of ALS gene somatic variants in ALS cases (are p-values presented after correction for multiple testing?), while the regional distribution of somatic variants does not argue in favor of one or another hypothesis, as somatic variants may be enriched in ALS/FTD affected regions because they are causal, or because they are secondary to neurodegeneration. If this is not technically possible, I would suggest tempering the message related to this part, or, at least, showing the BAM extracts supporting the variants in both directions. If neuronal and non neuronal cells actually show some proportion of true variants, then I agree with the conclusions of the authors.

We thank the reviewer for this insightful comment and suggestion. To further validate the presence of the reported somatic variants in neurons, we performed additional amplicon sequencing for several predicted deleterious somatic variants on sorted neurons (500 neurons) from two sALS cases that do not carry these somatic variants, as negative controls (five replicates per case). As shown in Rebuttal Figure. 1, neurons from the ALS cases showed clear variant-supporting reads in both directions, with the VAFs well above the background noise observed in the control samples. To quantify these differences, we conducted 10,000 group-label permutation tests comparing the mean VAFs between variant carriers and control groups and calculated empirical p -values. All tested variants showed significant differences, confirming the presence of these somatic variants in neurons from ALS cases. These results have been incorporated into the manuscript as Extended Data Fig. 10d. We have added a sentence in the Discussion noting that the cell-type analysis was based on a limited number of variants.

Figure 1. Visualization of variant-supporting reads at four somatic variant sites from validation sequencing of sorted neurons. Each subpanel shows a zoomed-in view of variant-supporting reads, with a consistent scale applied per variant. Pink and light blue horizontal lines represent the reads from positive and negative strands, respectively. Empirical p -values were calculated using a group label permutation test (10,000 iterations) comparing the mean VAF between variant carrier and control groups.

Regarding the p -values in Figure 4 of the main manuscript, we reported unadjusted p -values. Although multiple statistical tests were conducted, these analyses were not fully independent, but rather addressed distinct aspects of a unified biological hypothesis—whether somatic variant burden differs between disease and control groups across specific gene sets, genomic context, and brain regions. Key variables such as disease type (ALS or FTD), gene group (e.g., total genes, ALS/FTD genes), brain region, and genomic category (e.g., exonic, protein-altering) represent biologically distinct contexts with differing background mutation rates and model error structures. Therefore, applying a single, global multiple testing correction would not appropriately account for this heterogeneity. Moreover, overlapping categories (e.g., overlapping genomic regions such as ‘all’, ‘exonic’, ‘protein-altering’) further complicate the use of a unified multiple testing. Nevertheless, when we applied the Benjamini–Hochberg FDR correction separately within individual comparisons stratified by disease groups, target regions, or genomic context, several results remained statistically significant. These included:

- Total somatic variant burden within ALS/FTD genes in ALS cases ($adj.p = 0.025$)
- Exonic somatic variant burden across total genes in FTD cases ($adj.p = 0.011$)

- Protein-altering somatic variant burden within ALS/FTD genes in ALS cases (*adj.p* = 0.034)
- Protein-altering somatic variant burden across total genes in both ALS (*adj.p* = 0.049) and FTD cases (*adj.p* = 0.041)
- Exonic and protein-altering somatic variant burden across total genes in the PFC of FTD cases (*adj.p* = 0.013 and *adj.p* = 0.010)
- Total somatic variant burden within ALS/FTD genes in the motor cortex of ALS cases (*adj.p* = 0.049).

These findings continue to support our central hypothesis regarding increased somatic variant burden in disease brains.

I was also interested and surprised to read that C9ORF72 expansions might arise from a small normal allele. However, while the authors are very affirmative in the results section, they acknowledge that haplotyping was not clear, in the discussion. As this result is surprising, compared to what we know from this gene and other expansions, this limitation should appear along the results, not only in the discussion section.

We have added the following sentence to the Results section to clarify the limitation of our current approach:

“though future analysis of this and other cases is needed to further characterize this possibility.”

Other comments:

- It seems that there is one control with a germline pathogenic variant (C9 expansion?), but I could not find this control individual in supp tables (S4/S1).

We apologize for the confusion. One control individual carries a germline splicing variant in NEK1 (c.1665+2T>C), which has been previously reported in an ALS case and classified as likely pathogenic following ACMG guidelines (PMID: 33589474). This individual is included in the first tab of Supplemental Table 4. We originally described this case in the main text when discussing germline variants in ALS/FTD genes, but that sentence was removed in the revised version to streamline the paragraph in response to earlier reviewer feedback.

- C9 genotyping results of controls are not present in supp files.

We apologize for omitting the C9 genotyping results for control samples in the previous version. All control cases included in this study were confirmed to be negative for C9ORF72 repeat expansions based on our repeat-primed PCR assays. We have now included these results in the updated Supplemental Table 1.

- Germline variants: It is not clear how variants of unknown significance are in the “known pathogenic” variants supp tables.

We apologize for the confusion. These variants were initially predicted to be deleterious but were later reclassified as VUS according to ACMG guidelines. We have updated Supplemental Table 3 and listed these variants as predicted deleterious variants.

- It seems like NEK1 truncating variant are considered pathogenic, although they are risk factors, not fully penetrant; the ACMG criteria do not apply to risk factors.

We have changed the classification of this variant to VUS.

- Germline variants part: the classification of variants as known pathogenic versus novel is not very relevant in the whole paragraph, it should be pathogenic/likely pathogenic compared to VUS. Whatever a truncating GRN variant is novel or not, it is likely pathogenic (if predicted to trigger NMD), knowing that it is novel or not is not relevant as lab scientists would rate them as likely pathogenic whatever their known/novel status. These pathogenic variants among the novel variants are biasing these tests. These tests are interesting as they show what might be remaining, the potential of reclassification, but as it includes variants that are already clearly likely pathogenic, it is not very meaningful. The potential of reclassification is actually in some proportion of the VUS, i.e. among the remaining predicted deleterious variants.

We thank the reviewer for this helpful comment. We have removed the distinction between known and novel variants in our classification. All germline variants are now grouped as either pathogenic or predicted deleterious (the latter encompassing VUS). Figures 2a and 2d have been updated accordingly to reflect this revised classification. In addition, we have added a new column to Supplemental Tables 3 and 4 to indicate the novelty of each variant for reference. We point out that the term VUS has been developed for returning results from clinical testing, where there are understandably more restrictive criteria for defining pathogenicity. In this research study, we feel that showing case/control enrichment of pathogenic or predicted deleterious variants is a reasonably accepted standard for evidence of a biological effect, even if some of these variants do not satisfy all criteria to be returned in a clinical setting; and so it is appropriate to classify them in that way.

- Germlines variants part: "Consistent with previous reports, multiple cases showed evidence for possible oligogenic inheritance, including several with both C9ORF72 expansions and other variants (Fig. 2d and Supplementary Table 3)": as there are not so many examples here, it would be useful to specify how many cases have multiple pathogenic variants, instead of "multiple cases". In addition, counting only the (likely) pathogenic variants would be more appropriate, as we don't know the potential of all the VUS. Counting the VUS overestimates this proportion of cases with "double" variants.

We have revised the sentence to explicitly state that one case carries both a C9ORF72 expansion and a pathogenic variant in ANXA11, and that 12 additional cases carry combinations of pathogenic and predicted deleterious variants. While the additional variants are

currently classified as VUS, they are all predicted to be deleterious, and we believe it is informative to highlight them in both the main text and figure.

- Clarification of subgroups of genes is required. Sometimes we can read dementia genes and sometimes, neurodegeneration (e.g. "Our MIP panel contained not only ALS/FTD genes but also genes involved in other dementias. We first focused on somatic variants in all the targeted neurodegeneration genes."). CADASIL (NOTCH3) is not a neurodegenerative disease. Neurodevelopmental diseases associated with EP300 are not neurodegenerative at all, I don't understand why this gene is in the list.

To improve clarity, we have revised the manuscript to refer to these as "neurodegeneration/dementia-related genes" rather than "neurodegeneration-related genes." We agree that *EP300* is not traditionally classified as a neurodegeneration-related gene, although some studies have suggested its involvement in memory, cognition, and potential links to neurodegenerative pathways. Importantly, the *EP300* somatic variants in our study were detected only in control cases, so their inclusion does not impact the significance of our enrichment analyses or alter our overall conclusions.

- It still reads strange that the authors tried to exclude spinal muscular amyotrophy by long read sequencing, this method can miss the recurrent pathogenic deletions and this case is not SMA as it shows TDP43 pathology, indeed, maybe this sentence should be skipped in the main text?

We have removed the sentence from the main text as suggested.

- It is difficult to read figure 3c, could the authors apply colors that are easier to separate visually? The different levels of yellow are difficult to read.

We have revised the color scheme in Figure 3c to improve visual contrast.

- I would suggest to temper the message based on the *LMNA* variant, as there is no evidence that it is pathogenic, although the case is interesting.

While this *LMNA* variant has not been previously reported, it is predicted to be deleterious. We agree that its pathogenicity remains uncertain, but the main text includes only a single sentence describing this variant, without overstating its significance, and we believe this provides appropriate context without making a strong claim.

Although the LMNA (p.H566Y) variant was not previously reported, LMNA variants cause autosomal dominant laminopathies including Hutchinson-Gilford progeria and congenital muscular dystrophy, which are characterized by congenital defects and early lethality.